# Strategic Costs of Perceived Bias in Fair Selection

**L. Elisa Celis**
Yale University

**Lingxiao Huang**
Nanjing University

**Milind Sohoni**
IIT Bombay

**Nisheeth K. Vishnoi**
Yale University

## Abstract

Meritocratic systems, from admissions to hiring, aim to impartially reward skill and effort. Yet persistent disparities across race, gender, and class challenge this ideal. Some attribute these gaps to structural inequality; others to individual choice. We develop a game-theoretic model in which candidates from different socioeconomic groups differ in their perceived post-selection value—shaped by social context and, increasingly, by AI-powered tools offering personalized career or salary guidance. Each candidate strategically chooses effort, balancing its cost against expected reward; effort translates into observable merit, and selection is based solely on merit. We characterize the unique Nash equilibrium in the large-agent limit and derive explicit formulas showing how valuation disparities and institutional selectivity jointly determine effort, representation, social welfare, and utility. We further propose a cost-sensitive optimization framework that quantifies how modifying selectivity or perceived value can reduce disparities without compromising institutional goals. Our analysis reveals a perception-driven bias: when perceptions of post-selection value differ across groups, these differences translate into rational differences in effort, propagating disparities backward through otherwise "fair" selection processes. While the model is static, it captures one stage of a broader feedback cycle linking perceptions, incentives, and outcomes—bridging rational-choice and structural explanations of inequality by showing how techno-social environments shape individual incentives in meritocratic systems.

## 1 Introduction

Meritocratic selection systems, used by institutions and firms for admissions, hiring, and content curation, aim to allocate opportunities based on observable indicators of ability and effort rather than wealth, identity, or social status. They are widely viewed as promoting fairness and efficiency [41, 71, 37]. Examples include standardized tests such as the SAT and JEE [49, 8], structured interviews and assessments [69, 13], and algorithmic ratings on online platforms [24, 70, 81].

Yet, despite their formal neutrality, these systems often produce significant disparities in representation and outcomes. Women, racial minorities, and lower-income groups are consistently underrepresented in elite universities, leadership roles, and high-paying industries [61, 72]. These gaps persist even when evaluation procedures are blind to group identity, suggesting that there are additional factors that drive inequality in merit-based processes.

One set of explanations points to structural barriers: unequal access to resources that enhance merit (e.g., quality education, extracurricular activities), implicit biases in selection processes, and limited opportunities due to privileged networks [84, 39, 53, 58, 13, 60, 24, 70, 81]. Others suggest that individuals who face the same selection rules may simply make different choices, investing less effort because they perceive lower returns to success due to cultural preferences, opportunity costs, or labor market sorting [31, 11, 38, 63, 12]. These disparities under-utilize talent, reducing innovation, diversity of ideas, and social progress [42, 55, 64]. This raises a central question: *how can differences in perceived opportunity translate into systematic behavioral disparities even when evaluation is symmetric?*

39th Conference on Neural Information Processing Systems (NeurIPS 2025).

Expectations about what selection yields-admission, employment, mobility—are shaped not only by historical inequalities [73, 56, 48], but increasingly by algorithmic tools that mediate labor market signals [2]. Although meritocratic ideals suggest that pay should correlate with skills, productivity, and achievements, empirical studies reveal persistent wage disparities even after controlling for factors such as occupation, education, experience, and hours worked [38, 86, 83, 61, 80, 75, 65, 40]. For example, in the U.S., women earned just 83.1% of what men earned in 2021, despite outnumbering men in the college-educated labor force [80, 26]. Similar wage gaps persist across racial, class, caste, and ethnic lines, with Black, Hispanic, and Indigenous workers earning less than White and Asian peers in comparable roles [65]. Large language models (LLMs) may further exacerbate these disparities. Recent studies show that when asked for job or salary recommendations, LLMs return systematically different responses across demographic groups, even when qualifications are held constant [2]. Such signals can distort perceived opportunity and disincentivize effort long before any selection decision occurs. Taken together, these findings suggest a perception-driven bias: social and algorithmic cues about post-selection value shape pre-selection investment, reinforcing group-level disparities even under ostensibly meritocratic systems.

**Our contributions.** We introduce a game-theoretic model of meritocratic selection in which candidates from two groups differ in their perceived value of being selected. This model integrates contest theory with models of structural bias from algorithmic fairness and captures how valuation disparities influence effort, merit, and selection outcomes. While the model is static, it represents one stage of a broader feedback process linking perceptions, incentives, and outcomes. Our main contributions are:

1. **Modeling.** We formulate a two-group contest in which $n$ rational agents, divided into groups $G_1$ and $G_2$ (with proportion $\alpha \in (0,1)$), compete for $c \cdot n$ positions. Group-specific valuations follow distributions $p_1$ and $p_2$, with $p_2$ modeled as a $\rho$-biased version of $p_1$ for $\rho \in (0,1]$ [44], representing structural disparities. Each candidate chooses effort based on their valuation to maximize their expected payoff, which is then converted into observable merit used for selection.

2. **Equilibrium characterization.** We prove the existence and uniqueness of a symmetric Nash equilibrium in the large-agent limit, and express the equilibrium thresholds for both groups in terms of $(c, \alpha, p_1, p_2)$ (Theorem 3.1). We further show that the equilibrium in the finite-$n$ setting converges to this solution at rate $O(\sqrt{\log n / n})$.

3. **Micro to macro analysis of metrics.** Using the equilibrium solution, we derive closed-form expressions for key performance metrics—group-wise representation ratio $r_{\mathcal{R}}$, social welfare ratio $r_{\mathcal{S}}$, and institutional revenue in the case where $p_1$ is uniform and $p_2$ is $\rho$-biased (Proposition 4.1). These expressions reveal how small changes in $\rho$, $c$ or $\alpha$ can produce non-linear shifts in outcomes.

4. **Fairness-aware interventions.** We formulate a constrained optimization problem (Problem (6)) that allows institutions to trade off between increasing selectivity ($c$) and reducing valuation bias ($\rho$) under fairness constraints (e.g., 80%-rule). We solve this problem in closed form for linear cost functions and characterize when each intervention is most cost-effective (Figure 4).

Taken together, our framework provides a quantitative lens on how structural or algorithmic biases in perceived value can rationally produce effort and outcome disparities in meritocratic systems, and offers tools to design interventions that enhance both representation and efficiency.

**Related work.** Our work connects three areas: economic theories of meritocracy, game-theoretic models of contests, and algorithmic fairness. Social scientists have long examined the tension between meritocratic ideals and persistent disparities in outcomes, including gender and racial pay gaps [56, 73, 38, 15]. While prior models of statistical discrimination explain disparities through institutional uncertainty or group-dependent beliefs—often despite equal agent quality [3, 66, 21, 23, 5, 46]—our model assumes perfect institutional information and symmetric evaluation. Instead, we show how valuation asymmetries alone can induce disparities in effort and selection outcomes. From a game-theoretic perspective, our setting builds on all-pay auctions and Tullock contests, which model competitive effort under asymmetry [76, 29, 27]. However, these models rarely consider group-level valuation differences. To our knowledge, we are the first to study equilibrium behavior in large-agent contests with asymmetric valuation distributions across groups. Finally, our bias model extends work in algorithmic fairness that explores valuation gaps and signal noise [45, 28, 18], but previous strategic classification models typically lack inter-agent competition or valuation-based disparities [14, 29]. Thus, our work offers a novel integration of asymmetric group valuations into competitive contest frameworks, with implications for equilibrium behavior, fairness, and institutional design. See Section A for further discussion and detailed comparisons.

## 2 Model and metrics

We consider a population of $n$ agents competing for $k = cn$ indistinguishable spots, where $c \in (0, 1)$ denotes the selection fraction. Agents are partitioned into two groups: an *advantaged* group $G_1$ of size $(1 - \alpha)n$ and a *disadvantaged* group $G_2$ of size $\alpha n$, where $\alpha \in (0, 1)$. Each agent $i \in G_\ell$ ($\ell \in \{1, 2\}$) has a valuation $v_i \sim p_\ell$ supported on $\Omega_\ell \subseteq \mathbb{R}_{\geq 0}$. We model systemic disadvantage via a scaling of valuations: if $p_1$ is the valuation distribution for $G_1$, then $G_2$ has valuations drawn from $p_2(v) = \frac{1}{\rho}p_1(\frac{v}{\rho})$, where $\rho \in (0, 1]$ captures the degree of bias, implying $\mathbb{E}_{v \sim p_2}[v] = \rho \mathbb{E}_{v \sim p_1}[v]$. For instance, if $p_1$ is uniform on $[0, 1]$, then $p_2$ is uniform on $[0, \rho]$. Such a bias model has been widely studied in the fairness literature [45, 16, 19] and serves as a benchmark for understanding systemic disparities. Section D.1 discusses extensions where the valuation distributions $p_1$ and $p_2$ are truncated Gaussians, and the bias parameter $\rho$ may also be drawn from a distribution, introducing stochastic heterogeneity across candidates. These structural disparities across groups may stem from unequal access to opportunity, differences in marginal returns, labor market discrimination, or broader societal narratives about value; see also Remark 2.1 for practical scenarios.

Each agent also has an initial ability $a_i \sim p_a$ supported on $\Omega_a \subseteq \mathbb{R}_{\geq 0}$, drawn independently. We assume that $p_a$ is identical across groups. Agents choose policies $A_i : \Omega_\ell \times \Omega_a \to \mathbb{R}_{\geq 0}$ that map their type $\theta_i = (v_i, a_i)$ to an exerted effort $e_i = A_i(\theta_i)$. The agent's score is $s_i = e_i + a_i$. A strictly increasing merit function $m : \mathbb{R}_{\geq 0} \to \mathbb{R}_{\geq 0}$ maps scores to merit. The institution selects the $k$ agents with the highest merit values. Each agent's payoff is

$$f_i(v_i, a_i, e_i; s_1, \ldots, s_n) = \mathbb{I}(m(s_i) \text{ among top } k) \cdot v_i - (s_i - a_i) = \mathbb{I}(s_i \text{ among top } k) \cdot v_i - (s_i - a_i),$$

where the second equality uses the strict monotonicity of $m$. Agents know $n$, $k$, $p_1$, $p_2$, $p_a$, their group identity, and their own type $\theta_i = (v_i, a_i)$, but not others' types. Let $A = (A_1, \ldots, A_n)$ denote the joint policy profile. The probability that agent $i$ is selected after exerting effort $e$ is

$$P_i(e; a_i, A_{-i}) = \mathbb{P}\left(e + a_i \text{ is among the top } k \text{ scores of } \{A_j(\theta_j) + a_j\}_{j \neq i} \cup \{e + a_i\}\right),$$

where $\theta_j = (v_j, a_j)$ are drawn i.i.d. from the respective group distributions. The expected payoff is

$$\pi_i(v_i, a_i, e; A_{-i}) = \mathbb{E}_{s_j}[f_i(v_i, a_i, e; s_1, \ldots, s_n)] = P_i(e; a_i, A_{-i}) \cdot v_i - e.$$

A policy profile $A$ is a Nash equilibrium (NE) if, for all $i$, $v$, $a$, and $e$,

$$\pi_i(v, a, e; A_{-i}) \leq \pi_i(v, a, A_i(v, a); A_{-i}). \tag{1}$$

We implicitly assume that agents act rationally and strategically to maximize their expected payoffs, using their knowledge of the contest structure to compute the NE policy $A$ [74]. For simplicity, we sometimes assume that $p_a$ is a point mass at 0, so that policies depend only on valuations.

A special case of our model generalizes the classical *undifferentiated contest* (where $p_1 = p_2$), which has been extensively studied [76, 52, 82]. To the best of our knowledge, our work is the first to study strategic asymmetries arising from valuation differences in settings where group sizes are known and fixed, a structure commonly seen in admissions and hiring. We provide detailed comparisons with prior works [1, 29, 27] in Section A.1. Remark E.7 discusses extensions to multi-group settings and to heterogeneous effort-to-merit mappings, where each individual may convert effort into merit at a different (non-linear) rate.

**Metrics.** We study three metrics to evaluate fairness, efficiency, and institutional outcomes under a given policy $A$. Define $\mathcal{R}_\ell(A)$ as the (random) fraction of agents selected from group $G_\ell$. The *representation ratio* is $r_\mathcal{R}(A) := \mathbb{E}\left[\min\left\{\frac{\mathcal{R}_1(A)}{\mathcal{R}_2(A)}, \frac{\mathcal{R}_2(A)}{\mathcal{R}_1(A)}\right\}\right]$,[1] a metric commonly used in the fairness literature [22, 6, 17]. $r_\mathcal{R}(A) \in [0, 1]$, and low values indicate underrepresentation of one group. Building on standard notions of allocative efficiency [67], define group-wise social welfare as $\mathcal{S}_\ell(A) := \frac{1}{|G_\ell|}\sum_{i \in G_\ell}(\mathbb{I}(i \text{ selected}) \cdot v_i - e_i)$. The *social welfare ratio* is $r_\mathcal{S}(A) := \mathbb{E}\left[\min\left\{\frac{\mathcal{S}_1(A)}{\mathcal{S}_2(A)}, \frac{\mathcal{S}_2(A)}{\mathcal{S}_1(A)}\right\}\right]$. This metric measures disparities in average payoffs between groups. Define the *average revenue* as $\mathcal{RV}(A, m) := \mathbb{E}\left[\frac{1}{k}\sum_{i \text{ selected}} m(s_i)\right]$, capturing the average merit of selected agents and aligns with institutional objectives [33, 34].

---

[1] We consider the $\min$ operator since randomness can occasionally lead to equal or even higher representation for the disadvantaged group $G_2$. This becomes vanishingly rare as $n \to \infty$.

We analyze how the NE policy $A$ and associated metrics vary with the bias parameter $\rho$ and the selection fraction $c$. These parameters capture systemic disparities and selection competitiveness, respectively. Even for simple instances, deriving closed-form NE strategies under asymmetric valuations is significantly more complex than in the undifferentiated case. A worked example illustrating these challenges is provided in Section B.

**Remark 2.1** (**Practical settings with group-based valuation bias**). *Our model captures environments where disadvantaged groups anticipate lower returns from being selected—due to structural barriers, social context, or biased algorithmic feedback (see Section 1). For example, as discussed in Section 1, persistent wage gaps across gender and race—even after accounting for qualifications—as well as biased algorithmic recommendations can diminish expectations about the benefits of selection. These lower expectations can rationally reduce pre-selection effort, even under formally fair rules, and represent the main regime we study. That said, in domains such as credit, housing, or education, disadvantaged groups may instead face higher marginal returns due to limited outside options; this can be modeled by reversing which group has the compressed valuation distribution.*

## 3 Theoretical results: Nash equilibrium and metrics for large $n$

The first question we address is whether a Nash equilibrium (NE) policy exists for the two-group contest and how it can be computed. While characterizing NE policies for finite $n$ is challenging, the large-population limit ($n \to \infty$) reveals an interesting and tractable structure. The following result shows that in this limit, it is possible to describe how the strategies of the two groups, $G_1$ and $G_2$, converge. However, the absence of an explicit policy formulation for finite $n$ complicates the interpretation of convergence, which we address by adopting the notion of an approximate NE policy.

**Definition 3.1** ($\varepsilon$-**Nash equilibrium [51]**). *For an $\varepsilon > 0$, a policy $A$ is said to be an $\varepsilon$-NE policy if for any $\ell \in \{1, 2\}$, agent $i \in G_\ell$, type $(v, a) \in \Omega_\ell \times \Omega_a$, and effort $e \geq 0$, the following condition is met: $\pi_i(v, a, e; A_{-i}) \leq \pi_i(v, a, A_i(v, a); A_{-i}) + \varepsilon$.*

An $\varepsilon$-NE permits stability violations up to $\varepsilon$, with exact NE recovered when $\varepsilon = 0$. This notion allows us to formalize the convergence of NE policies in the following theorem.

**Theorem 3.1** (**The two-group contest: Large $n$ limit**). *Let $\alpha, c \in (0, 1)$. For $\ell = 1, 2$, let $p_\ell$ be a density supported on a domain $\Omega_\ell \subseteq \mathbb{R}_{\geq 0}$. Let $p_a$ be a density supported on a domain $\Omega_a \subseteq \mathbb{R}_{\geq 0}$. Let $m : \mathbb{R}_{\geq 0} \to \mathbb{R}_{\geq 0}$ be a merit function that is strictly increasing. For $\ell = 1, 2$, let $F_\ell$ be a cumulative density function (CDF) of the sum of valuation and initial ability such that for any $\zeta \in \mathbb{R}_{\geq 0}$, $F_\ell(\zeta) = \Pr_{v \sim p_\ell, a \sim p_a} [v + a \leq \zeta]$. Suppose $(\Omega_1 \cup \Omega_2) + \Omega_a$ is connected[2] and densities $p_1, p_2, p_a$ are positive at any point of their own domains. Let $t$ be the unique solution to the equation*

$$(1 - \alpha)F_1(\zeta) + \alpha F_2(\zeta) = 1 - c. \tag{2}$$

$$\text{Define } s(v, a) := 0 \text{ if } v + a < t \text{ and } s(v, a) := \max\{t - a, 0\} \text{ if } v + a \geq t \tag{3}$$

*and let policy $A$ be: each agent $i \in G_1$ uses the restriction $A_i = s|_{\Omega_1 \times \Omega_a}$, while each agent $j \in G_2$ uses the restriction $A_j = s|_{\Omega_2 \times \Omega_a}$. Moreover, this solution $t$ is monotonically decreasing with $c$.*

*This $A$ is the unique policy such that there exists an infinite sequence $A^{(1)}, \ldots, A^{(n)}, \ldots$, where $A^{(n)}$ is a policy for the two-group contest with $n$ agents characterized by a threshold function $s^{(n)} : (\Omega_1 \cup \Omega_2) \times \Omega_a \to \mathbb{R}_{\geq 0}$, such that the followings hold: (1) For every integer $n \geq 1$, agent $i \in G_1$ uses the restriction $A_i^{(n)} = s^{(n)}|_{\Omega_1 \times \Omega_a}$, while each agent $j \in G_2$ uses the restriction $A_j^{(n)} = s^{(n)}|_{\Omega_2 \times \Omega_a}$ and $\lim_{n \to \infty} s^{(n)} = s$; (2) Every $A^{(n)}$ is an $\varepsilon_n$-NE policy with $\lim_{n \to \infty} \varepsilon_n = 0$.*

This theorem characterizes the policy $A$ through a threshold function $s$ parameterized by $t$, establishing that there exists a sequence of policies $A^{(1)}, \ldots, A^{(n)}, \ldots$ that converge towards $A$, progressively approximating it. In Section E, we provide the explicit form of policies $A^{(n)}$ characterized by $s^{(n)}$ (in Theorem E.2) and a complete proof. Note that the value $t$ defines the threshold function $s$, and consequently, the policy $A$. Therefore, we focus below on analyzing $t$. We first remark on the uniqueness of $t$ guaranteed by Equation (2) under certain assumptions on the domains and densities (see Lemma E.1), which are natural in real-world contexts and satisfied by the distributions discussed in Section 2. These assumptions ensure that each $F_\ell$, being a CDF, is strictly monotonic over its

---

[2]Here, symbol $+$ represents the Minkowski sum of domains, where $A + B = \{a + b : a \in A, b \in B\}$.

domain $\Omega_\ell + \Omega_a$. Consequently, the combined CDF $(1-\alpha)F_1 + \alpha F_2$ must also be strictly monotonic over the connected domain $(\Omega_1 \cup \Omega_2) + \Omega_a$, which guarantees the uniqueness of the solution $t$.

A key takeaway from Theorem 3.1 is that, in the large-$n$ limit, each agent makes a binary decision based on their combined valuation and initial ability $v + a$: either exert effort $\max\{t - a, 0\}$ to ensure a score of at least $t$, or put in no effort at all. The threshold $t$, determined by Equation (2), plays a central role in this decision. It is chosen such that a fraction $1 - c$ of the agent population has $v + a \leq t$, meaning that exactly a fraction $c$ is expected to exert effort and compete. Thus, $t$ implicitly encodes the level of competition: higher values of $t$ reflect more intense competition, requiring higher effective scores for selection.

**Computing the threshold in the NE policy.** Equation (2) is crucial for applying Theorem 3.1, as it enables the explicit computation of $t$, facilitating analysis in Section 4. Let $F_a$ be the CDF of the initial ability.[3] Note that $F_\ell(\zeta) = \Pr_{v \sim p_\ell, a \sim p_a}[v + a \leq \zeta] = \int_{\Omega_\ell} p_\ell(v) F_a(\zeta - v) dv$. Thus, Equation (2) is equivalent to $(1-\alpha) \int_{\Omega_1} p_1(v) F_a(\zeta - v) dv + \alpha \int_{\Omega_2} \alpha p_2(v) F_a(\zeta - v) dv = 1 - c$. Specifically, when $p_2(v) = \frac{1}{\rho} p_1\left(\frac{v}{\rho}\right)$ for some $\rho \in (0, 1]$, this equation becomes becomes

$$(1-\alpha) \int_{\Omega_1} p_1(v) \cdot F_a(\zeta - v) dv + \frac{\alpha}{\rho} \int_{\Omega_2} p_1(\frac{v}{\rho}) F_a(\zeta - v) dv = 1 - c. \qquad (4)$$

We illustrate how to use this equation to compute the explicit form of $t$. Let $p_1$ be uniform on $\Omega_1 = [0, 1]$, $p_2$ be uniform on $\Omega_2 = [0, \rho]$, and $p_a$ be uniform on $\Omega_a = [0, 1]$. Such uniform densities are often used in studies and analyses [45, 16, 19], serves as a fundamental benchmark for insights into decision-making, allocation mechanisms, and strategic behavior. Moreover, domain $(\Omega_1 \cup \Omega_2) + \Omega_a = [0, 2]$ is connected for any value of $\rho \in (0, 1]$, satisfying assumptions in Theorem 3.1. Since $p_1(v) = 1$ for $v \in [0, 1]$, $p_2(v) = \frac{1}{\rho}$ for $v \in [0, \rho]$ and $F_a(\zeta - v) = \min\{1, (\zeta - v)_+\}$ (Here, $x_+ = \max\{0, x\}$), Equation (2) reduces to $\int_0^1 (1-\alpha) \cdot \min\{1, (\zeta - v)_+\} dv + \int_0^\rho \frac{\alpha}{\rho} \cdot \min\{1, (\zeta - v)_+\} dv = 1 - c$. Consequently, the solution $t$ is a piecewise function of parameters $\rho$, $c$, and $\alpha$; see Proposition F.4 for its explicit form. Here, we illustrate the behavior of $t$ over a representative range where $\alpha = 0.5$ (equal-sized groups) and $0 < c \leq \frac{1}{4}$ (high selectivity).

$$t = 2 - 2\sqrt{c} \text{ if } \rho < 1 - 2\sqrt{c} \text{ and } t = \frac{1 + 3\rho}{1 + \rho} - \frac{\sqrt{4c\rho(1+\rho) - \rho(1-\rho)^2}}{1 + \rho} \text{ if } \rho \geq 1 - 2\sqrt{c}. \qquad (5)$$

Note that, while $t$ is the same for both groups, it may happen that $t > 1 + \rho$ (when $\rho < 1 - 2\sqrt{c}$), implying that no agent in $G_2$ exerts any effort. We note that for other densities, such as piecewise linear and polynomial, including Pareto, explicit forms of the solution $t$ are achievable. For instance, consider a Pareto distribution defined by $p_1(v) = \frac{2}{v^3}$ for $v \geq 1$, a $\rho$-biased density $p_2(v) = \frac{1}{\rho} p(\frac{v}{\rho})$, and $p_a$ is a point mass at 0. Here, $t$ can be explicitly calculated: If $\alpha + c - 1 > 0$ and $\rho < \sqrt{(\alpha + c - 1)/\alpha}$, then $t = \rho\sqrt{\alpha/(\alpha + c - 1)}$ otherwise, $t = \sqrt{(1 - \alpha + \alpha\rho^2)/c}$.

**Computing the metrics.** We next ask whether the key metrics associated with the NE policy $A$ from Section 2 can be computed in closed form. Given the simple threshold structure of $A$, these metrics can indeed be expressed as functions of the scalar threshold $t$. However, for general densities $p_1$ and $p_2$, the expressions for the representation ratio $r_\mathcal{R}(A)$ and social welfare ratio $r_\mathcal{S}(A)$ become more complex due to the presence of the $\min$ operator and the convolution involved between $p_\ell$ and $p_a$ (see Theorem F.3). For clarity, we focus on the special case where $p_2$ is a $\rho$-biased version of $p_1$ and $p_a$ is a point mass at 0, which admits more tractable expressions. Since $t$ depends on the parameters $\rho$, $c$, and $\alpha$, the resulting metrics are also functions of these parameters. The following theorem characterizes both the explicit forms and their monotonicity behavior.

**Theorem 3.2 (Metrics and their monotonicity).** *Assume $p_2(v) = \frac{1}{\rho} p_1\left(\frac{v}{\rho}\right)$ for some $\rho \in (0, 1]$ and $p_a$ is a mass point at 0. Let policy $A$ be defined as in Theorem 3.1, characterized by $t$ being the unique solution of Equation* (4)*. Then for any density $p_1$,*

$$r_\mathcal{R}(A) = \frac{1 - F_1(t/\rho)}{1 - F_1(t)}, \quad r_\mathcal{S}(A) = \frac{\rho \int_{t/\rho}^\infty (v - t/\rho) p_2(v) dv}{\int_t^\infty (v - t) p_1(v) dv}, \quad \text{and } \mathcal{RV}(A, m) = m(t).$$

*Moreover, $r_\mathcal{R}(A)$ and $r_\mathcal{S}(A)$ are monotonically increasing w.r.t. $\rho$, while $\mathcal{RV}(A, m)$ is monotonically increasing w.r.t. $\rho$ and monotonically decreasing w.r.t. $c$ and $\alpha$, for any merit function $m$.*

---

[3]Throughout this paper, we extend the domain of a CDF $F$ to the entire real line $\mathbb{R}$ such that $F$ is monotonically non-decreasing, with $F(-\infty) = 0$ and $F(\infty) = 1$.

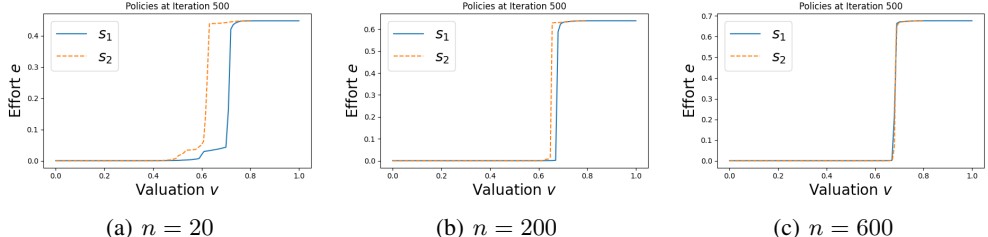

|  | (a) $n = 20$ | (b) $n = 200$ | (c) $n = 600$ |
|---|---|---|---|

Figure 1: Evolution of group effort policies at iteration 500 for various $n$ with $\rho = 0.8$ and $c = 0.2$.

The proof is deferred to Section F. Combined with the closed-form expression for $t$, this result enables direct computation of the metrics, which we use for contest analysis in Section 4. Notably, $r_{\mathcal{R}}(A)$ and $r_{\mathcal{S}}(A)$ increase with $c$, while $\mathcal{RV}(A, m)$ decreases, highlighting both benefits and trade-offs for the institute. The qualitative behavior of $r_{\mathcal{R}}(A)$ and $r_{\mathcal{S}}(A)$ w.r.t. $c$ and $\alpha$, however, depends on the underlying densities; see Remark F.2.

**Discussion of NE policies for finite $n$.** To assess the robustness and practical relevance of our theoretical results, we study the closeness between the finite-$n$ NE policy and the infinite-population threshold policy $s$ defined in Equation (3) (see Section C). We propose a dynamic procedure (Algorithm 1) that initializes with $s_1 = s_2 = s$ for groups $G_1$ and $G_2$, and iteratively updates them.

We simulate this dynamics under the setting $p_1 = \text{Unif}[0, 1]$, $p_2 = \text{Unif}[0, \rho = 0.8]$, $p_a = \delta_0$, with $c = 0.2$, $\alpha = 0.5$, and $n = 20, 200, 600, 1200$, running 500 iterations in each case. Figure 1 shows representative results; full plots are in Figures 5–8.

The simulations show that even moderate population sizes ($n \geq 600$) yield policies closely tracking the infinite NE, validating its use as a practical approximation. We also observe group-level differences in convergence speed and stability (Figure 9), with smoother and faster stabilization as $n$ increases.

Finally, using the proof of Theorem 3.1, we establish that the finite-$n$ NE policy is $O(\log n/n)$-close in value and yields an $O(\sqrt{\log n/n})$-NE. Concretely, this means that for large $n$, the finite policy takes the form $s^{(n)} = 0$ for $v < t - O(\sqrt{\log n/n})$ and $s^{(n)} = t$ otherwise. For general distributions, closeness depends on the density structure and is more involved. Aligning with our empirical observations, these results reinforce the practical relevance of the large-$n$ analysis, which captures the incentive-aligned baseline under strategic behavior. See Section C.2 for details.

**Key ideas in the proof of Theorem 3.1.** The proof involves two main steps: hypothesizing the NE policy structure and verifying that it is indeed an equilibrium. We sketch the core ideas below; a full overview appears in Section E.1. For clarity, we focus on the case where $p_a$ is a point mass at 0.

*Hypothesizing the structure of the NE policy.* The key challenge in characterizing the NE policy lies in the absence of its explicit form for finite $n$. Drawing intuition from the undifferentiated case with density $p$, where the NE policy converges to a threshold function $s(v) = F_p^{-1}(1-c)$ if $v \geq F_p^{-1}(1-c)$ and 0 otherwise, the first idea is to hypothesize that in the two-group case, the NE policies $s_1, s_2$ also take threshold forms with group-dependent thresholds $t_1, t_2$. However, asymmetry in $p_1, p_2$ complicates the expression of winning probabilities $P_i$ and prevents a straightforward computation of $t_1, t_2$. Focusing on the uniform case where $p_1$ is $\text{Unif}[0, 1]$ and $p_2$ is the $\rho$-biased version supported on $[0, \rho]$, the second idea is that in the limit $n \to \infty$, NE stability requires $t_1 = t_2 = t$. If $t_1 > t_2$, then some agents in $G_1$ benefit by reducing their effort to slightly above $t_2$, contradicting NE; a symmetric argument holds if $t_1 < t_2$. Although $s_1$ and $s_2$ are defined on different domains, they can be viewed as restrictions of the same threshold function $s$ characterized by $t$. This justifies setting $s_1 = s_2 = s$ with a shared threshold $t$, and modeling the two-group contest using an effective mixture density $p = (1 - \alpha)p_1 + \alpha p_2$ supported on $\Omega_1 \cup \Omega_2$. As $n \to \infty$, the contest becomes indistinguishable from the undifferentiated case with $p$, yielding the same threshold $t = F_p^{-1}(1 - c)$ as in Equation (2). Thus, the NE policy $A$ defined in Theorem 3.1 is a natural candidate for the limiting equilibrium. Note that this $p$ is only used for hypothesizing NE rather than demonstrating the convergence; see discussion in Section E.4. Moreover, the above argument suggests that, in the limit, the strategic environment becomes uniform across all agents, motivating us to develop an *infinite contest* (Definition E.5) and provide an alternative proof; see Section E.5.

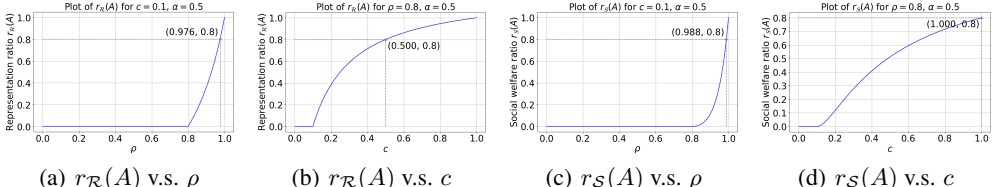

| (a) $r_\mathcal{R}(A)$ v.s. $\rho$ | (b) $r_\mathcal{R}(A)$ v.s. $c$ | (c) $r_\mathcal{S}(A)$ v.s. $\rho$ | (d) $r_\mathcal{S}(A)$ v.s. $c$ |

Figure 2: Plots of the representation ratio $r_\mathcal{R}(A)$ and the social welfare ratio $r_\mathcal{S}(A)$ as parameters $\rho$ and $c$ vary for Proposition 4.1, with default settings of $(\rho, c, \alpha) = (0.8, 0.1, 0.5)$. A dotted line in these plots indicates the threshold at which $r_\mathcal{R}(A) = 0.8$ or $r_\mathcal{S}(A) = 0.8$.

*Showing $A$ is an NE.* Although we have a solid guess for the NE policy $A$, a key challenge arises: the NE policy for finite $n$ lacks an explicit form, making it unclear how to define convergence to $A$. Towards this end, the third key idea is to provide a different proof of convergence to a threshold function for an undifferentiated case with density $p$, without using the explicit NE formulations. The first attempt to prove that $A$ is an $\varepsilon$-NE for sufficiently large $n$ fails—even in the simple case with $p = \mathrm{Unif}[0, 1]$ and $c = 0.5$. One can construct a valuation $v = 0.2$ such that the agent benefits by exerting a small effort $e = 0.01$, achieving a winning probability $P_i(e; A_{-i}) \approx 0.5$ and obtaining a payoff of approximately 0.09. To bypass this, the idea is to define a proxy sequence $A^{(n)}$ with threshold policies $s^{(n)}$ that 1) converge to $A$ as $n \to \infty$ and 2) ensure that $P_i(e; A^{(n)}_{-i}) \to 0$ for $e < t$, making $A^{(n)}$ an $\varepsilon_n$-NE with $\varepsilon_n \to 0$ (see Section E.1.3). To this end, we define $s^{(n)}(v) = t$ if $v \geq t - \sqrt{\log n / n}$ and 0 otherwise, so that a $(c + \sqrt{\log n / n})$-fraction of agents put in effort $t$. This yields $P_i(e; A^{(n)}_{-i}) \leq \sqrt{1/n}$ by concentration, ensuring the payoff from deviation is at most $\sqrt{1/n}$, and $A^{(n)}$ is an $O(\sqrt{\log n / n})$-NE. Finally, we adapt this new proof technique to the two-group case with general $p_1, p_2$ by carefully selecting the following threshold shift (Definitions E.1 and E.2):

$$\Delta_n := \min \left\{ F_1^{-1}(F_1(t) - \sqrt{\log n / n}), \; F_2^{-1}(F_2(t) - \sqrt{\log n / n}) \right\},$$

and set $s^{(n)}(v) = t$ if $v \geq t - \Delta_n$, 0 otherwise (see Theorem E.2). We find that $\lim_{n \to \infty} \Delta_n = t$, indicating that $s^{(n)}$ converges to $s$. Crucially, such a $\Delta_n$ ensures at least a $(c + \sqrt{\log n / n})$-fraction of agents, in expectation, putting in effort $t$. Using this property, we establish a concentration tail bound for the winning probability $P_i$ analogous to the undifferentiated contest: $P_i(e; A^{(n)}_{-i}) \to 0$ if $e < t$ (see Lemma E.4). Furthermore, this minimal winning probability guarantees that $A^{(n)}$ is an $\varepsilon_n$-NE with $\lim_{n \to \infty} \varepsilon_n = 0$ (Lemma E.6). This analysis concludes Theorem 3.1.

**Uniqueness of the NE guaranteed by Theorem 3.1.** First, if $\Omega_1 \cup \Omega_2$ is not connected, the CDF $(1 - \alpha)F_1(v) + \alpha F_2(v)$ may not be strictly monotonic over $\Omega_1 \cup \Omega_2$. For instance, if $\alpha = c = 0.5$, $p_1$ is uniform on $\Omega_1 = [0, 1]$ and $p_2$ is uniform on $\Omega_2 = [2, 3]$, then $(1 - \alpha)F_1(1) + \alpha F_2(1) = (1 - \alpha)F_1(2) + \alpha F_2(2) = 0.5 = 1 - c$. Consequently, there may exist two distinct points $t_1 = 1$ and $t_2 = 2$ in $\Omega_1 \cup \Omega_2$, leading to non-unique NEs. In contrast, when $\Omega_1 \cup \Omega_2$ is connected and each $p_\ell$ is positive on $\Omega_\ell$, the unique solution $t$ to Equation (3) ensures a unique NE policy. To see this, by Corollary 3.2 of [20], symmetric agents use symmetric policies in NE, so we hypothesize a policy pair $(s_1, s_2)$ for $G_1$ and $G_2$, respectively. As $n \to \infty$, both $s_1$ and $s_2$ manifest as threshold functions, leading to $s_1 = s_2 = s$, ensuring the uniqueness of NE. Additionally, it arises because if $s_1 \neq s_2$, the NE would destabilize, as agents from one group would adjust their thresholds to gain higher payoffs. This proof can be easily extended to multiple groups and non-identical cost of effort; see Remark E.7.

## 4 Analysis: metric behavior and intervention design

**Variation of metrics with $\rho$ and $c$.** Using Theorems 3.1 and 3.2, we analyze how the metrics—representation ratio $r_\mathcal{R}(A)$, social welfare ratio $r_\mathcal{S}(A)$, and average revenue $\mathcal{RV}(A, m)$—respond to changes in the bias parameter $\rho$ and the selectivity parameter $c$. We study whether these effects are linear or non-linear, and whether they exhibit sharp thresholds.

*Setup.* We adopt the setting from Section 3: $p_1$ is uniform on $[0, 1]$, $p_2$ is its $\rho$-biased variant, uniform on $[0, \rho]$, and $p_a$ is a point mass at 0. This isolates the effect of asymmetric valuations while simplifying calculations. The corresponding CDFs are $F_1(v) = v$ and $F_2(v) = v/\rho$, with both saturating to 1 outside their support. Unless varied explicitly, we use default values $\rho = 0.8$, $c = 0.1$,

and $\alpha = 0.5$, representing moderate bias, high selectivity, and balanced group sizes. We refer to Section D.2 for analogous analysis under truncated Gaussian distributions.

*Closed-form metrics.* Fixing the density setup above, we apply Theorems 3.1 and 3.2 to derive closed-form expressions for $t$, $r_\mathcal{R}(A)$, and $r_\mathcal{S}(A)$. These are summarized below (proof in Section F).

**Proposition 4.1** (**Metrics for uniform densities**). *Let $p_1$ be uniform on $[0, 1]$, $p_2$ uniform on $[0, \rho]$, and $p_a$ a point mass at 0. Let $A$ be the NE policy as $n \to \infty$. Let $\rho_c := 1 - \frac{c}{1-\alpha}$. Then:*

$$t = 1 - \frac{c}{1-\alpha} \quad \text{if } \rho < \rho_c, \quad t = \frac{\rho(1-c)}{\rho - \alpha\rho + \alpha} \quad \text{if } \rho \geq \rho_c,$$

$$r_\mathcal{R}(A) = 0 \quad \text{if } \rho < \rho_c, \quad r_\mathcal{R}(A) = \frac{\rho - \alpha\rho + \alpha + c - 1}{\alpha - \alpha\rho + c\rho} \quad \text{if } \rho \geq \rho_c,$$

$$r_\mathcal{S}(A) = 0 \quad \text{if } \rho < \rho_c, \quad r_\mathcal{S}(A) = \frac{\rho(\rho - \alpha\rho + \alpha + c - 1)^2}{(\alpha - \alpha\rho + c\rho)^2} \quad \text{if } \rho \geq \rho_c.$$

*Moreover, $\mathcal{RV}(A, m) = m(t)$ for any merit function $m(\cdot)$; $r_\mathcal{R}(A)$ and $r_\mathcal{S}(A)$ are monotonically increasing functions of parameters $\rho, c, \alpha$.*

As in Equation (5), Proposition 4.1 reveals a sharp threshold at $\rho = 1 - \frac{c}{1-\alpha}$. When $\rho < 1 - \frac{c}{1-\alpha}$, $t$ lies above the maximum valuation in $G_2$, implying that no agents from that group participate. Consequently, $r_\mathcal{R}(A)$ and $r_\mathcal{S}(A)$ are zero and independent of $\rho$. When $\rho$ crosses this threshold, these metrics become positive and increase monotonically with $\rho$, reaching 1 at $\rho = 1$, the symmetric case.

*Metric behavior.* Figure 2 plots the representation ratio $r_\mathcal{R}(A)$ and the social welfare ratio $r_\mathcal{S}(A)$ as functions of the bias parameter $\rho$ and selectivity parameter $c$. The corresponding threshold values $t$ are shown in Appendix Figure 13(b). Both $r_\mathcal{R}(A)$ and $r_\mathcal{S}(A)$ exhibit non-linear growth with increasing $\rho$ and $c$, and drop sharply—super-linearly—when these parameters decrease. For instance, with $c = 0.1$, $\alpha = 0.5$, and $\rho \leq 0.85$, we observe that $r_\mathcal{R}(A) \leq 0.2$, indicating notably low representation for group $G_2$. This highlights a key practical insight: in highly selective environments, such as contests with a 1-in-10 selection rate, strategic behavior amplifies disparities. These trends echo empirical findings on under-representation in competitive domains [77, 61]. Moreover, reductions in $c$ (i.e., increased selectivity) lead to pronounced declines in both representation and social welfare.

These trends of metrics offer designers of meritocratic selection processes critical insights into strategies for countering under-representation and elevating $r_\mathcal{R}(A)$ (or mitigating disparities in average payoffs and elevating $r_\mathcal{S}(A)$). We recall the two main criteria for identifying representation bias: 1) Ensuring the selection of at least one agent from every group, and 2) adhering to the 80% rule, which serves as a guideline for identifying potential adverse impact if the hiring rate for $G_2$ falls below 80% of that for $G_1$, i.e., $r_\mathcal{R}(A) \geq 0.8$. Given the fixed nature of $\alpha$ within the population structure, the main avenues for interventions aimed at improving $r_\mathcal{R}(A)$ focus on adjusting the parameters $\rho$ or $c$. Below, we explore potential interventions for both approaches:

(1) Increasing $\rho$ effectively means increasing the valuation of agents in $G_2$. Various strategies have been proposed and implemented to achieve this goal. For instance, [38] highlights several approaches to narrow the pay gap, including enhancing workplace flexibility, decreasing the cost associated with temporal flexibility, and improving the availability of high-quality, affordable childcare. These interventions aim to increase job valuation for women, analogous to increasing $\rho$. Figure 2(a) quantifies the required increase in $\rho$: to ensure at least one agent from $G_2$ is selected, $\rho$ must exceed 0.8; to adhere to the 80%-rule, it should be at least 0.976.

(2) As shown in Figure 2(b), raising $c$ above 0.1 satisfies the criterion for selecting at least one agent from $G_2$, while elevating it to 0.5 meets the 80%-rule requirement. Increasing $c$ represents a more straightforward approach than boosting $\rho$ and might be more feasible for institutions. This could involve pre-selecting a larger subset of candidates and applying a distinct selection process to this subset, based on institutional priorities and the likelihood of successful candidates following the expected trajectories.

Finally, regarding the average revenue $\mathcal{RV}(A, m) = m(t)$, it immediately follows from Proposition 4.1 that $\mathcal{RV}(A, m)$ significantly decreases as competition within the entire population intensifies—either through a decrease in $\rho$ or an increase in $c$. Given that average revenue is indicative of the benefit of the institute, this trend underscores the critical need for contest designers to mitigate systemic biases in valuations. The decline in average revenue with increasing bias compromises not

only the fairness of the contest, but also the overall quality of the outcomes it produces. This aligns with research that has explored the losses attributed to systemic biases [42, 55, 64].

**Optimizing interventions under cost and fairness constraints.** Having identified two policy levers—reducing bias ($\rho$) and increasing selectivity ($c$)—a natural question arises: how should institutions choose between these interventions to improve outcomes such as the representation ratio or social welfare ratio? To address this, we formulate a constrained optimization problem for cost-effective intervention design. We allow two interventions: increasing $\rho$ by $\Delta_\rho \in [0, 1 - \rho]$, and increasing $c$ by $\Delta_c \in [0, 1 - c]$. Let $r_\mathcal{R}(\Delta_\rho, \Delta_c)$ denote the representation ratio under the NE policy with updated parameters $\rho + \Delta_\rho, c + \Delta_c$. The goal is to ensure $r_\mathcal{R}(\Delta_\rho, \Delta_c) \geq \tau$ while minimizing intervention cost. We define two components of the cost function:

(1) *Resource cost of increasing $\rho$.* Let $f : [0, 1 - \rho] \to \mathbb{R}_{\geq 0}$ be monotonic, modeling the institutional cost of boosting valuation. A simple form is linear: $f(\Delta_\rho) = a\Delta_\rho$, justified by first-order Taylor approximation when $\Delta_\rho$ is small. Other variants include $f(\Delta_\rho) = a\Delta_\rho^\beta$ for $\beta > 1$, representing the increase in the marginal cost of continuously improving bias.

(2) *Cost via revenue loss.* Increasing $c$ reduces average revenue $\mathcal{RV}(A, m) = m(t)$, as it lowers the score threshold $t$. Let $g(\Delta_\rho, \Delta_c) = m(t(\rho, c)) - m(t(\rho + \Delta_\rho, c + \Delta_c))$, represent the revenue decline. Since the institution seeks to maximize value, this loss contributes to total intervention cost.

*Optimization problem.* We formalize the intervention design as:

$$\min_{\Delta_\rho \in [0, 1-\rho],\ \Delta_c \in [0, 1-c]} \quad f(\Delta_\rho) + g(\Delta_\rho, \Delta_c) \quad \text{s.t.} \qquad r_\mathcal{R}(\Delta_\rho, \Delta_c) \geq \tau. \tag{6}$$

This framework also applies to reducing welfare disparities by replacing $r_\mathcal{R}$ with $r_\mathcal{S}$ in the constraint.

*Empirical calibration.* To demonstrate real-world applicability, we calibrate the model using gender-disaggregated data from *JEE Advanced 2024*, a highly competitive entrance exam for India's IITs. Of 180,200 candidates, 139,180 were male and 41,020 female; 40,284 males and 7,964 females qualified. This yields admit rates of 28.9% for males and 19.4% for females, giving an observed representation ratio of $r_{\text{obs}} \approx 0.671$. The overall selection rate is $c \approx 0.268$, and the female applicant fraction is $\alpha \approx 0.228$. These values anchor our analysis of strategic disparities and potential interventions. In Section G.1, under the uniform density setup in Proposition 4.1, we compute $\rho \approx 0.882$ using the explicit form of $r_{\text{obs}} = r_\mathcal{R}(A) = 1 - \frac{(1-c)(1-\rho)}{\alpha - \alpha\rho + c\rho}$.

*Explicit solution under uniform densities.* We now solve the optimization problem under this uniform density setup with $(\rho, c, \alpha) = (0.882, 0.268, 0.228)$, setting $m(e) = e$, $f(\Delta_\rho) = 5\Delta_\rho^{1.1}$, and $g(\Delta_\rho, \Delta_c) = t(\rho, c) - t(\rho + \Delta_\rho, c + \Delta_c)$. This yields the objective: $5\Delta_\rho^{1.1} - [t(\rho + \Delta_\rho, c + \Delta_c) - t(\rho, c)]$, subject to the condition $r_\mathcal{R}(\Delta_\rho, \Delta_c) \geq \tau$ and feasibility constraints.

*Insights.* Figure 4 shows the optimal intervention as a function of the target threshold $\tau$. For $\tau \leq 0.92$, increasing $c$ (lowering selectivity) is more cost-effective. For $\tau > 0.92$, increasing $\rho$ (mitigating bias) becomes preferable. This suggests that expanding access is more impactful under high disparity, while improving group valuation is better when gaps are narrower.

We conducted additional simulations by varying $\alpha$ and $c$ beyond the default values (see Section G.2). The results remain consistent with our main findings, confirming the robustness of the above key insights. In Section G.3, we also offer a concrete example to illustrate how our model supports interpretable predictions and can inform data-grounded interventions—while also noting what is required to operationalize it in practice.

$$\min_{\Delta_\rho \in [0, 1-\rho], \Delta_c \in [0, 1-c]} \quad 5\Delta_\rho^{1.1} - \frac{(\rho + \Delta_\rho)(1 - c - \Delta_c)}{(1 - \alpha)(\rho + \Delta_\rho) + \alpha}$$

$$s.t. \quad \frac{(1 - \alpha)(\rho + \Delta_\rho) + \alpha + c + \Delta_c - 1}{\alpha - \alpha(\rho + \Delta_\rho) + (c + \Delta_c)(\rho + \Delta_\rho)} \geq \tau$$

$$\rho + \Delta_\rho \geq 1 - \frac{c + \Delta_c}{1 - \alpha}.$$

Figure 3: Explicit form of Problem (6).

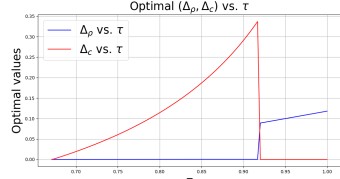

Figure 4: Plot of optimal interventions $(\Delta_\rho, \Delta_c)$ for various $\tau \in (0.671, 1]$.

**Alternative potential interventions.** Having discussed interventions based on adjusting $\rho$ and $c$, we next consider alternative approaches that modify the contest structure itself. These interventions can further reduce disparities in representation or social welfare ratios, though they depart from the baseline two-group contest formulation. A detailed analysis of these extensions is provided in Section G.4.

*Introducing preference heterogeneity.* One approach is to apply group-specific merit mappings of the form $m_\ell(s) = x_\ell s + y_\ell$ for group $G_\ell$ ($\ell = 1, 2$), with parameters $x_\ell, y_\ell \geq 0$. Here, $x_\ell$ acts as a scaling factor or "handicap," and $y_\ell$ as an offset or "head start" (see also Appendix A). With this intervention, we can still compute the Nash equilibrium for infinite $n$ (Theorem G.1), which implies:

- $r_{\mathrm{R}}(A)$ and $r_{\mathcal{S}}(A)$ increase with $\rho$, $c$, and $\alpha$, as well as with $x_2, y_2$, and decrease with $x_1, y_1$;
- Choosing merit parameters with $x_2 > x_1$ and $y_2 > y_1$ can sustain high representation and welfare ratios (e.g., $r_{\mathrm{R}}(A), r_{\mathcal{S}}(A) \geq 0.8$) even in highly selective settings;
- Increasing $x_2$ or $y_2$ can thus serve as an effective disparity-reducing intervention.

*Incorporating outside options.* Another possibility is to assign each agent in group $G_\ell$ a reservation payoff $\lambda_\ell \geq 0$ if not selected. Because this payoff is earned only upon losing, a higher $\lambda_\ell$ lowers the marginal benefit of effort, acting opposite to the merit parameters $x_\ell$ and $y_\ell$. Hence, increasing $\lambda_1$ (the outside option for the advantaged group) reduces their effort incentives and can help narrow representation and welfare gaps.

*Setting group-specific selection rates.* Finally, the institution can set separate capacity constraints for each group—for instance, selecting a $c$-fraction of agents from $G_1$ and $G_2$ independently. This decomposes the overall model into two within-group contests, fixing $r_{\mathcal{R}}(A) = 1$ under equal selection rates. Compared to the combined contest, agents in $G_2$ now face a lower bar for selection and exert more effort on average.

## 5 Conclusion, limitations, and future work

This work highlights a central tension in modern meritocratic systems: even when selection mechanisms are formally unbiased, systemic disparities in how groups perceive value can lead rational agents to behave in ways that perpetuate inequality. Our model captures this dynamic through a strategic contest framework that extends all-pay auctions to multi-group settings. By analyzing Nash equilibria in the large population limit, we characterize how group-level biases ($\rho$) and selectivity ($c$) affect fairness and institutional metrics such as representation, social welfare, and revenue. A central contribution is Theorem 3.1, which provides an explicit form for equilibrium strategies under broad conditions. Our framework enables interpretable predictions and supports data-grounded policy interventions.

Our model makes simplifying assumptions to enable analytical tractability. Most notably, it assumes agents are fully rational and that merit is captured by a single-dimensional notion of effort. In practice, decision-making is shaped by uncertainty, cultural context, and multifaceted criteria for merit. Extending the framework to incorporate bounded rationality, noisy information, or multidimensional effort remains an important direction for future work. Several application-driven extensions are also promising. One involves modeling university admissions systems with external incentives (e.g., brand-based free-riding), which may result in over-representation of certain groups ($\rho > 1$). Another is to study how affirmative action or group-dependent costs reshape equilibrium behavior. These variants would help bridge theory with institutional design. Beyond these extensions, an important avenue is to embed this static framework within dynamic feedback environments where perceptions evolve over time in response to outcomes and institutional signals. Such models could capture how bias propagates or attenuates across repeated selection cycles. Finally, while our model isolates a tractable facet of systemic inequality, real-world disparities—especially in AI-mediated evaluations—demand broader integration with social and historical context. As algorithmic tools shape hiring, admissions, and promotion, our framework helps explain how group-level differences in perceived value can interact with selection to amplify or mitigate bias. More broadly, we view this work as a step toward unifying rational-choice and structural perspectives on inequality through formal, data-driven modeling. We hope this work informs the design of more equitable, data-driven decision systems.

## Acknowledgments

LH acknowledges support from the State Key Laboratory of Novel Software Technology, the New Cornerstone Science Foundation, and NSFC Grant No. 625707396. LEC was supported in part by NSF Award IIS-2045951. NKV was supported in part by NSF Grant CCF-2112665 and by grants from Tata Sons Private Limited, Tata Consultancy Services Limited, and Titan.

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

# Contents

# A   Detailed related work

*Meritocratic selection process, pay gap, and statistical discrimination feedback loop.* In the social sciences, there is a large body of work that studies meritocratic selection processes and their limitations; see [56, 54, 73] and the references therein. [15, 59] discuss the pay gap in meritocratic systems, shedding light on how merit-based reward systems and gender wage gaps intersect. [38], in an extensive line of work, discusses the gender pay gap and addresses the economic and social factors contributing to wage disparities between men and women. Another line of research focuses on studying statistical discrimination feedback loops, which model how firms update their beliefs about group quality over time, reinforcing disparities [3, 66, 21, 23, 5, 46]. For instance, [3] emphasizes how the cost of individualized assessment incentivizes reliance on priors, which can become self-fulfilling and reinforce structural inequality. [3] models a profit-maximizing employer who faces noisy signals of productivity and rationally uses group-level statistics, leading to persistent wage gaps even with equal underlying abilities. [21] show that pessimistic beliefs about a group's productivity can result in tougher standards, reduced investment incentives, and discriminatory equilibria. [23] extend this to two-sided settings where firms and workers both act on noisy beliefs, reinforcing low-investment, low-opportunity equilibria. A key distinction, as we understand it, is that classical models of statistical discrimination typically generate disparities through *imperfect* and *group-dependent* beliefs about identical underlying abilities. In contrast, our framework allows *perfect, unbiased* information at the institutional level and identical selection criteria for all candidates. We focus instead on *valuation asymmetries*—that is, differences in the *perceived benefit* of success across groups—and show that these differences alone can lead to disparities in effort and representation, even under meritocratic selection.

*All-pay auctions and Tullock contests.* In game theory, there is a significant body of literature that investigates all-pay auctions. For instance, [76, 52, 82] study the setting in which every agent knows their private valuations and the distribution of other agents. Specifically, [76] study a "biased" 2-agent contest in which the designer is allowed to give a "headstart" to the effort of one agent. This headstart can be interpreted as differing merits of the agents, which corresponds to the initial abilities in our model. They characterize the optimal design for maximizing the expected highest effort or total effort of agents. In this case, the bias is introduced by the designer, rather than inherent in the system. [82] study the undifferentiated case for a single winner, while the contest designer is allowed to select the contest success function (CSF) based on agents' efforts. Their main focus is on studying the optimal design of the CSF that maximizes the total expected effort. The main difference from our model is that they consider bias in the efforts instead of in the valuations.

The all-pay auction with complete information has also been well-studied. Unlike the setting in this paper, these works assume that the valuations of all agents are known. [7] initiated the study of an $n$-agent $k$-winner all-pay auction and provided a complete characterization of the NE distribution. A line of research investigates the optimal design for maximizing the total expected effort/revenue, including imposing a multiplicative bias on the effort of agents [32, 36] or introducing an additional headstart [35, 33, 34, 36, 87].

Tullock contests [79, 78, 30, 34, 25, 50, 47] model the probability of winning based on relative effort without direct costs for participation, whereas an all-pay auction requires all agents to pay their bid amounts regardless of winning, with only the highest bidder(s) securing the prize. [62] study the dynamics of large contests, where a significant number of agents compete. Such contests pose unique analytical challenges and offer insights into the behavior of agents in mass competition scenarios. The works of [32, 33, 34] also investigate how the design of contests can be optimized to maximize revenue, considering factors like bias in efforts, headstarts, and the structure of the CSF. Across these studies, a common theme is the characterization of Nash equilibrium strategies within the context of different contest models, and identifying designs that encourage maximal effort or revenue.

*Strategic classification and ranking.* Another related direction is strategic learning, which mainly includes strategic classification [14, 43, 57, 9] and strategic ranking [29]. In strategic classification, agents can exert effort to alter their features to achieve higher values according to the published classifier. The designer's aim is to select a classifier that is robust to the manipulation of inputs by strategic agents. However, in this setting, agents' efforts are influenced solely by the published classifier, with no competition among them. In strategic ranking problems, agents' payoffs depend on their post-ranking, which is determined by a combination of their prior rankings and efforts. While

there is competition in this problem, all agents have the same valuation, which is different from our model.

*Models of bias in valuations.* Several works have modeled group-level biases based on empirical observations [4, 10, 45, 28, 18]. Additive and multiplicative skews in the valuations have also been modeled [45, 10]. [45] consider valuations $v > 0$ of the advantaged group distributed according to the uniform or Pareto density and, for the disadvantaged group, they model the output as $v/\beta$ for some fixed $\beta \geq 1$. We consider a class of bias models inspired by this model, the $\rho$ in our case corresponds exactly to $1/\beta$. The implicit variance model of [28] models differences in the amount of noise in the valuations for individuals in different groups. Here, the output estimate is drawn from a Gaussian density whose mean is the valuation $e$ (which can take any real value) and whose variance depends on the group of the individual being evaluated: The variance is higher for individuals in the disadvantaged group compared to individuals in the advantaged group. [18] proposes an optimization-based approach to model how group-wise valuation distributions can be obtained by tuning parameters such "information constraints" or "risk aversion".

## A.1 Comparison of the two-group contest with relevant models

To the best of our knowledge, our model is novel and has not been studied in the literature. Below, we compare our model with the most relevant models. Firstly, [1] examines a specific case of our model with $n = 2$, $k = 1$, and $\alpha = 0.5$, demonstrating the existence of a unique NE under certain conditions. While their analysis is limited to a two-player scenario, our model generalizes this by considering any number of players and allowing for multiple winners. [29] propose another two-group contest model. However, a key distinction in our model is the consideration of asymmetric valuation distributions across groups, whereas [29] introduces bias through the cost of effort, assuming symmetric valuations for all agents. This asymmetry in valuation distributions in our model adds complexity to the analysis.

Another related work is [27], which explores an all-pay auction with two groups. In their model, agents' abilities are symmetrically distributed, and those in the advantaged group may receive additional rewards with equivalent bids. Despite the symmetric strategic environment in [27], our model features asymmetric valuation distributions between groups, resulting in an asymmetric strategic environment. This asymmetry introduces further computational challenges for deriving the NE; details can be found below.

**A detailed comparison with [27].** We provide a detailed comparison between our two-group model and that in [27]. The primary distinction is that their model results in a symmetric strategic environment, while ours creates an asymmetric one. Below, we provide further details on this difference.

In the model of [27], each agent belongs to the target group with probability $\mu$ or to the non-target group with probability $1 - \mu$, independently of the other agents. The ability of an agent is then drawn i.i.d. from distribution $F$ if they belong to the target group, and from $G$ if they belong to the non-target group. As a result, the ability of each agent is drawn identically and independently from the joint distribution $\mu F + (1 - \mu)G$. Let $H$ be the CDF of this joint distribution. The probability that a given ability $v$ is among the top $k$ abilities is then given by $\sum_{i=n-k}^{n-1} \binom{n-1}{i} H(v)^i (1 - H(v))^{n-1-i}$, which is the same for each agent, thereby resulting in a symmetric strategic environment.

In our model, consider a simplified case where $p_a$ is a point mass at 0. Then, $F_1$ and $F_2$ correspond to the cumulative distribution functions (CDFs) of $p_1$ and $p_2$, which represent the valuation densities of $G_1$ and $G_2$, respectively. The probability that, for an agent in $G_1$, a given valuation $v$ is among the top $k$ valuations is given by:

$$\sum_{i=0}^{n-1} \sum_{j=n-k-i}^{n-1-i} \binom{n-1}{i} \binom{n-1-i}{j} F_1(v)^i (1 - F_1(v))^{(1-\alpha)n-1-i} F_2(v)^j (1 - F_2(v))^{\alpha n - j}.$$

In contrast, the probability for an agent in $G_2$ is given by

$$\sum_{i=0}^{n-1} \sum_{j=n-k-i}^{n-1-i} \binom{n-1}{i} \binom{n-1-i}{j} F_1(v)^i (1 - F_1(v))^{(1-\alpha)n-i} F_2(v)^j (1 - F_2(v))^{\alpha n - 1 - j}.$$

These two expressions differ whenever $p_1 \neq p_2$, leading to asymmetry in the strategic environment. This asymmetry significantly complicates the computation of the order statistics for the $(k-1)$-th

effort compared to the symmetric ones. E.g., for strategies $s_1$ and $s_2$, let $F_{s_\ell}(v)$ denote the CDF of efforts $s_\ell(v)$ when $v \sim p_\ell$. The cumulative distribution of the $(k-1)$-th effort $e^\star$ from an agent in $G_1$ is then given by:

$$\Pr[e^\star \leq v]$$
$$= \sum_{i=0}^{n-1} \sum_{j=(1-c)n-i}^{n-1-i} \binom{n-1}{i} \binom{n-1-i}{j} F_{s_1}(v)^i (1 - F_{s_1}(v))^{(1-\alpha)n-1-i} F_{s_2}(v)^j (1 - F_{s_2}(v))^{\alpha n - j}.$$

In contrast, the cumulative distribution of the $(k-1)$-th effort $e^\star$ from an agent in $G_2$ is given by:

$$\Pr[e^\star \leq v]$$
$$= \sum_{i=0}^{n-1} \sum_{j=(1-c)n-i}^{n-1-i} \binom{n-1}{i} \binom{n-1-i}{j} F_{s_1}(v)^i (1 - F_{s_1}(v))^{(1-\alpha)n-i} F_{s_2}(v)^j (1 - F_{s_2}(v))^{\alpha n - 1 - j}.$$

In the symmetric ones ($s_1 = s_2 = s$), the computation simplifies to:

$$\Pr[e^\star \leq v] = \sum_{i=(1-c)n}^{n-1} \binom{n-1}{i} F_s(v)^i (1 - F_s(v))^{n-1-i}.$$

Thus, the calculus and approximations for the two-group contest is significantly more difficult, making it harder to arrive at the equilibrium policies than in the contest with a symmetric strategic environment.

## B  Illustrative examples for the two-group case

In this section, we present a two-agent example with a biased valuation distribution to illustrate both the difficulty of computing the Nash equilibrium (NE) policy and the significant impact of valuation bias on the contest outcome. Let $c = 0.5$. Let the density $p_1$ of agent 1 be the uniform distribution on $\Omega_1 = [0, 1]$ and $p_2$ of agent 2 be the $\rho$-biased version of $p_1$ supported on $\Omega_2 = [0, \rho]$. Let the density $p_a$ be a point mass at 0. Let $A_\ell : \Omega_\ell \to \mathbb{R}_{\geq 0}$ be the NE policy that maps valuation $v_\ell$ to effort $A_\ell(v_\ell)$. We assume $A_\ell$ is monotonically increasing on the domain $\Omega_\ell$.

In this example, if agent 1 puts in effort $e$, it wins if the effort of agent 2 is smaller than $e$. Thus, its winning probability $P_1 = \frac{A_2^{-1}(e)}{\rho}$[4] and its payoff is $\pi_1(v, e; A_2) = \frac{A_2^{-1}(e)}{\rho} v - e$. Similarly, if agent 2 puts in effort $e$, its winning probability $P_2 = A_1^{-1}(e)$ and payoff is $\pi_2(v, e; A_1) = A_1^{-1}(e)v - e$. Then, by the stability condition (1), we have $\frac{\partial \pi_\ell(v, e; A_{3-\ell})}{\partial e} |_{e = A_\ell(v)} = 0$, implying that

$$A_2'(A_2^{-1}(A_1(v))) = \frac{v}{\rho}, \text{ and } A_1'(A_1^{-1}(A_2(v))) = v.$$

Solving this gives us the following explicit forms:

$$\forall v \in \Omega_1, \ A_1(v) = \frac{\rho}{\rho + 1} v^{\rho + 1}; \text{ and } \forall v \in \Omega_2, \ A_2(v) = \frac{\rho^{-1/\rho}}{\rho + 1} v^{1 + 1/\rho}. \tag{7}$$

Specifically, when $\rho = 1$ (the unbiased case), we have $A_1 = A_2 = A$, which simplifies the stability condition to $A'(v) = v$, yielding the NE policy $A(v) = \frac{v^2}{2}$. Also note that for $v \in \Omega_2$,

$$\frac{A_1(v)}{A_2(v)} = \rho^{1 + 1/\rho} v^{\rho - 1/\rho} \leq \rho^{1 + 1/\rho} \rho^{\rho - 1/\rho} = \rho^{1 + \rho} \leq 1,$$

which implies that $A_1(v) \leq A_2(v)$. Thus, agent 2 is more inclined to put in greater effort than agent 1 for identical valuations.

---

[4]Here, we assume $A_2^{-1}(e) = \rho$ if $A_2(\rho) < e$.

Imagine an institute that is unaware of the bias in valuations across two agents and thus applies the unbiased NE policy $A(v) = \frac{v^2}{2}$ to predict the contest outcome. For instance, it would predict the average revenue

$$\mathcal{RV}(A, m) = \int_0^1 \Pr_{v_\ell \sim p_\ell}\left[m\left(\max\left\{\frac{v_1^2}{2}, \frac{v_2^2}{2}\right\}\right) > t\right] dt = 0.25,$$

where the merit function is $m(t) = t$. However, under a $\rho$-biased valuation distribution, the true average revenue is $\mathcal{RV}_\rho = \frac{\rho}{2(\rho+1)}$, which decreases monotonically with $\rho$. This implies that the institute could overestimate its expected benefit $\mathcal{RV}$ by a fraction of

$$\frac{0.25 - \mathcal{RV}_\rho}{\mathcal{RV}_\rho} = \frac{1 - \rho}{2\rho},$$

which amounts to approximately 13% when $\rho = 0.8$. This example underscores the importance of studying asymmetric valuations and highlights the relevance of our proposed metrics for analyzing their impacts.

We also observe that even for this simple two-agent example, the stability condition is considerably more complex than in the undifferentiated case. In more general settings—such as those involving multiple spots, non-uniform valuation densities, or non-trivial ability densities—the explicit forms of NE policies for a two-group contest become even more complicated, making direct computation and explicit analysis impractical.

## C Analysis of finite NE policies in the uniform distribution case

In this section, we use the uniform distribution example from Section 3 as a running example, introduce a dynamic algorithm (Algorithm 1) to approximate the finite NE policies, and perform a statistical comparison between the finite and infinite cases. Additionally, we provide a theoretical analysis of the closeness between the NE policies and associated metrics in the finite and infinite cases.

### C.1 Empirical analysis

**Dynamics for computing finite NE policies.** Recall that we consider $p_1 = p = \text{Unif}[0, 1]$, $p_2 = p_2 = \text{Unif}[0, \rho]$ for $\rho \in [0, 1]$, and $p_a \equiv 0$. Algorithm 1 presents a dynamic procedure to approximately compute the finite-population NE policies.

We initialize each group's policy $s_\ell^{(0)}$ with a smoothness variant of the infinite NE policy (Lines 3–4), then iteratively update these policies over $N$ steps and return the final output as an approximation of the finite NE (Lines 5–38). At each iteration $t$:

1. We first update the effort set $E_t$ based on the policies $s_\ell^{(t-1)}$ from the previous iteration (Line 6). Since the action space is continuous, we restrict agents to choose efforts only from this finite set $E_t$.

2. Next, we update the policy for group $G_1$ using the policy $s_2^{(t-1)}$ from the previous iteration (Lines 7–21). The computation is performed over a finite set $V^{(1)}$ of discrete valuation levels (Line 1). For each valuation $v$, we determine the best-response effort that maximizes the agent's expected payoff by computing winning probabilities through a convolution of binomials (Lines 9–19). Specifically, we set $p_1 = 1 - v$ in Lines 9 and 12, consistent with the monotonicity constraint enforced in Line 11. Finally, Line 21 updates the policy using a carefully chosen step size $a_\ell^{(t)}$ to ensure convergence.

3. We then update the policy for group $G_2$ based on the policy $s_1^{(t-1)}$ from the previous iteration (Lines 7–21). This process mirrors that of $G_1$, with the main difference lying in the computation of winning probabilities for each effort in $E_t$ due to the asymmetric valuation distributions.

The resulting policies $s_\ell = s_\ell^{(T)}$ are defined on discrete valuation grids. To obtain continuous policies, we interpolate them by connecting adjacent valuation points with straight lines, resulting in piecewise-linear approximations.

**Choice of hyperparameters.** In our simulations, we set the valuation resolution $m_v = 101$, effort resolution $m_e = 101$, total number of iterations $T = 500$, and step sizes $a_1^{(t)} = a_2^{(t)} = \frac{1}{10T}$. We always set $n_1 = n_2 = \frac{n}{2}$, which means $\alpha = 0.5$.

**Metrics.** For each iteration $t$, we compute the following metric to evaluate the updated policies $s_\ell^{(t)}$:

$$\Delta^{(\ell,t)} := \frac{\sum_{v \in V^{(\ell)}} |\pi_\ell^{(t)}(v) - s_\ell^{(t-1)}(v)|}{m_v} = \frac{\sum_{v \in V^{(\ell)}} |s_\ell^{(t)}(v) - s_\ell^{(t-1)}(v)|}{a_\ell^{(t)} m_v},$$

which quantifies the average policy update for group $G_\ell$ at iteration $t$. Intuitively, a decreasing $\Delta^{(\ell,t)}$ indicates convergence of the policy sequence $s_\ell^{(t)}$. However, since we work with discretized valuation and effort sets, we do not expect $\Delta^{(\ell,t)}$ to vanish entirely.

**Results.** Figures 5, 6, 7, and 8 present the evolution of equilibrium policies for population sizes $n \in \{20, 200, 600, 1200\}$ across four time snapshots $t \in \{50, 150, 300, 500\}$, with fixed parameters $\rho = 0.8$ and $c = 0.2$. Although all runs begin with a smoothed version of the infinite-population NE, the dynamics vary significantly with population size. For small $n$ (e.g., $n = 20$), we observe noticeable fluctuations in early iterations, particularly in group $G_2$, whose valuation distribution is more concentrated. By $t = 500$, both policies stabilize, though they retain visible irregularities due to stochasticity in rank-based feedback. Even though all runs begin with a smooth initialization based on the infinite-population NE, the dynamics unfold differently depending on population size. For small $n$ (e.g., $n = 20$), we observe noticeable fluctuations in the early iterations, especially in group $G_2$, whose valuation distribution is more concentrated. At $t = 500$, the policies stabilize but retain some irregularity, reflecting noise in the agent-level ranking and feedback structure.

As $n$ increases, both groups' policies become smoother and stabilize more quickly. By $n = 600$, the effort policies align closely with the infinite NE, and further updates beyond $t = 300$ are negligible. These trends are confirmed by the convergence plots in Figure 9, which show a sharp reduction in the $\ell_1$-norm policy update $\Delta^{(\ell,t)}$ with increasing $n$. Group $G_1$ consistently converges faster than $G_2$, a pattern attributable to its broader valuation support and greater flexibility in effort choice. Overall, the results illustrate that the infinite-population equilibrium is a good predictor even for moderately sized finite systems, while also quantifying the transient effects and instability that emerge in low-$n$ regimes.

Interestingly, we also observe from these plots that when $n$ is small ($n = 20, 200$), $s_2(v) > s_1(v)$, while for larger values of $n$ ($n = 600, 1200$), $s_2(v) < s_1(v)$. In the subsequent subsection, we will provide a theoretical analysis to explain the underlying reasons for this behavior.

### C.2 Theoretical analysis

We begin by presenting theoretical evidence for the alignment between finite and infinite NEs, a relationship that is observed empirically. In the proof of Theorem 3.1 (see Section E), we show that for any finite $n$, $\varepsilon_n$-NE policy $s^{(n)}$ stated in Theorem 3.1 is "$O(\sqrt{\log n/n})$-close" to the policy $s$ for infinite $n$ and is "$O(\sqrt{\log n/n})$-close" to an NE policy. Specifically, when $p_1, p_2$ are uniform distributions and $p_a$ is a point mass at 0, for any constant $\alpha$, the closeness between $s^{(n)}$ and $s$ can be directly translated into the policy form: $s^{(n)} = 0$ for $v < t - O(\sqrt{\log n/n})$ and $s^{(n)} = t$ for $v \geq t - O(\sqrt{\log n/n})$. For general $p_1, p_2, p_a$, we note that the $O(\sqrt{\log n/n})$-closeness depends on the concept of densities, which is more complex. Corollary 3.2 from [20] implies that the NE policy must be symmetric within each group. Let $s_1$ represent the policy for $G_1$ and $s_2$ for $G_2$. The above analysis indicates that the closeness between $s_1$, $s_2$, and $s$ (from Equation (3)) is expected to be bounded by $O(\sqrt{\log n/n})$.

In the following, we analyze the empirical observations regarding the scaling of $s_1$ and $s_2$. Intuitively, $s_1(v)$ increases from approximately 0 to approximately $t$ as $v$ increases from $t - \sqrt{\log n/n}$ to $t + \sqrt{\log n/n}$. If an agent in $G_2$ exerts an effort of $t(1 - \sqrt{\log n/n})$, the agent's winning probability could exceed 95%. This observation motivates the choice of setting $s_2(v) = t(1 - \sqrt{\log n/n})$, rather than 0, to generate positive profits when $v > t(1 - \sqrt{\log n/n})/95\%$. As a result, if $t(1 - $

$\sqrt{\log n/n}/95\% \leq t$, i.e., $\sqrt{\frac{\log n}{n}} \geq 0.05$, then $s_2(t) \geq t(1 - \sqrt{\log n/n}) \geq s_1(v)$. Therefore, when $n$ is small and $\sqrt{\frac{\log n}{n}} \geq 0.05$, it holds that $s_2(v) > s_1(v)$ for $v \in \Omega_2$. Conversely, when $n$ is large, $\sqrt{\frac{\log n}{n}}$ becomes small, and $s_1(v) > s_2(v)$, as agents in $G_2$ consistently receive lower payoffs than those in $G_1$ when $s_1 = s_2$. This behavior explains the observed scaling between $s_1$ and $s_2$, as discussed in Section C.1.

Finally, we discuss the impact on metrics when the number $n$ of agents is finite. Recall that in the finite $n$ case, we can assume NE policies $s_1$ and $s_2$ for group $G_1$ and $G_2$, respectively. As discussed above, when $n$ is not too small, we have 1) $s_1 > s_2$ and 2) $s_1$, $s_2$, and $s$ are $O(\sqrt{\log n/n})$-close. Since $s_1 > s_2$ and they converge to the same policy as $n$ grows, the representation ratio $\mathcal{R}_1(A) = \mathbb{E}\left(\frac{|S \cap G_1|}{|G_1|}\right)$ decreases with $n$, while $\mathcal{R}_2(A) = \mathbb{E}\left(\frac{|S \cap G_2|}{|G_2|}\right)$ increases with $n$, where $S$ is the (random) winning set. Consequently, the representation ratio $r_\mathcal{R}(A) = \frac{\mathcal{R}_2(A)}{\mathcal{R}_1(A)}$ increases as $n$ grows. Since the gap between $s_1$ and $s_2$ is bounded by $O(\sqrt{\log n/n})$, this results in an increase of $O(\sqrt{\log n/n})$ in $\mathcal{R}_2(A)$ and a similar decrease of $O(\sqrt{\log n/n})$ in $\mathcal{R}_1(A)$ compared to the infinite case. Thus, the increase in $r_\mathcal{R}(A)$ should be bounded by $O(\sqrt{\log n/n})$. A similar quantitative analysis applies to the social welfare ratio $r_\mathcal{S}(A)$ and average revenue $\mathcal{RV}(A, m)$.

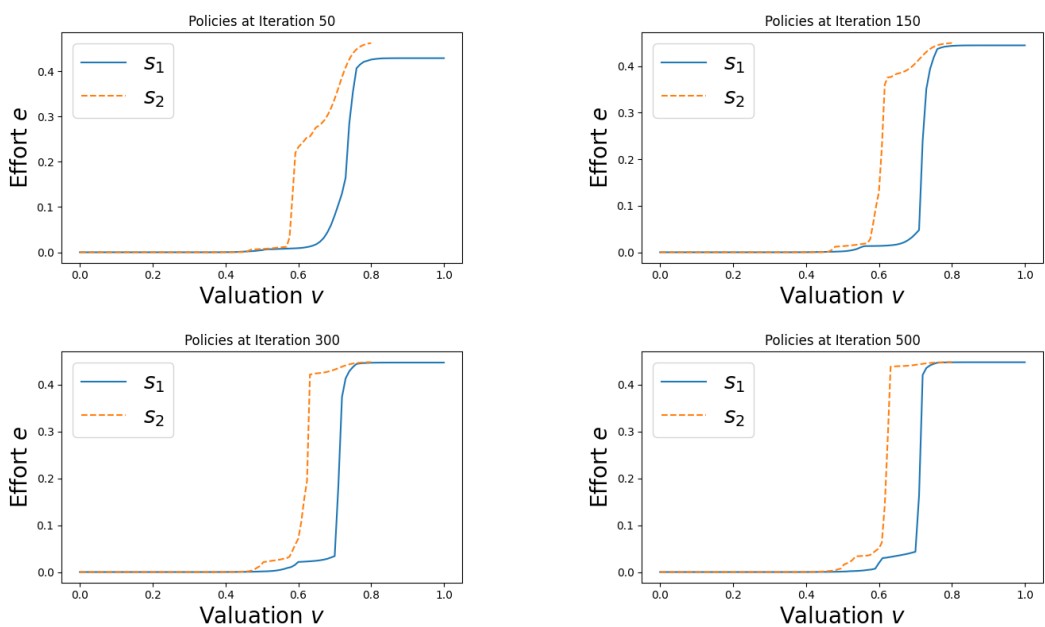

Figure 5: Evolution of group effort policies over time for $n = 20$, $\rho = 0.8$, and $c = 0.2$.

# D  Other bias models and analysis of metrics for their Nash equilibrium

## D.1  Other bias models

A natural extension of $p_1 = \text{Unif}[0, 1]$ in Section 2 is when $p_1$ is the density of the uniform distribution on an interval $[a, b]$ ($0 < a < b \leq \infty$) and $p_2$ is the density of the uniform distribution on $\Omega_2 = [\rho a, \rho b]$. Then $p_2(v) = \frac{1}{\rho(b-a)}$ and again, $\mathbb{E}_{v \sim p_2}[v] = \rho \cdot \frac{a+b}{2} = \rho \cdot \mathbb{E}_{v \sim p_1}[v]$. More generally, one might consider a density $p_1$ that is supported on a domain $\Omega_1 = [a, \infty]$, along with a $\rho$-biased density defined as $p_2(v) = \frac{1}{\rho} p_1(\frac{v}{\rho})$ for $\rho \in (0, 1]$ and $v \in [\rho a, \infty]$.

Besides the uniform distribution case, we consider valuations coming from a truncated normal distribution supported on $[0, 1]$. Formally, let $p_1$ be the density of a truncated normal distribution $N(\mu, \sigma^2)$ on the interval $\Omega_1 = [0, 1]$, where $\mu$ lies within $(0, 1)$ and $\sigma > 0$. Let $p_2$ be the density of a

**Algorithm 1** Dynamics for Computing Finite NE Policies for the Uniform Distribution Case

**Input:** Group sizes $n_1, n_2 \geq 1$, selection rate $c \in (0,1)$, bias $\rho \in [0,1]$, valuation steps $m_v \geq 1$,
effort steps $m_e \geq 1$, an interger $T \geq 1$, and two sequences of step sizes $\{a_\ell^{(t)}\}_{t\in[T],\ell\in\{1,2\}}$

**Output:** Policies $s_1, s_2$ for $G_1$ and $G_2$, respectively

1: Initialize grids $V^{(1)} \leftarrow \text{linspace}(0, 1, m_v)$, $V^{(2)} \leftarrow \text{linspace}(0, \rho, m_v)$, and $E = \text{linspace}(0, 1, m_e)$.

2: Compute $\alpha \leftarrow \frac{n_2}{n_1+n_2}$ and $k \leftarrow \lfloor c(n_1 + n_2) \rfloor$

3: Compute threshold $t = \begin{cases} 1 - \dfrac{c}{1-\alpha}, & \rho < 1 - \dfrac{c}{1-\alpha}, \\ \dfrac{\rho(1-c)}{\rho - \rho\alpha + \alpha}, & \text{otherwise,} \end{cases}$

4: Initialize $s_1^{(0)}(v) = s_2^{(0)}(v) = \frac{t}{1+e^{-50(v-\theta)}}$      ▷ Smoothness of the infinite NE in Proposition 4.1

5: **for** $t = 1$ **to** $T$ **do**

6:     $E_t \leftarrow E \cup \left\{ s_\ell^{(t-1)}(v) \mid v \in V^{(\ell)}, \ell \in \{1,2\} \right\}$                    ▷ Updating effort set

7:     $last_e \leftarrow 0$

8:     **for** $i = 1$ **to** $m_v$ **do**

9:         $v \leftarrow V_i^{(1)}, best_e \leftarrow last_e, p_1 \leftarrow 1 - v$ and $p_2 \leftarrow \max_{v' \in V^{(2)}: s_2^{(t-1)}(v') \geq best_e} \frac{\rho - v'}{\rho}$

10:         $p^{(1)}(best_e) \leftarrow \sum_{a=0}^{k-1} \sum_{b=0}^{a} \left( \binom{n_1-1}{b} p_1^b (1-p_1)^{n_1-1-b} \right) \cdot \left( \binom{n_2}{a-b} p_2^{a-b}(1-p_1)^{n_2-a+b} \right)$

11:         **for all** $e \in E_t$ with $e \geq last_e$ **do**

12:             $p_1 \leftarrow 1 - v$ and $p_2 \leftarrow \max_{v' \in V^{(2)}: s_2^{(t-1)}(v') \geq best_e} \frac{\rho - v'}{\rho}$

13:             $p^{(1)}(e) \leftarrow \sum_{a=0}^{k-1} \sum_{b=0}^{a} \left( \binom{n_1-1}{b} p_1^b (1-p_1)^{n_1-1-b} \right) \cdot \left( \binom{n_2}{a-b} p_2^{a-b}(1-p_1)^{n_2-a+b} \right)$

14:             $pay \leftarrow p^{(1)}(e)\,v - e$

15:             **if** $pay > p^{(1)}(best_e)\,v - best_e$ **then**

16:                 $best_e \leftarrow e$

17:             **end if**

18:         **end for**

19:         $\pi_1^{(t)}(v) \leftarrow best_e, \ last_e \leftarrow best_e$

20:     **end for**

21:     $s_1^{(t)} \leftarrow s_1^{(t-1)} + a_1^{(t)}(\pi_1^{(t)} - s_1^{(t-1)})$

                                                    ▷ Update policy for $G_1$ at iteration $t$

22:     $last_e \leftarrow 0$

23:     **for** $i = 1$ **to** $m_v$ **do**

24:         $v \leftarrow V_i^{(2)}, best_e \leftarrow last_e, p_1 \leftarrow \max_{v' \in V^{(1)}: s_1^{(t)}(v') \geq best_e} 1 - v$ and $p_2 \leftarrow \frac{\rho - v'}{\rho}$

25:         $p^{(2)}(best_e) \leftarrow \sum_{a=0}^{k-1} \sum_{b=0}^{a} \left( \binom{n_1}{b} p_1^b (1-p_1)^{n_1-b} \right) \cdot \left( \binom{n_2-1}{a-b} p_2^{a-b}(1-p_1)^{n_2-1-a+b} \right)$

26:         **for all** $e \in E_t$ with $e \geq last_e$ **do**

27:             $p_1 \leftarrow \max_{v' \in V^{(1)}: s_1^{(t)}(v') \geq best_e} 1 - v$ and $p_2 \leftarrow \frac{\rho - v'}{\rho}$

28:             $p^{(2)}(e) \leftarrow \sum_{a=0}^{k-1} \sum_{b=0}^{a} \left( \binom{n_1}{b} p_1^b (1-p_1)^{n_1-b} \right) \cdot \left( \binom{n_2-1}{a-b} p_2^{a-b}(1-p_1)^{n_2-1-a+b} \right)$

29:             $pay \leftarrow p^{(2)}(e)\,v - e$

30:             **if** $pay > p^{(2)}(best_e)\,v - best_e$ **then**

31:                 $best_e \leftarrow e$

32:             **end if**

33:         **end for**

34:         $\pi_2^{(t)}(v) \leftarrow best_e, \ last_e \leftarrow best_e$

35:     **end for**

36:     $s_2^{(t)} \leftarrow s_2^{(t-1)} + a_2^{(t)}(\pi_2^{(t)} - s_2^{(t-1)})$

                                                    ▷ Update policy for $G_2$ at iteration $t$

37: **end for**

38: **return** Policies $s_1 \leftarrow s_1^{(T)}, \ s_2 \leftarrow s_2^{(T)}$

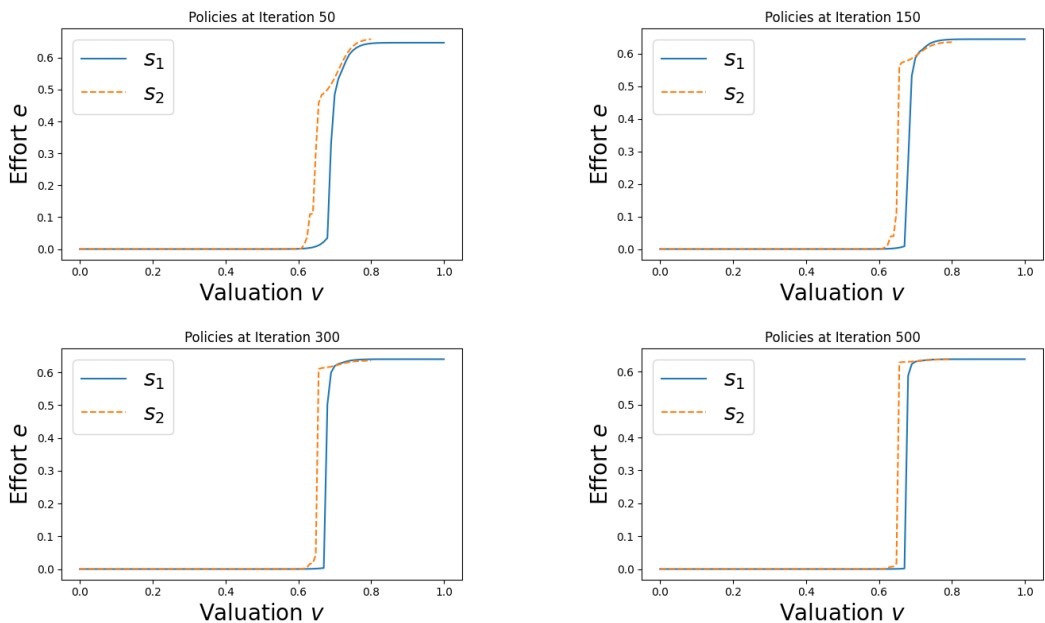

Figure 6: Evolution of group effort policies over time for $n = 200$, $\rho = 0.8$, and $c = 0.2$.

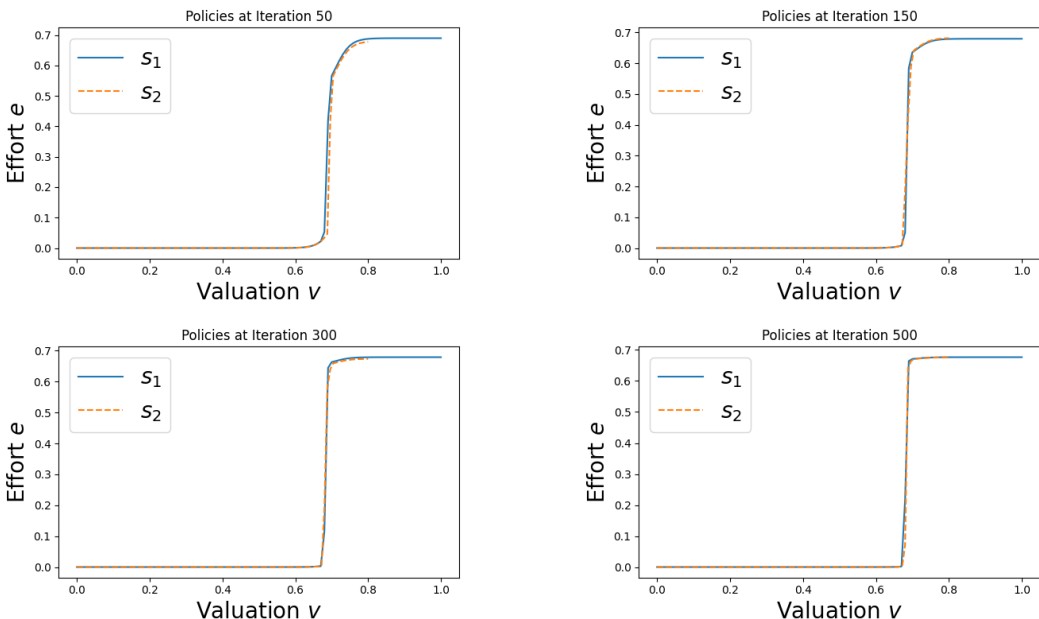

Figure 7: Evolution of group effort policies over time for $n = 600$, $\rho = 0.8$, and $c = 0.2$.

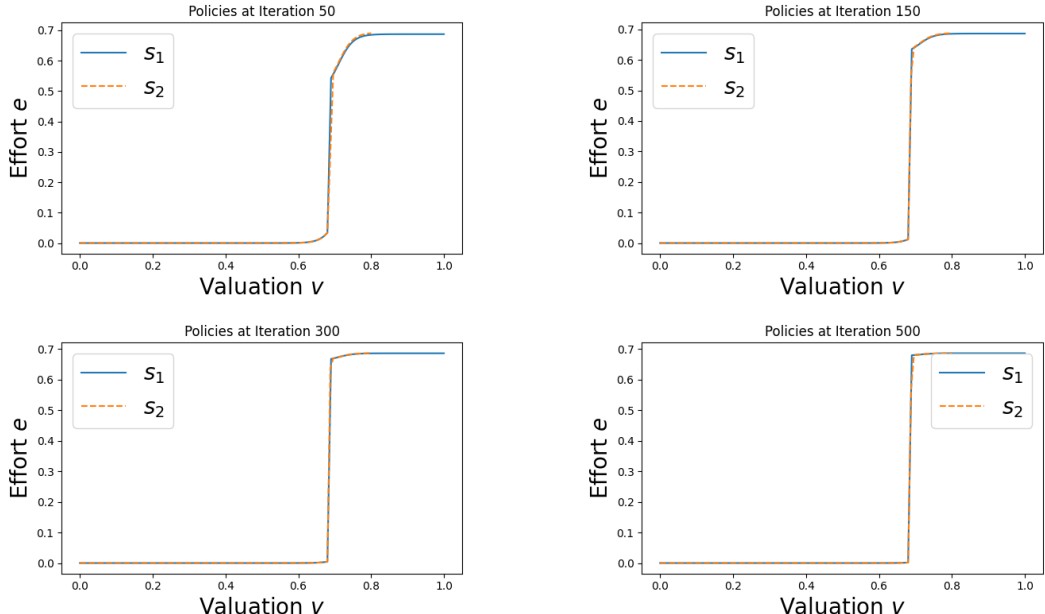

Figure 8: Evolution of group effort policies over time for $n = 1200$, $\rho = 0.8$, and $c = 0.2$.

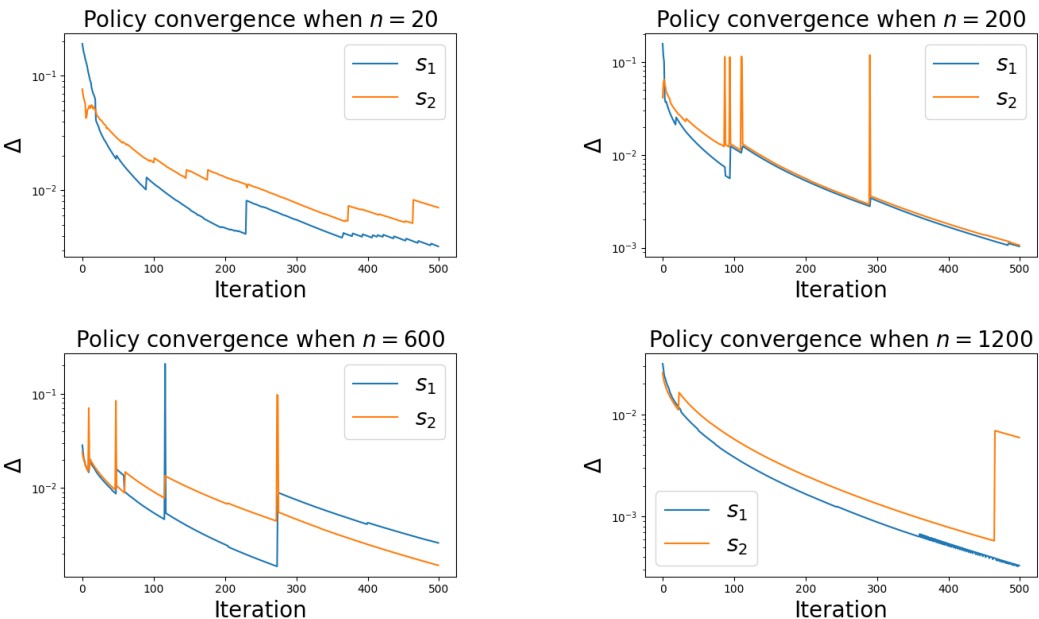

Figure 9: Convergence of group-wise policy updates $\Delta^{(\ell,t)}$ for different population sizes $n$, with fixed parameters $\rho = 0.8$ and $c = 0.2$.

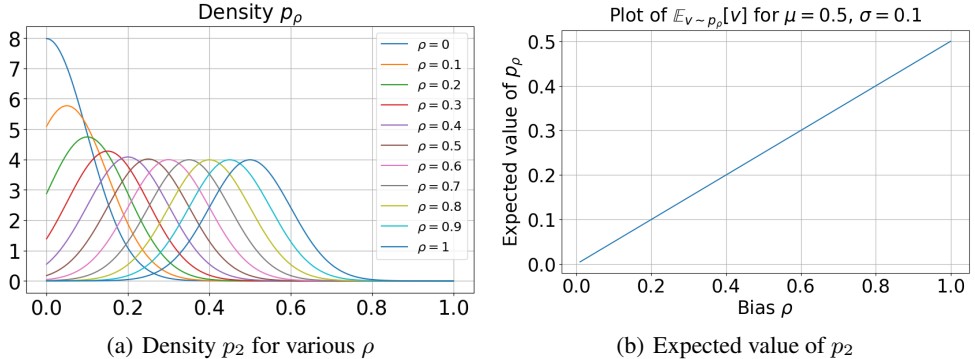

(a) Density $p_2$ for various $\rho$    (b) Expected value of $p_2$

Figure 10: Statistics for truncated normal distribution with $\mu = 0.5$, $\sigma = 0.1$.

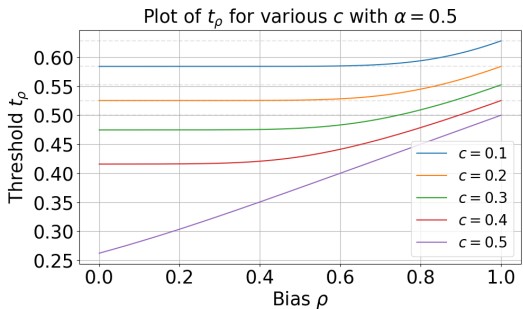

Figure 11: Plots of $t$ versus $\rho$ for various $\alpha$ with $c = 0.1$ for the truncated normal distribution. The dotted line $t_1 = 0.9$ corresponds to the undifferentiated contest with density $p = p_1$.

truncated normal distribution $N(\rho\mu, \sigma^2)$ on the interval $\Omega_2 = [0, 1]$. Since the bias is multiplicative, the domain of $p_1$, $\Omega_1 = [0, 1]$, does not influence the assessment of the contest's results. Note that $\mathbb{E}_{v \sim p_2}[v] = \rho\mu + \frac{\phi(\frac{-\rho\mu}{\sigma}) - \phi(\frac{1-\rho\mu}{\sigma})}{\Phi(\frac{1-\rho\mu}{\sigma}) - \Phi(\frac{-\rho\mu}{\sigma})}$, where $\phi(x)$ is the probability density function of the standard normal distribution $N(0, 1)$ and $\Phi(x)$ is its cumulative distribution function. The expectation of $p_2$ does not decrease linearly with $\rho$ as in the uniform case, but it closely approximates a linear function and monotonically decreases with $\rho$. This is motivated by real-world settings where the valuations (such as pay or SAT scores) exhibit a truncated normal distribution [85]. Other variants of distributions include piecewise-linear, polynomial (such as Pareto), and log-normal distributions, along with their biased versions.

We implicitly assume that the bias parameter $\rho$ is fixed and identical for all agents in $G_2$ above. However, $\rho$ could be noisy and non-identical to agents. For instance, let $p_\rho$ be a density supported on $[0, 1]$. We assume each agent $i \in G_2$ has an individual bias $\rho_i$ i.i.d. drawn from $p_\rho$, and its valuation is drawn from the $\rho_i$-biased density of $p_1$. Then $p_2$ is supported on $\Omega_2 = \Omega_1$, and satisfies that for any $v \in \Omega_1$,

$$p_2(v) = \int_0^1 \frac{1}{x} p_\rho(x) p_1(\frac{v}{x}) dx.$$

### D.2 Analysis of metrics for Nash equilibrium in the truncated normal distribution case

In this section, we do a similar analysis as in Section 4 for the case that $p_1$ is a truncated normal distribution $N(\mu, \sigma^2)$ supported on $[0, 1]$, $p_2$ is a $\rho$-biased truncated normal distribution $N(\rho\mu, \sigma^2)$ supported on $[0, 1]$, and $p_a$ is a point mass at 0. We choose $\mu = 0.5$ and $\sigma = 0.1$. This selection ensures that the density function is narrowly focused around the mean and the expected value of $p_2$ is approximately $\rho\mu$; see Figure 10 for illustration. Note that $t$ analogues to Proposition 4.1 is the

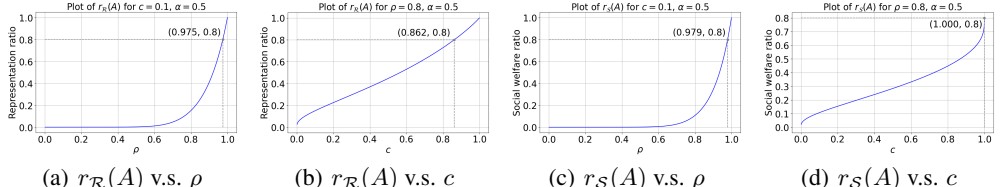

| (a) $r_{\mathcal{R}}(A)$ v.s. $\rho$ | (b) $r_{\mathcal{R}}(A)$ v.s. $c$ | (c) $r_{\mathcal{S}}(A)$ v.s. $\rho$ | (d) $r_{\mathcal{S}}(A)$ v.s. $c$ |

Figure 12: Plots illustrating the group-wise social welfare $\mathcal{S}_1(A), \mathcal{S}_2(A)$, and the social welfare ratio $r_{\mathcal{S}}(A)$ as functions of the parameters $\rho$, $c$, and $\alpha$ for the truncated normal distribution. By default, we set $(\rho, c, \alpha) = (0.8, 0.1, 0.5)$. A dotted line within these plots indicates the threshold at which $r_{\mathcal{S}}(A) = 0.8$.

solution of the following equation:

$$(1 - \alpha) \cdot \frac{\phi(\frac{v-\mu}{\sigma}) - \phi(\frac{1-\mu}{\sigma})}{\Phi(\frac{1-\mu}{\sigma}) - \Phi(\frac{-\mu}{\sigma})} + \alpha \cdot \frac{\phi(\frac{v-\rho\mu}{\sigma}) - \phi(\frac{1-\rho\mu}{\sigma})}{\Phi(\frac{1-\rho\mu}{\sigma}) - \Phi(\frac{-\rho\mu}{\sigma})} = 1 - c. \tag{8}$$

Unlike the uniform distribution case, it is hard to derive closed-form expressions for metrics on the outcomes of the contest. However, one can do numerical computations and we plot solution $t$, representation ratio $r_{\mathcal{R}}(A)$ and group-wise social welfare $\mathcal{S}_\ell(A)$ together with social welfare ratio $r_{\mathcal{S}}(A)$ in Figures 11 and 12 respectively. All plots exhibit a monotonic behavior similar to that observed with the uniform distribution. Next, we highlight some distinctions with the uniform distribution.

**No inflection point.** A notable feature of the truncated normal distribution is its lack of an inflection point. This trait is observed not only for $t$, but also in the behaviors of $r_{\mathcal{R}}(A)$ and $r_{\mathcal{S}}(A)$. This difference arises because the domain $\Omega_2 = [0, 1]$ remains consistent across all values of $\rho$.

**Representation ratio.** Figure 12 shows that to achieve a representation ratio $r_{\mathcal{R}}(A) \geq 0.8$, it is necessary to adjust $\rho$ to a minimum of 0.979 or increase $c$ to at least 0.862. The need to elevate $c$ is more pronounced than the required 0.5 observed with the uniform distribution in Figure 2(b). This difference arises because the truncated normal distribution tends to be more focused around its mean, leading to a higher number of agents in $G_1$ possessing valuations greater than the expected value $\approx 0.4$ of $p_2$.

# E Proof of Theorem 3.1: two-group contest

In this section, we begin by providing a more detailed technical overview of the proof of Theorem 3.1 (Section E.1). Next, we present a more comprehensive version of Theorem 3.1, including the explicit form of $A^{(n)}$ (Theorem E.2 in Section E.2). Finally, we provide the proof of Theorem E.2 (Section E.3).

## E.1 Technical overview

We present an overview of the proof of Theorem 3.1, which characterizes an NE policy for the two-group contest. Recall that, there are $n$ agents belonging to one of the two disjoint groups $G_1, G_2$ with $|G_1| = (1 - \alpha)n$ and $|G_2| = \alpha n$. The valuations of agents in $G_1$ come from the density $p_1$ supported on $\Omega_1$ and those of $G_2$ come from the density $p_2$ supported on $\Omega_2$. Each agent has an initial ability drawn from the density $p_a$. The selectivity of the contest is a constant $0 < c < 1$. Theorem 3.1 asserts that, as $n \to \infty$, there is an NE policy for the agents which is determined by a threshold $t \in (\Omega_1 \cup \Omega_2) + \Omega_a$ when the $(\Omega_1 \cup \Omega_2) + \Omega_a$ is a connected subset of $\mathbb{R}_{\geq 0}$.

For ease of analysis, we first consider the simple case where $p_a$ is a point mass at 0, so that agents' policies only depend on their valuations. We then show that the extension to a general $p_a$ is straightforward. We start by first quickly showing how to compute the NE policy in the special case when $p_1 = p_2$ for finite $n$ and why this approach does not extend to the two-group setting of interest (Section E.1.1). In Section E.1.2 we show that even though there are major challenges in extending the one-group case, it leads us to the right form of the NE policy for the two-group case as $n \to \infty$: a single threshold function that defines the strategies of agents in both $G_1$ and $G_2$. This also explains how we arrive at Equation (2) that characterizes the threshold $t$. Finally, in Section E.1.3, we present the approach to formally argue about and prove the convergence of the finite $n$ two-group contest to

this pair of NE policies. This analysis also reveals why the conjectured policy is a Nash equilibrium. With all the background, we conclude Theorem 3.1.

### E.1.1 NE policy for the undifferentiated contest for finite $n$ and obstacle in extending it

Here, we consider the special case when $p_1 = p_2 = p$ is the density of the uniform distribution on $[0, 1]$, and $p_a$ is a point mass at 0. The argument for other densities is similar. First note that by symmetry among agents, it is reasonable to assume that, at equilibrium, each agent will follow the same policy $s$. Moreover, since the domain of $p_1$ is nonnegative reals, it is reasonable to assume that the NE policy is monotone in an agent's valuation. The following calculations show that both symmetry and monotonicity hold.

Recall that an agent $i$ with valuation $v$ is selected if the effort $s(v)$ is among the top $c$ fraction of efforts. Thus, assuming that all agents follow $s$, agent $i$ is selected if and only if there are at least $(1-c)n$ distinct agents $j$ with $s(v_j) < s(v)$. (We ignore the issue that there may be ties for this discussion.) Since $s$ is a monotone function, $s(v_j) < s(v)$ holds if and only if $v_j < v$. Thus, the probability of selection of this agent is

$$P_i(s(v); A_{-i}) = \sum_{j=(1-c)n}^{n-1} \binom{n-1}{j} v^j (1-v)^{n-1-j}.$$

Hence, its expected payoff is $\pi_i(v, s(v); A_{-i}) = P_i(s(v); A_{-i}) \cdot v - s(v)$. (See Section 2 for notation). A key observation is that the calculation of $P_i(s(v); A_{-i})$ only depends on density $p_1$ and $v$, and is independent of the choice of $s$, under the monotonicity and symmetry assumptions on $s$. If $s$ is to be an NE, then it must satisfy that for any other effort $e$, $\pi_i(v, s(v); A_{-i}) \geq \pi_i(v, e; A_{-i})$. This follows from the condition that the derivative with respect to $v$, $\pi_i'(v, s(v); A_{-i}) = P_i'(s(v); A_{-i}) \cdot v - s'(v) = 0$. As noted above, $P_i(s(v); A_{-i})$ does not depend on $s$, so we get a simple differential equation involving the derivative of $s'(v) = \left( \sum_{j=(1-c)n}^{n-1} \binom{n-1}{j} v^j (1-v)^{n-1-j} \right)' \cdot v$. Thus,

$$s(v) = (1-c) \cdot \sum_{j=(1-c)n+1}^{n} \binom{n}{j} v^j (1-v)^{n-j} \tag{9}$$

is the unique NE for the undifferentiated contest and it that this $s$ is monotone. For a general density $p$, we can apply a similar analysis to obtain that

$$s(v) := Q_p(v) \cdot v - \int_{\underline{\Omega}}^{v} Q_p(x) \, dx, \tag{10}$$

where $Q_p(v) = \sum_{i=n-k}^{n-1} \binom{n-1}{i} \cdot F_p(v)^i \cdot (1 - F_p(v))^{n-i-1}$ for any $v \in \Omega$. We also note that the computation of $s$ can become significantly more complicated for a general density $p_a$, as it requires considering the stability condition for two partial derivatives: $\frac{\partial s(v,a)}{\partial v}$ and $\frac{\partial s(v,a)}{\partial a}$.

We now attempt to extend the analysis above to the two-group contest. Since each agent in each group uses the same valuation density, we can still hope for a symmetric and monotone NE policy for each group; say $s_1$ for $G_1$ and $s_2$ for $G_2$. However, we can no longer assume that $s_1 = s_2$. There are simple examples (see Section B) for which it can be shown that $s_1 \neq s_2$. This considerably complicates the calculation of probability $P_i$ for the $i$th agent getting selected since the order of agents' valuations may differ from that of agents' efforts. For instance, it is now possible that for agent $i \in G_1$ and agent $j \in G_2$, $v_i < v_j$ but $s_1(v_i) > s_2(v_j)$. Thus, $P_i$ must depend on functions $s_1$ and $s_2$, instead of only depending on density $p = p_1$ and $v$ as in the undifferentiated case. Thus, it is no longer possible to write a simple differential equation as in the undifferentiated case.

### E.1.2 A conjectured NE policy in two-group contests for large $n$

Since it seems intractable to find an NE policy for the two-group case, we study whether the situation becomes easier when $n$ is large. This hope is rooted in the observation that for the undifferentiated contest when $n$ is large, the NE policy $s(v)$ defined in Equation (9) converges to a threshold function. To see this, recall that for the uniform distribution, $s(v) = (1-c) \cdot \sum_{j=(1-c)n+1}^{n} \binom{n}{j} v^j (1-v)^{n-j}$. Since $s(v)$ is the probability associated with a sum of i.i.d. random variables, it follows from the

Chernoff bound that as $n \to \infty$, $s(v) \to 1 - c$ for any $v > 1 - c$ and $s(v) \to 0$ for any $v < 1 - c$. This argument is not specific to the uniform distribution and extends to any density $p_1$ with NE policy defined in Equation (10). In particular, if $F_1$ denotes the CDF of $p_1$, the limiting NE is given by $s(v) = F_1^{-1}(1-c)$ if $v \geq F_1^{-1}(1-c)$ and $s(v) = 0$ otherwise. The threshold $F_1^{-1}(1-c)$ guarantees that the expected fraction of agents that put in a nonzero effort is $1 - F_1(F_1^{-1}(1-c)) = c$. The rationale for $s(v) = F_1^{-1}(1-c)$ when $v$ is above the threshold is twofold: 1) agents would not exert effort beyond their valuation, ensuring $s(F_1^{-1}(1-c)) \leq F_1^{-1}(1-c)$, and 2) agents with valuations below $F_1^{-1}(1-c)$ are disincentivized from participating, leading to $s(F_1^{-1}(1-c)) \geq F_1^{-1}(1-c)$.

Thus, one may hope that in a two-group contest, as $n \to \infty$, the NE policy might similarly converge to two threshold functions $s_1$ and $s_2$, each with a corresponding threshold $t_\ell$. While this assumption allows us to give an explicit form for the probability $P_i$ of agent $i$ getting selected, this expression is quite complicated and, importantly, depends on $t_1$ and $t_2$. Thus, we are unable to obtain conditions that determine $t_1$ and $t_2$ from the NE condition.

Going back to the setting when $p_1$ is the density of the uniform distribution over $[0,1]$ and $p_2 = p_2$ is the density of the uniform distribution over $[0, \rho]$, first observe that as $\rho \to 0$, $t_2 \to 0$. Thus, one would expect $t_1$ to be more than $t_2$. We now argue that counterintuitive to the above observation, $t_1 > t_2$ cannot lead to an NE. To see this, first observe that when $t_1 > t_2$, if an agent puts in effort $t_1$, then it will get selected. Thus, the probability of an agent in $G_1$ getting selected is $1 - F_1(t_1)$. Hence, if $1 - F_1(t_1) < \frac{c}{1-\alpha}$, fewer than $cn$ agents in $G_1$ get selected. Thus, agents in $G_1$ getting selected will find that putting in effort slightly larger than $t_2$ instead of $t_1$ suffices to ensure their effort is larger than all agents in $G_2$, and consequently, they will still be selected. Through this reduction in effort, they can gain an additional payoff of $t_1 - t_2$, which violates the stability condition. A similar argument holds for $G_2$ when $1 - F_1(t_1) > \frac{c}{1-\alpha}$. Thus, $t_1 > t_2$ leads to instability. Similarly, we can argue that $t_1 < t_2$ also leads to instability. Thus, in case the NE policies for $G_1$ and $G_2$ are thresholds, it must be the case that $t_1 = t_2$ when $n$ is large. However, it is not clear how this can hold given that the domains of $p_1$ and $p_2$ are different.

To explore this, we consider a scenario when $\alpha = 0.5$, $p_1$ is the density of the uniform distribution over $[0,1]$ and $p_2$ is the density of the uniform distribution over $[0, 0.5]$ ($\rho = 0.5$). With high probability, there would be more than $0.1n$ agents from $G_1$ whose valuation is larger than $0.5$. Hence, if we set $c = 0.1$, no agent from $G_2$ will have any incentive to put in an effort while for agents in $G_1$ a threshold of $t_1 = 0.8$ suffices. The key observation is that even though the two policies are different, the policy of $G_2$ is just 0. This suggests that both policies can be seen as restrictions of the same threshold function to their respective domains.

Now we show how to compute the threshold $t$. The idea is to reduce the two-group contest to an undifferentiated one whose density $p = (1 - \alpha)p_1 + \alpha p_2$, supported on the domain $\Omega_1 \cup \Omega_2$. In this undifferentiated contest, as $n \to \infty$, it is likely that $(1 - \alpha)$-fraction of agents with valuation come from $p_1$ and $\alpha$-fraction of agents come from $p_2$. This suggests that these two contests are increasingly indistinguishable as $n$ grows, leading to the same limiting threshold $t = F_p^{-1}(1-c)$. Thus, threshold $t$ is the solution of the equation $(1 - \alpha)F_1(v) + \alpha F_2(v) = 1 - c$ – the one denoted in Equation (2). This argument can be extended to general $p_a$, resulting in the following lemma.

**Lemma E.1** (Unique solution). *Let $\alpha, c \in (0,1)$, $p_1$ be a density supported on the domain $\Omega_1 \subseteq \mathbb{R}_{\geq 0}$, $p_2$ be a density supported on the domain $\Omega_2 \subseteq \mathbb{R}_{\geq 0}$, and $p_a$ is a density supported on the domain $\Omega_a \subseteq \mathbb{R}_{\geq 0}$. If $(\Omega_1 \cup \Omega_2) + \Omega_a$ is connected and each density $p_1, p_2, p_a$ is positive at any point of its domain, then there exists a unique solution $t \in \Omega_1 \cup \Omega_2$ for the following equation: $(1 - \alpha)F_1(\zeta) + \alpha F_2(\zeta) = 1 - c$, where for any $\zeta \in \mathbb{R}_{\geq 0}$, $F_\ell(\zeta) = \Pr_{v \sim p_\ell, a \sim p_a}[v + a \leq \zeta]$.*

The assumption is naturally met in cases such as the uniform distribution and the truncated normal distribution discussed in Section 2. Since $F_\ell$ is a CDF, it must be strictly monotonic across its domain $\Omega_\ell + \Omega_a$. For any $\zeta, \zeta' \in (\Omega_1 \cup \Omega_2) + \Omega_a$ with $\zeta < \zeta'$, if $(1-\alpha)F_1(\zeta) + \alpha F_2(\zeta) = (1-\alpha)F_1(\zeta') + \alpha F_2(\zeta')$, we must have both $F_\ell(\zeta) = F_\ell(\zeta')$ holds. Then $(\zeta, \zeta') \cap ((\Omega_1 \cup \Omega_2) + \Omega) = \emptyset$, which contradicts the connected domain assumption. Hence, $(1 - \alpha)F_1(\zeta) + \alpha F_2(\zeta)$ is strictly monotonic across the domain $(\Omega_1 \cup \Omega_2) + \Omega_a$, which ensures the uniqueness of solution $t$. The proof can be found in Section E.3.1.

### E.1.3 Proving convergence to the conjectured NE policy

Now we outline how to prove that, as $n \to \infty$, the NE policy for the two-group contest converges to the threshold policy corresponding to $t$ as guaranteed by Lemma E.1. To do so, first, we have to make it precise what convergence means. Towards this, we revisit the undifferentiated contest. While we argued that in this case, as $n \to \infty$, the NE policy tends to a threshold function, recall that we used the explicit form of the NE policy for finite $n$. Unfortunately, since we do not have an explicit form for the two-group contest (for finite $n$), we need a strategy for the undifferentiated case that works without the knowledge of the explicit NE for finite $n$.

Let $A$ be the NE policy of the undifferentiated contest as $n \to \infty$, characterized by a function $s$ which is a threshold with parameter $t$. It suffices to prove that for policy $A$ and every $\varepsilon > 0$ there is an $n_\varepsilon$ such that for $n \geq n_\varepsilon$, $A$ is an $\varepsilon$-NE. However, we find that there exists an $\varepsilon > 0$ such that for any $n \geq 1$, $A$ is not an $\varepsilon$-NE policy. We revisit the simple example of uniform distribution discussed above, in which $c = 0.5$, $p_1$ is the density of the uniform distribution on $[0, 1]$, and $p_a$ is a point mass at 0. Recall that the threshold $t = 1 - c = 0.5$. Then, in expectation, $0.5n$ agents have valuations at least 0.5 and put in effort 0.5. As $n \to \infty$, it follows from symmetry that the probability that fewer than $0.5n$ agents with valuation $\geq t$ approaches approximately 0.5. Thus, an agent $i$ with a valuation $v = 0.2$ and putting in an effort $e = 0.01$ would have about a 0.5 probability of being selected, i.e., $P_i(e; A_{-i}) \approx 0.5$. Since $s(v) = 0$, the probability $P_i(A_i(v); A_{-i}) = 0$. Thus, we have

$$\pi_i(v, e; A_{-i}) - \pi_i(v, A_i(v); A_{-i}) = P_i(e; A_{-i}) \cdot v - e - 0 \approx 0.2 * 0.5 - 0.01 \gg 0.$$

This inequality implies that when $\varepsilon = 0.08$, for any $n \geq 1$, $A$ is not an $\varepsilon$-NE policy.

To bypass this, we consider a sequence of proxies for $A$, denoted by $A^{(n)}$, and characterized by threshold functions $s^{(n)}$. These proxies aim to ensure that the winning probability $P_i(e; A_{-i}) \approx 0$ and hence, serve as an approximate NE policy (see Definition 3.1). Therefore, we need $A^{(n)}$ to satisfy two conditions:

1. $A^{(n)}$ converges to $A$ as $n$ approaches infinity, i.e., $\lim_{n \to \infty} s^{(n)} = s$, and

2. The winning probability under $A^{(n)}$ approaches zero in the limit, i.e., $\lim_{n \to \infty} P_i(e; A^{(n)}_{-i}) \to 0$.

Ensuring $P_i(e; A^{(n)}_{-i}) \approx 0$ essentially involves guaranteeing that the probability of having fewer than $cn$ agents with valuation $\geq t$ is negligible. Specifically, for the uniform case, by adjusting the threshold by $\sqrt{\log n/n} = o(1)$, we define the policy $s^{(n)}(v)$ as follows: $s^{(n)}(v) = t$ if $v \geq t - \sqrt{\log n/n}$ and $s^{(n)}(v) = 0$ otherwise. By concentration, this $s^{(n)}$ ensures that the probability $P_i(e; A^{(n)}_{-i})$ is bounded above by $\sqrt{1/n}$ (Lemma E.4). This bounded probability ensures $A^{(n)}$ to be an $\sqrt{1/n}$-NE policy (Lemma E.6). This concludes the proof that the policy $A$ is an NE in the large $n$ limit for the uniform distribution case for one group.

Finally, we adapt this new proof technique of constructing $A^{(n)}$ to the two-group case with general densities $p_1$ and $p_2$. Mirroring the strategy employed in the uniform distribution case, we would like to shift the threshold of $s^{(n)}$ to ensure that, in expectation, a $(c + \sqrt{\log n/n})$-fraction of agents put in effort $t$. To satisfy this, we define the following threshold $\Delta_n$ (Definitions E.1 and E.2) for the policy $A^{(n)}$:

$$\Delta_n := \min \left\{ F_1^{-1}(F_1(t) - \sqrt{\log n/n}), F_2^{-1}(F_2(t) - \sqrt{\log n/n}) \right\}.$$

Consequently, we define the policy $s^{(n)}$ as follows: $s^{(n)}(v) = t$ if $v \geq t - \Delta_n$ and $s^{(n)}(v) = 0$ otherwise (see Theorem E.2). We find that $\lim_{n \to \infty} \Delta_n = t$, indicating that $s^{(n)}$ converges to $s$. Crucially, such a $\Delta_n$ ensures that $(1 - \alpha)F_1(\Delta_n) + \alpha F_2(\Delta_n) \leq 1 - (c + \sqrt{\log n/n})$, thereby maintaining at least a $(c + \sqrt{\log n/n})$-fraction of agents, in expectation, putting in effort $t$. Using this property, we establish a bound for the winning probability $P_i$ analogous to the undifferentiated contest: $P_i(e; A^{(n)}_{-i}) \leq n^{-(1-\alpha)} + n^{-\alpha}$ if $e < t$ (see Lemma E.4). The factors $n^{-(1-\alpha)}$ and $n^{-\alpha}$ derive from concentration bounds applicable to group $G_1$ and $G_2$, respectively. Furthermore, this minimal winning probability guarantees that $A^{(n)}$ is an $\varepsilon_n$-NE with $\lim_{n \to \infty} \varepsilon_n = 0$ (Lemma E.6). The extension to the general $p_a$ is straightforward. The main difference is that the initial ability $a$

may already surpass the threshold $t$, in which case the agent does not need to exert any effort to be selected. This is characterized by the amount of effort $s^{(n)}(v, a) = \max\{t - a, 0\}$ if $v \geq t - \Delta_n$.

To summarize, we first arrived at a policy $A$ that is characterized by a function $s$ which is parameterized by a threshold $t$ defined by Lemma E.1 and then we constructed a sequence of "proxies" $A^{(n)}$ that converge to $A$ as $n \to \infty$. Moreover, we construct a sequence $\varepsilon_1, \ldots, \varepsilon_n, \ldots$ with limit 0 such that $A^{(n)}$ is an $\varepsilon_n$-NE policy for every $n$. This implies that $A$ is an NE policy as $n \to \infty$. Thus, the overview above allows us to prove Theorem 3.1.

## E.2 A more comprehensive version of Theorem 3.1: convergence form

Now we show how to construct a series of policies $\{A^{(n)}\}_n$ that approach $A$, the NE policy from Equation (3) in Theorem 3.1, as $n \to \infty$. The most technical part will be to prove that $A^{(n)}$ acts as an $\varepsilon_n$-NE policy, where $\varepsilon_n \to 0$ with increasing $n$.

Suppose $(\Omega_1 \cup \Omega_2) + \Omega_a$ is connected and let $t \in (\Omega_1 \cup \Omega_2) + \Omega_a$ be a unique solution of the equation $(1 - \alpha)F_1(\zeta) + \alpha F_2(\zeta) = 1 - c$ (the uniqueness of $t$ is ensured by Lemma E.1). Since $c \in (0, 1)$, we have that either $F_1(t) > 0$ or $F_2(t) > 0$. Accordingly, we define a threshold $n_t$ as follows.

**Definition E.1 (Threshold $n_t$).** *Given a value $t \in \Omega_1 \cup \Omega_2$, we define a threshold $n_t$ as follows:*

- *If both $F_1(t) > 0$ and $F_2(t) > 0$, let $n_t$ be the smallest integer such that $F_\ell(t) - \sqrt{\frac{\log n_t}{n_t}} > 0$ for $\ell = 1, 2$.*

- *If $F_1(t) > 0$ and $F_2(t) = 0$, let $n_t$ be the smallest integer such that $F_1(t) - \sqrt{\frac{\log n_t}{n_t}} > 0$.*

- *If $F_1(t) = 0$ and $F_2(t) > 0$, let $n_t$ be the smallest integer such that $F_2(t) - \sqrt{\frac{\log n_t}{n_t}} > 0$.*

Note that such $n_t$ is finite and always exists since $\sqrt{\frac{\log n}{n}} \to 0$ as $n \to \infty$. Also note that $n_t$ is monotonically decreasing to $t$ across the domain $\Omega_1 \cup \Omega_2$. This value of $n_t$ is useful for defining $\Delta_n$, which is essential for the construction of policy $A^{(n)}$.

**Definition E.2 (Threshold $\Delta_n$).** *Let $n \geq n_t$ be an integer. We define a threshold $\Delta_n$ as follows:*

- *If both $F_1(t) > 0$ and $F_2(t) > 0$, let [5]*

$$\Delta_n := \min\left\{F_1^{-1}(F_1(t) - \sqrt{\frac{\log n}{n}}), F_2^{-1}(F_2(t) - \sqrt{\frac{\log n}{n}})\right\}.$$

- *If $F_1(t) > 0$ and $F_2(t) = 0$, let $\Delta_n := F_1^{-1}(F_1(t) - \sqrt{\frac{\log n}{n}})$.*

- *Otherwise if $F_1(t) = 0$ and $F_2(t) > 0$, let $\Delta_n := F_2^{-1}(F_2(t) - \sqrt{\frac{\log n}{n}})$.*

The requirement that $n \geq n_t$ ensures the proper definition of $\Delta_n$. As the number of agents $n$ grows indefinitely, the term $\sqrt{\frac{\log n}{n}}$ approaches 0, leading $\Delta_n$ to converge towards $t$. The threshold function $s^{(n)}$ defined as in Equation (11), designed as a threshold function, incorporates $\Delta_n$ as its threshold. The convergence of $\Delta_n$ to $t$ as $n \to \infty$ is crucial for ensuring that $A^{(n)}$ gradually aligns with the policy $A$ over large populations.

We are ready to provide the formal statement of Theorem 3.1.

**Theorem E.2 (Two-group contest: Large $n$ limit).** *Let $\alpha, c \in (0, 1)$. For $\ell = 1, 2$, let $p_\ell$ be a density supported on a domain $\Omega_\ell \subseteq \mathbb{R}_{\geq 0}$. Let $m : \mathbb{R}_{\geq 0} \to \mathbb{R}_{\geq 0}$ be a merit function that is strictly increasing. Suppose $\Omega_1 \cup \Omega_2$ is connected and each density $p_\ell$ is positive at any point of domain*

---

[5]If the inverse function $F_\ell^{-1}(F_\ell(t) - \sqrt{\frac{\log n}{n}})$ yields multiple values, it is defined to be the maximum of these values.

$\Omega_\ell$. *Let $t \in \Omega_1 \cup \Omega_2$ be a unique solution of the equation $(1 - \alpha)F_1(v) + \alpha F_2(v) = 1 - c$ (the uniqueness of $t$ is ensured by Lemma E.1). Let $n_t$ be defined as in Definition E.1. Let $n \geq n_t$ be an integer and let $\Delta_n$ be defined as in Definition E.2. Define*

$$s^{(n)}(v, a) := \begin{cases} 0 & \text{if } v < \Delta_n \\ \max\{t - a, 0\} & \text{if } v \geq \Delta_n \end{cases} \tag{11}$$

*gives rise to a policy for the two-group contest: Under this policy, agent $i \in G_1$ uses the restriction $A_i^{(n)} = s^{(n)}|_{\Omega_1}$, while each agent $j \in G_2$ uses the restriction $A_j^{(n)} = s^{(n)}|_{\Omega_2}$. We have $\lim_{n \to \infty} s^{(n)} = s$, where $s$ is the threshold function defined as in Equation (3). Moreover, the sequence of policies $A^{(n_t)}, A^{(n_t+1)}, \dots$ satisfies the following property:*

$$\forall \varepsilon > 0, \ \exists n_\varepsilon \geq n_t, \ \text{s.t. } \forall n \geq n_\varepsilon, \ A^{(n)} \text{ is an } \varepsilon\text{-NE policy}. \tag{12}$$

This theorem establishes how an NE policy for the two-group contest approaches a limit as the number of agents, $n$, grows indefinitely. It reveals that the sequence of policies $\{A^{(n)}\}_n$ not only converges to $A$ but also aligns with an NE policy for the two-group contest. Thus, it validates the assertion made in Theorem 3.1 that $A$ serves as an NE policy for the two-group contest in the limit as $n \to \infty$.

### E.3 Proof of Theorem E.2

We provide an overview of the proof, summarized as follows.

1. In Section E.3.1, we prove Lemma E.1 for the uniqueness of solution $t$ that decides the threshold function $s$.

2. In Section E.3.2, we bound the winning probabilities $P_i(e; A_{-i}^{(n)})$ under policy $A^{(n)}$; summarized by Lemma E.4. Its proof relies on the winning probability for the undifferentiated contest (Lemma E.5), whose computation is via an auxiliary function defined in Definition E.3.

3. In Section E.3.3, we apply Lemma E.4 to prove that $A^{(n)}$ is approximate NE (Lemma E.6).

4. Finally in Section E.3.4, we show that Theorem E.2 is a corollary of Lemma E.6.

For simplicity, we first assume that $p_a$ is a point mass at 0, such that policies depend solely on valuations. In this case, $s(v, a), P_i(e; a, A_{-i}), \pi(v, a, e; A_{-i})$ are simplified to $s(v), P_i(e; A_{-i}), \pi(v, e; A_{-i})$ respectively. At the end, we will show how to extend this to a general $p_a$.

#### E.3.1 Proof of Lemma E.1: solution uniqueness

Instead of proving Lemma E.1, we directly prove for the general multi-group case. Let $G_1, \dots, G_m$ be $m \geq 2$ groups where each $G_\ell$ has size $n_\ell = \alpha_\ell n$ and valuation distribution $p_\ell$ on the domain $\Omega_\ell \subseteq \mathbb{R}_{\geq 0}$. We have $\alpha_\ell \in (0, 1)$ for every $\ell \in [m]$ and $\sum_{\ell \in [m]} \alpha_\ell = 1$. We have the following lemma that generalizes Lemma E.1.

**Lemma E.3** (**Unique solution for multiple groups**). *Suppose $(\cup_{\ell \in [m]} \Omega_\ell) + \Omega_a$ is connected and each density $p_\ell$ and $p_a$ is positive at any point of its domain. There exists a unique solution $t \in \cup_{\ell \in [m]} \Omega_\ell$ for the equation $\sum_{\ell \in [m]} \alpha_\ell F_\ell(\zeta) = 1 - c$, where for any $\zeta \in \mathbb{R}_{\geq 0}$, $F_\ell(\zeta) = \Pr_{v \sim p_\ell, a \sim p_a}[v + a \leq \zeta]$.*

**Proof:** Fix $\ell \in [m]$. Recall that we expand the domain of every CDF $F_\ell$ to $\mathbb{R}_{\geq 0}$. We have the following properties for $F_\ell$:

1. $F_\ell(\cdot)$ is non-decreasing across the domain $\mathbb{R}_{\geq 0}$, i.e., for any $\zeta, \zeta' \in \mathbb{R}_{\geq 0}$ with $\zeta < \zeta'$, $F_\ell(\zeta) \leq F_\ell(\zeta')$ holds.

2. $F_\ell(\cdot)$ is strictly monotonous across the domain $\Omega_\ell + \Omega_a$, i.e., for any $\zeta, \zeta' \in \Omega_\ell + \Omega_a$ with $\zeta < \zeta'$ and $(\zeta, \zeta') \cap (\Omega_\ell + \Omega_a) \neq \emptyset$, we have $F_\ell(v) < F_\ell(v')$.

Define a function $g : \mathbb{R}_{\geq 0} \to \mathbb{R}_{\geq 0}$ such that for any $\zeta \in \mathbb{R}_{\geq 0}$, $g(\zeta) = \sum_{\ell \in [m]} \alpha_\ell F_\ell(\zeta)$. Since $g(\cdot)$ is a convex combination of $F_\ell(\cdot)$'s, we know that $g(\cdot)$ is also non-decreasing across the domain $\mathbb{R}_{\geq 0}$. Moreover, since $(\bigcup_{\ell \in [m]} \Omega_\ell) + \Omega_a$ is connected, for any $\zeta, \zeta' \in (\bigcup_{\ell \in [m]} \Omega_\ell) + \Omega_a$ with $\zeta < \zeta'$, there must exist at least one $\ell \in [m]$ such that $F_\ell(\zeta) < F_\ell(\zeta')$ and $(\zeta, \zeta') \cap (\Omega_\ell + \Omega_a)$. This implies that $g(\cdot)$ is strictly monotonous across the domain $(\bigcup_{\ell \in [m]} \Omega_\ell) + \Omega_a$.

Now let $L$ and $U$ denote the infimum and the supremum of domain $(\bigcup_{\ell \in [m]} \Omega_\ell) + \Omega_a$ respectively. We have $0 = g(L) < 1 - c < g(U) = 1$. Thus, there must exist a unique point $t \in (\bigcup_{\ell \in [m]} \Omega_\ell) + \Omega_a$ such that $g(t) = 1 - c$. This completes the proof. $\square$

### E.3.2 Bounding winning probability

We first have the following lemma that bounds the winning probability under policy $A^{(n)}$.

**Lemma E.4 (Bounding winning probability).** *For every integer $n \geq n_t$, we have*

$$\forall i \in [n], \ P_i(e; A_{-i}^{(n)}) = 1 \ \text{if } e \geq t; \ \text{and } P_i(e; A_{-i}^{(n)}) \leq n^{-\alpha} + n^{-(1-\alpha)} \ \text{if } e < t.$$

For preparation, we define the following function that is useful for computing the winning probability $P_i(e; A_{-i})$ for the undifferentiated contest.

**Definition E.3 (Function for computing winning probability).** *Given integers $n, k \geq 1$ and a density $p_1$ supported on $\Omega \subseteq \mathbb{R}_{\geq 0}$, we denote a function $Q_p^{(n,k)} : \Omega \to \mathbb{R}_{\geq 0}$ to be for any $v \in \Omega$,*

$$Q_p^{(n,k)}(v) = \sum_{i=n-k}^{n-1} \binom{n-1}{i} \cdot F_1(v)^i \cdot (1 - F_1(v))^{n-i-1} = \sum_{i=n-k}^{n-1} B(n-1, i, F_1(v)), \quad (13)$$

*where $B(n, k, x) = \binom{n}{k} x^k (1-x)^{n-k}$ is the Bernstein polynomial.*

By definition, $Q_p^{(n,k)}(v)$ represents the probability that, when sampling $n - 1$ independent and identically distributed (i.i.d.) values $v_1, \ldots, v_{n-1}$ from distribution $p_1$, the value $v$ ranks among the top $k$ values in the set $\{v_1, \ldots, v_{n-1}, v\}$. Given its algebraic significance, the function $Q_p^{(n,k)}(\cdot)$ is monotonically increasing to $v$ across the domain $\Omega$. This means that as $v$ increases, the probability of $v$ being in the top $k$ also increases. Also note that for any integers $n, n' \geq 1$ with $n < n'$,

$$Q^{(n,k)}(v) \geq Q^{(n',k)}(v). \quad (14)$$

This means that as the number of agents $n$ increases, $v$ is less likely to be in the top $k$. This function can be used to compute $P_i(e; A_{-i})$ for the undifferentiated contest in the following sense.

**Lemma E.5 (Computation of winning probability for the undifferentiated contest).** *Let $n, k \geq 1$ be integers and $p_1$ be a density supported on $\Omega \subseteq \mathbb{R}_{\geq 0}$. Let $A = (A_1, \ldots, A_n)$ be a symmetric policy for the undifferentiated contest satisfying that every $A_i$ is strictly monotonically increasing to $v$ across the domain $\Omega$. Then for every $i \in [n]$ and $v \in \Omega$, we have $P_i(A_i(v); A_{-i}) = Q_p^{(n,k)}(v)$.*

**Proof:** By symmetric, we only need to prove the lemma for $i = n$, i.e., proving $P_n(A_n(v); A_{-n}) = Q_p^{(n,k)}(v)$. Let $v_1, \ldots, v_{n-1}$ be i.i.d. samples from $p_1$. Since $A_i$ is strictly monotonically increasing to $v$ across the domain $\Omega$, we note that the sequence $v, v_1, \ldots, v_{n-1}$ should have the same order as the sequence $A_n(v), A_1(v_1), \ldots, A_{n-1}(v_{n-1})$. Hence, $A_n(v)$ is among the top $k$ of $\{A_1(v_1), \ldots, A_{n-1}(v_{n-1}), A_n(v)\}$ if and only if $v$ is among the top $k$ of $\{v_1, \ldots, v_{n-1}, v\}$. By the definition of winning probabilities and Definition E.3, this implies that $P_n(A_n(v); A_{-n}) = Q_p^{(n,k)}(v)$. This completes the proof of Lemma E.5. $\square$

Now we are ready to prove Lemma E.4.

**Proof [:** of Lemma E.4] It suffices to prove for the case that both $F_1(t) > 0$ and $F_2(t) > 0$. Proof for the other two cases is identical. By Definition E.2, we have

$$\Delta_n = \min \left\{ F_1^{-1}(F_1(t) - \sqrt{\frac{\log n}{n}}), F_2^{-1}(F_2(t) - \sqrt{\frac{\log n}{n}}) \right\}.$$

Let event $E_1^{(n,e)}$ be that there are at least $(1 - F_1(t))(1 - \alpha)n$ agents in $G_1 \setminus \{i\}$ that put in effort larger than $e$; and let $E_2^{(n,e)}$ be that there are at least $(1 - F_2(t)) \cdot \alpha n$ agents in $G_2 \setminus \{i\}$ that put in effort larger than $e$. Note that

$$(1 - F_1(t))(1 - \alpha)n + (1 - F_2(t)) \cdot \alpha n = cn.$$

When $e \geq t$, we have

$$P_i(e; A_{-i}^{(n)}) \geq \Pr\left[\overline{E_1^{(n,e)}} \cup \overline{E_2^{(n,e)}}\right] \geq 1 - \Pr\left[E_1^{(n,e)}\right] - \Pr\left[E_2^{(n,e)}\right].$$

Then to prove $P_i(e; A_{-i}^{(n)}) = 1$, it suffices to show that $\Pr\left[E_1^{(n,e)}\right] = \Pr\left[E_2^{(n,e)}\right] = 0$. Note that by policy $A^{(n)}$, the maximum effort put in by an agent is $t \leq v$. Hence, no agent can put in effort larger than $e$, which implies that $\Pr\left[E_1^{(n,e)}\right] = \Pr\left[E_2^{(n,e)}\right] = 0$. This completes the proof of $P_i(e; A_{-i}^{(n)}) = 1$ when $e \geq t$.

When $e < t$, we note that if both $E_1^{(n,e)}$ and $E_2^{(n,e)}$ happen, there are at least $k$ agents that put in effort $t$. Since events $E_1^{(n,e)}$ and $E_2^{(n,e)}$ are independent, we have

$$P_i(e; A_{-i}^{(n)}) \leq 1 - \Pr\left[E_1^{(n,e)} \cap E_2^{(n,e)}\right] = 1 - \Pr\left[E_1^{(n,e)}\right] \cdot \Pr\left[E_2^{(n,e)}\right].$$

Then to prove $P_i(e; A_{-i}^{(n)}) \leq n^{-\alpha} + n^{-(1-\alpha)}$, it suffices to show that $\Pr\left[E_1^{(n,e)}\right] \geq 1 - n^{-(1-\alpha)}$ and $\Pr\left[E_2^{(n,e)}\right] \geq 1 - n^{-\alpha}$.

We first bound $\Pr\left[E_1^{(n,e)}\right]$. Note that there are at least $(1 - \alpha)n$ agents in $G_1 \cup \{i\}$. Also, note that an agent $j \in G_1 \setminus \{i\}$ puts in effort $A_j(v_j) > e$ if and only if their valuation $v_j \geq \Delta_n$ holds. Now consider the undifferentiated contest among $G_1 \cup \{i\}$ with $k_1 = (1 - F_1(t))(1 - \alpha)n$ and density $p_1$. By Lemma E.5, we have

$$
\begin{aligned}
\Pr\left[E_1^{(n,e)}\right] =\ & 1 - Q_{p_1}^{(|G_1 \cup \{i\}|, k_1)}(\Delta_n) && \text{(Lemma E.5)} \\
\geq\ & 1 - Q_{p_1}^{((1-\alpha)n, k_1)}(\Delta_n) && \text{(Ineq. (14))} \\
\geq\ & 1 - Q_{p_1}^{((1-\alpha)n, k_1)}\left(F_1^{-1}\left(F_1(t) - \sqrt{\frac{\log n}{n}}\right)\right) && \text{(Defn. of } \Delta_n\text{)} \quad (15) \\
=\ & 1 - \sum_{j=(1-\alpha)n-1-k_1}^{(1-\alpha)n-1} B\left((1-\alpha)n - 1, j, F_1(t) - \sqrt{\frac{\log n}{n}}\right). && \text{(Eq. (13))}
\end{aligned}
$$

Let $X_1, \ldots, X_n$ be $(1-\alpha)n - 1$ i.i.d. random variables where each $X_i = 0$ with probability $F_1(t) - \sqrt{\frac{\log n}{n}}$ and otherwise $X_i = 1$. We note that $\sum_{j=(1-\alpha)n-1-k_1}^{(1-\alpha)n-1} B\left((1-\alpha)n - 1, j, F_1(t) - \sqrt{\frac{\log n}{n}}\right)$ is equivalent to the probability that $\sum_{i \in [n-1]} X_i \leq k_1 - 1$. Also note that

$$\mathbb{E}\left[\sum_{i \in [(1-\alpha)n-1]} X_i\right] = ((1-\alpha)n - 1) \cdot \left(1 - F_1(t) + \sqrt{\frac{\log n}{n}}\right). \tag{16}$$

Then by the Chernoff bound, we have

$$
\begin{aligned}
& \sum_{j=(1-\alpha)n-1-k_1}^{(1-\alpha)n-1} B\left((1-\alpha)n - 1, j, F_1(t) - \sqrt{\frac{\log n}{n}}\right) \\
=\ & \Pr\left[\sum_{i \in [(1-\alpha)n-1]} X_i \leq k_1 - 1\right] \\
\leq\ & \Pr\left[\sum_{i \in [(1-\alpha)n-1]} X_i \leq \mathbb{E}\left[\sum_{i \in [(1-\alpha)n-1]} X_i\right] - (1-\alpha)n \cdot \sqrt{\frac{\log n}{n}}\right] && \text{(Eq. (16) and Defn. of } k_1\text{)} \\
\leq\ & e^{-\frac{2(1-\alpha)^2 n \log n}{(1-\alpha)n-1}} \leq n^{-(1-\alpha)}. && \text{(Chernoff bound)}
\end{aligned}
$$

Combining with Inequality (15), we prove that $\Pr\left[E_1^{(n,e)}\right] \geq 1 - n^{-(1-\alpha)}$. By a similar argument, we can also prove $\Pr\left[E_2^{(n,e)}\right] \geq 1 - n^{-\alpha}$. Overall, we prove that $P_i(e; A_{-i}^{(n)}) \leq n^{-\alpha} + n^{-(1-\alpha)}$ when $e < t$. This completes the proof of Lemma E.4. ∎

### E.3.3  Proof that $A^{(n)}$ is approximate NE

Based on Lemma E.4, we are now ready to prove the approximate degree of $A^{(n)}$ to be an NE policy.

**Lemma E.6** ($A^{(n)}$ **is approximate NE**). *For any $n \geq n_t$, $A^{(n)}$ is an $\varepsilon_n$-NE policy, where $\varepsilon_n = (n^{-\alpha} + n^{-(1-\alpha)})\Delta_n + t - \Delta_n$.*

**Proof:** Fix $\ell = 1, 2$, $i \in G_\ell$, and $v, e \in \Omega_\ell$. We discuss the value $\pi_i(v, e; A_{-i}^{(n)}) - \pi_i(v, A_i^{(n)}(v); A_{-i}^{(n)})$. By Lemma E.4, we know that

$$\pi_i(v, e; A_{-i}^{(n)}) = v - e \text{ if } e \geq t; \text{ and } \pi_i(v, e; A_{-i}^{(n)}) \leq (n^{-\alpha} + n^{-(1-\alpha)})v - e \text{ if } e < t.$$

Then if $v < \Delta_n$, we have $\pi_i(v, A_i^{(n)}(v); A_{-i}^{(n)}) = \pi_i(v, 0; A_{-i}^{(n)}) = 0$, which implies that

$$\pi_i(v, e; A_{-i}^{(n)}) - \pi_i(v, A_i^{(n)}(v); A_{-i}^{(n)}) \leq \begin{cases} (n^{-\alpha} + n^{-(1-\alpha)})v - e \leq (n^{-\alpha} + n^{-(1-\alpha)})\Delta_n & \text{if } e < t \\ v - e \leq 0 & \text{if } e \geq t \end{cases}$$

Otherwise if $v \geq \Delta_n$, we have $\pi_i(v, A_i^{(n)}(v); A_{-i}^{(n)}) = \pi_i(v, t; A_{-i}^{(n)}) = v - t$, which implies that

$$\pi_i(u, v; A_{-i}^{(n)}) - \pi_i(u, A_i^{(n)}(v); A_{-i}^{(n)}) \leq \begin{cases} (n^{-\alpha} + n^{-(1-\alpha)})v - e + t - v & \text{if } e < t \\ t - e \leq 0 & \text{if } e \geq t \end{cases}$$

Note that when $v \geq \Delta_n$ and $e < t$,

$$(n^{-\alpha} + n^{-(1-\alpha)})v - e + t - v \leq (n^{-\alpha} + n^{-(1-\alpha)})\Delta_n + t - \Delta_n.$$

Overall, we conclude that the following inequality always holds:

$$\pi_i(v, e; A_{-i}^{(n)}) - \pi_i(v, A_i^{(n)}(v); A_{-i}^{(n)}) \leq (n^{-\alpha} + n^{-(1-\alpha)})\Delta_n + t - \Delta_n = \varepsilon_n.$$

This verifies that $A^{(n)}$ is an $\varepsilon_n$-NE policy for the two-group contest. □

### E.3.4  Completing the proof of Theorem E.2

**Proof [: of Theorem E.2]** Assume $p_a$ is a point mass at 0. We first prove that $\lim_{n \to \infty} s^{(n)} = s$. This is a direct corollary of the fact that $\lim_{n \to \infty} \Delta_n = t$. Consequently, for any $v \in \mathbb{R}_{\geq 0}$, there exists $n_v$ such that for any integer $n \geq n_v$, $s^{(n)}(v) = s(v)$ holds.

By Lemma E.6, $A^{(n)}$ is a $\varepsilon_n$-NE policy, where $\varepsilon_n = (n^{-\alpha} + n^{-(1-\alpha)})\Delta_n + t - \Delta_n$. Since $\lim_{n \to \infty} \Delta_n = t$, we have

$$\lim_{n \to \infty} \varepsilon_n = \lim_{n \to \infty} (n^{-\alpha} + n^{-(1-\alpha)})\Delta_n + t - \Delta_n = 0,$$

This completes the proof of Equation (12).

**Uniqueness of $A$.**  To prove that $A$ is the unique NE, we first recall Corollary 3.2 of [20] that says that a subset of symmetric agents should have the same policy in an NE. Thus, assuming $A'$ is an NE policy for the two-group contest as $n \to \mathbb{R}_{\geq 0}$, all agents $i \in G_1$ use a common threshold policy $s_1$, and those in $G_2$ use $s_2$. Suppose the threshold for $s_\ell$ is $t_\ell$. We next prove that $t_1 = t_2$, which implies that $s_1 = s_2$. When $t_1 > t_2$, if an agent puts in effort $t_1$, then it will get selected. Thus, the probability of an agent in $G_1$ getting selected is $1 - F_1(t_1)$. Hence, if $1 - F_1(t_1) < \frac{c}{1-\alpha}$, fewer than $cn$ agents in $G_1$ get selected. Thus, agents in $G_1$ getting selected will find that putting in effort slightly larger than $t_2$ instead of $t_1$ suffices to ensure their effort is larger than all agents in $G_2$, and consequently, they will still be selected. Through this reduction in effort, they can gain an additional payoff of $t_1 - t_2$, which violates the stability condition. A similar argument holds for $G_2$ when

$1 - F_1(t_1) > \frac{c}{1-\alpha}$. Thus, $A'$ is not an NE when $t_1 > t_2$. Similarly, we can prove that $A'$ is not an NE when $t_1 < t_2$. Thus, we must have $t_1 = t_2 = t$ and $s_1 = s_2 = s$.

If $(1-\alpha)F_1(t) + \alpha F_2(t) > 1 - c$, then fewer than $cn$ agents put in a non-zero effort and get selected. Thus, an agent with valuation $t' < t$, has a willingness to put in an effort $\varepsilon$ slightly larger than 0 instead of 0. Through this increase in effort, it can gain an additional payoff of $t' - \varepsilon$. Thus, $A'$ is not an NE. Similarly, we can prove that $A'$ is not an NE if $(1-\alpha)F_1(t) + \alpha F_2(t) < 1 - c$. Thus, for an NE policy, $t$ must be the solution of $(1-\alpha)F_1(v) + \alpha F_2(v) = 1 - c$. By Lemma E.1, the equation $(1-\alpha)F_1(v) + \alpha F_2(v) = 1 - c$ has a unique solution. Thus, $A' = A$, which proves the uniqueness.

**Extension to general $p_a$.** For general $p_a$, the solution $t$ of Equation (3) represents a score that a $c$-fraction of agents can achieve without making their expected payoff negative ($v + a \geq t$). The proof is almost identical to that when $p_a$ is a point mass at 0, except that the effort an agent with $v + a \geq t$ is willing to put in should be $s(v, a) = \max\{t - a, 0\}$ instead of $t$. This is because $t$ represents the score that the agent aims to reach, rather than the effort itself.

Overall, we complete the proof of Theorem E.2. ∎

**Remark E.7** (**Extension of Theorem 3.1**). *Using the same proof technique, Theorem 3.1 can be extended to handle multiple groups and non-identical effort costs. Let $G_1, \ldots, G_m$ represent $m \geq 2$ groups, where $|G_\ell| = \alpha_\ell n$ and the valuation density for each group is $p_\ell$ over the domain $\Omega_\ell \subseteq \mathbb{R}_{\geq 0}$. Each $\alpha_\ell \in (0, 1)$ satisfies the condition $\sum_{\ell \in [m]} \alpha_\ell = 1$. Recall that $p_a$ represents the initial ability density over the domain $\Omega_a \subseteq \mathbb{R}_{\geq 0}$, and we introduce an additional effort cost density $p_\kappa$ over the domain $\Omega_\kappa \subseteq \mathbb{R}_{>0}$. Each agent $\bar{i} \in [n]$ knows its type $\theta_i = (v_i, a_i, \kappa_i)$ and selects an effort level $e_i \geq 0$. The agent's score is given by $m(e_i + a_i)$, and their expected payoff is $P_i v_i - \kappa_i e_i$. It is important to note that agents' costs of effort $\kappa_i$ may vary and affect only their expected payoff, not their score.*

*In this extended multi-group contest, for $\ell \in [m]$, we extend CDF $F_\ell$ to be $F_\ell(\zeta) = \Pr_{v \sim p_\ell, a \sim p_a, \kappa \sim p_\kappa}\left[\frac{v}{\kappa} + a \leq \zeta\right]$. Now suppose domains $\bigcup_{\ell \in [m]} \Omega_\ell, \Omega_a, \Omega_\kappa$ are connected and densities $p_\ell, p_a, p_\kappa$ are positive at any point of their own domains. Let $t$ be the unique solution to the equation $\sum_{\ell \in [m]} \alpha_\ell F_\ell(\zeta) = 1 - c$. The infinite NE policy (3) extends to:*

$$s(v, a) := 0 \text{ if } \frac{v}{\kappa} + a < t \text{ and } s(v) := \max\{t - a, 0\} \text{ if } \frac{v}{\kappa} + a \geq t.$$

### E.4 Comparing with a distributional two-group contest

Recall that the technical challenges for Theorem 3.1 are mainly caused by the asymmetric strategic environment across groups. To avoid asymmetry, one may consider the following variant of the two-group contest. Note that for simplicity, we also assume that $p_a$ is a point mass at 0.

**Definition E.4** (**Distributional two-group contest**). *Let $n \geq k \geq 1$ be integers, $\alpha \in (0, 1)$, $\rho \in (0, 1]$, and $p_\ell$ be a density supported on a domain $\Omega_\ell \subseteq \mathbb{R}_{\geq 0}$ for $\ell = 1, 2$. Let each agent $i \in [n]$ belong to $G_1$ with probability $1 - \alpha$ and belong to $G_2$ with probability $\alpha$ independently. Let the valuation of each agent in $G_1$ be drawn i.i.d. from $p_1$, and the valuation of each agent in $G_2$ be drawn i.i.d. from $p_2$. Assume that each agent $i \in G_\ell$ ($\ell = 1, 2$) knows $n_1, n_2, k, p_1, p_2$, the group it belongs to and its valuation, and has to choose a policy $A_i : \Omega_\ell \to \mathbb{R}_{\geq 0}$ to maximize its expected payoff. The goal of the distributional two-group contest is to compute the NE policy satisfying Equation (1).*

The main difference from the two-group contest is that this distributional variant's group identity is random and the valuation density of each agent is identical, say $p = (1 - \alpha)p_1 + \alpha p_2$. Thus, using a similar argument as in an undifferentiated contest, it is easy to verify that the NE policy of the distributional two-group contest is identical to that of an undifferentiated contest with density $p_1$. Consequently, in the infinite $n$ case, the NE policy of the distributional two-group contest is identical to that of the two-group contest, say $A$ in Theorem 3.1. Then one may wonder whether this distributional two-group contest can also be used to simplify the proof of Theorem 3.1, as the infinite contest does. Below, we show that this is not the case and discuss the essential differences between the two models. For simplicity, we call the two-group contest *Model I* and call the distributional two-group contest *Model II*.

**Model distinction.** Firstly, Model I itself is of relevant interest as it captures real-world examples in which group sizes are well understood, while in Model II the group sizes are random variables.

**Convergence distinction.** Though Model I and Model II share the same NE policy $A$ in the infinite case, we would like to clarify that our main convergence result (Theorem E.2) cannot be inferred simply from knowing that the limit of the NEs of the two models is the same. To put it in simplest terms, consider two sequences $a_1, \ldots, a_n, \ldots$ and $b_1, \ldots, b_n, \ldots$ that converge to the same limit point $z$. The proof of convergence for the first sequence does not necessarily provide any information about the convergence of the second sequence. Therefore, the convergence result for our model cannot be simply inferred from prior work.

**Analysis distinction.** Moreover, the analysis of Model II relies heavily on symmetric policies for all agents (enabled precisely by the fact that group sizes are random), allowing the use of order statistics of $p_1$. In contrast, in Model I, we expect asymmetric policies across groups. For example, consider a two-agent case with $k = 1$: Agent 1's valuation follows a uniform distribution on $[0, 0.5]$, while Agent 2's valuation follows a uniform distribution on $[0.5, 1]$. In Section B, we show that the NE policy for this example must be asymmetric. Any symmetric policy $A$ would result in a near-zero winning probability for Agent 1, leading to a negative expected payoff and implying that $A$ is not an NE. This negative payoff arises from the asymmetric strategic environment faced by Agents 1 and 2, where the density of the highest valuation among other agents differs for each agent. Consequently, the order of winning probabilities of agents ($P_i$) can differ from the order of valuations ($v_i$), posing a significant mathematical challenge for determining the NE. E.g., for strategies $s_1$ and $s_2$, let $F_{s_\ell}(v)$ denote the cumulative distribution of efforts $s_\ell(v)$ when $v \sim p_\ell$. The cumulative distribution of the $(k-1)$-th effort $e^\star$ from an agent in $G_1$ is then given by:

$$
\begin{aligned}
&\Pr[e^\star \leq v] \\
&= \sum_{i=0}^{n-1} \sum_{j=(1-c)n-i}^{n-1-i} \binom{n-1}{i} \binom{n-1-i}{j} F_{s_1}(v)^i (1 - F_{s_1}(v))^{(1-\alpha)n-1-i} F_{s_2}(v)^j (1 - F_{s_2}(v))^{\alpha n - j}.
\end{aligned}
$$

Compare this to the expression for the symmetric case

$$
\Pr[e^\star \leq v] = \sum_{i=(1-c)n}^{n-1} \binom{n-1}{i} F_s(v)^i (1 - F_s(v))^{n-1-i}.
$$

Thus, the calculus and approximations for the expression for the two-group contest are significantly more difficult, making it much harder to arrive at the equilibrium policies than for Model II.

### E.5 An alternative proof using an infinite contest

Recall that Theorem 3.1 studies the case of $n \to \infty$ for the two-group contest. To increase the understanding of the infinite case, we propose the following infinite version of the two-group contest, where every real number in the interval $[0, 1 - \alpha]$ corresponds to an agent in $G_1$ and in the interval $(1 - \alpha, 1]$ corresponds to an agent in $G_2$. For simplicity, we still assume that $p_a$ is a point mass at 0. Formally, we provide the following definition.

**Definition E.5 (Infinite contest).** *Let $k \geq 1$ be integers, $\alpha \in (0, 1)$, $\rho \in (0, 1]$, and $p_\ell$ be a density supported on a domain $\Omega_\ell \subseteq \mathbb{R}_{\geq 0}$ for $\ell = 1, 2$. Let every real number in the interval $[0, 1 - \alpha]$ correspond to an agent in group $G_1$, and in the interval $(1 - \alpha, 1]$ correspond to an agent in group $G_2$. For $\ell \in \{1, 2\}$, let the valuation of every agent in $G_\ell$ draw i.i.d. from $p_\ell$. Assume that each agent $i \in G_\ell$ ($\ell = 1, 2$) knows $\alpha$, $k$, $p_1$, $p_2$, the group it belongs to and its valuation, and has to choose a policy $A_i : \Omega_\ell \to \mathbb{R}_{\geq 0}$ to maximize its expected payoff.*

There are countless agents in this infinite contest. Also, note that $G_1$ contains $(1 - \alpha)$-fraction of agents while $G_2$ contains the remaining $\alpha$-fraction. Below, we show how to use this infinite contest to hypothesize the NE policy $A$ for the two-group contest defined in Theorem 3.1. It mainly consists of two steps: Showing that two-group contests converge to the infinite contest as $n \to \infty$ and computing NE for the infinite contest.

**Showing two-group contests converge to the infinite contest as $n \to \infty$.** We first show that the infinite contest is the limit of two-group contests. Let $g_n$ denote a two-group contest as defined in

the two-group contest with an NE policy $A_n$. Let $g$ denote the infinite game as defined in Problem E.5. We view $g_n$ as a collection of $n$ density functions of agents, with the $i$-th agent represented by the real number $\frac{i-1}{n-1}$. Agent $i$ belongs to group $G_1$ if $1 \leq i \leq (1-\alpha)n$ and to group $G_2$ otherwise. From this viewpoint, we propose the following theorem.

**Theorem E.8** (**Two-group contests converge to the infinite contest**). *$g_n$ converges to $g$ in the following sense: For any $\varepsilon > 0$, there exists a sufficiently large $n_0$ such that for all $n \geq n_0$,*

- *For any $t \in [0, 1-\alpha]$, $|\int_0^t dx - \frac{|\{i \in G_1 : \frac{i-1}{n-1} \leq t\}|}{n}| \leq \varepsilon$, i.e., the difference in the fraction of agents in $G_1$ associated with real number at most $t$ between $g$ and $g_n$, is at most $\varepsilon$.*

- *For any $t \in (1-\alpha, 1]$, $|\int_t^1 dx - \frac{|\{i \in G_2 : \frac{i-1}{n-1} \geq t\}|}{n}| \leq \varepsilon$, i.e., the difference in the fraction of agents in $G_2$ associated with real number at least $t$ between $g$ and $g_n$, is at most $\varepsilon$.*

Note that the agents in $g_n$ can be captured by a uniform distribution $\mu_n$ over real numbers $\frac{i-1}{n-1}$ ($i \in [n]$). The convergence conditions in the theorem state that the limit of $\mu_n$ is the uniform density $\mu$ over $[0, 1]$, where $\mu$ represents the density of agents in $g$.

**Proof:**[of Theorem E.8] Let $n_0 = \lceil \varepsilon^{-1} \rceil$. Then we have $n \geq \varepsilon^{-1}$. For any $t \in [0, 1-\alpha]$, we have

$$\left| \int_0^t dx - \frac{\left|\left\{i \in G_1 : \frac{i-1}{n-1} \leq t\right\}\right|}{n} \right| = \left| t - \frac{\lfloor (n-1)t + 1 \rfloor}{n} \right|$$

and

$$t - \frac{t}{n} \leq \frac{\lfloor (n-1)t + 1 \rfloor}{n} \leq t + \frac{1-t}{n}.$$

We conclude that

$$\left| \int_0^t dx - \frac{\left|\left\{i \in G_1 : \frac{i-1}{n-1} \leq t\right\}\right|}{n} \right| \leq \max\left\{ \frac{t}{n}, \frac{1-t}{n} \right\} \overset{t \in [0, 1-\alpha]}{\leq} \frac{1}{n} \overset{n \geq \varepsilon^{-1}}{\leq} \varepsilon.$$

Similarly, for any $t \in (1-\alpha, 1]$, we can prove that $\left| \int_t^1 dx - \frac{|\{i \in G_2 : \frac{i-1}{n-1} \geq t\}|}{n} \right| \leq \varepsilon$. This completes the proof of Theorem E.8. $\qquad \square$

**Computing NE for the infinite contest.** It follows from Theorem E.8 that the limit of the two-group contests $g_n$ is the infinite contest $g$. Then, assuming the NE policy of $g_n$ is $A^{(n)}$, we can infer that the limit of $A^{(n)}$ is the NE policy of the infinite contest. Thus, to hypothesize the NE policy for $g_n$ as $n \to \infty$, it suffices to compute the NE policy for the infinite contest.

We first observe that the strategic environment for all agents in the infinite contest is the same, i.e., the probability that a given valuation $v$ is among the top $c$-fraction is the same for all agents. This property reduces the infinite contest to an undifferentiated contest (except for the different domains of valuation densities), leading to a symmetric NE policy. Formally, we provide the following lemma that shows that $A$ is exactly the unique NE policy for the infinite contest.

**Lemma E.9** (**The infinite contest**). *Suppose $\Omega_1 \cup \Omega_2$ is connected and each density $p_\ell$ is positive at any point of domain $\Omega_\ell$. Then policy $A$ defined in Equation (3) is the unique NE policy for the infinite contest.*

**Proof:** We first note that for any agent (whether in $G_1$ or $G_2$), the probability that a given valuation $v$ is among the top $c$-fraction is given by:

$$p_1(v) := \lim_{n \to \infty} \sum_{i=(1-c)n}^{n-1} \binom{n-1}{i} F(v)^i (1 - F(v))^{n-1-i},$$

where $F$ is the CDF of the joint density $(1-\alpha)p_1 + \alpha p_2$. Recall that $t$ is the unique solution to the equation

$$F(v) = (1-\alpha)F_1(v) + \alpha F_2(v) = 1 - c,$$

i.e., $t = F^{-1}(1 - c)$. Then through a straightforward calculation, it follows that $p_1(v) = 1$ for $v > t$ and $p_1(v) = 0$ for $v < t$. Since the winning probability function $p_1$ is identical for all agents, the strategic environment for all agents in the infinite contest is the same. Recall that by Corollary 3.2 of [20], symmetric agents will use a symmetric policy in an NE. Thus, we can assume an increasing symmetric policy $s : \Omega_1 \cup \Omega_2 \to \mathbb{R}_{\geq 0}$ for all agents.

By the equilibrium condition, for any valuation $v$ and effort $e$,

$$p_1(v)v - s(v) \geq P(s^{-1}(e))v - e.$$

By a similar argument as for Equation (10) (undifferentiated contest), we can compute that $s(v) = t$ for $v > t$ and $s(v) = 0$ for $v < t$. This turns out to be the threshold function defined in Equation (3). Thus, the policy $A$, where each agent restricts $s$ to its valuation the domain, is indeed the unique NE for the infinite contest. This completes the proof of Lemma E.9. $\qquad\square$

**Using the infinite game to provide an alternative proof of Theorem E.2.**    As shown in Lemma E.9, instead of relying on observations from the finite case as in Section E.1.2, we can use this infinite contest to hypothesize the NE policy $A$ for the two-group contest in the infinite $n$ case.

Once we have a solid guess for the NE policy $A$ using the infinite contest, we need to show that $A$ remains an NE as $n \to \infty$. While this approach simplifies the initial hypothesis, the challenge remains in proving that $A$ is indeed an NE. Similar to our current proof of Theorem 3.1, we must construct a series of proxies that converge to $A$ and increasingly approximate an NE policy. As detailed in Section E.2, this step remains technically challenging.

Overall, using the infinite contest could provide an alternative proof of Theorem E.2. Moreover, the symmetric strategic environment of this infinite contest can provide deeper insights into why the NE policy remains symmetric, even when valuations are asymmetric across groups.

# F    Omitted details for uniform distribution analysis from Sections 3 and 4

**Theorem F.1** (**Restatement of Theorem 3.2**).  *Assume $p_2(v) = \frac{1}{\rho} p_1\left(\frac{v}{\rho}\right)$ for some $\rho \in (0, 1]$ and $p_a$ is a mass point at 0. Let policy $A$ be defined as in Theorem 3.1, characterized by $t$ being the unique solution of Equation (4). Then for any density $p_1$,*

$$r_{\mathcal{R}}(A) = \frac{1 - F_1(t/\rho)}{1 - F_1(t)}, \ r_{\mathcal{S}}(A) = \frac{\rho \int_{t/\rho}^{\infty}(v - t/\rho)p_2(v)dv}{\int_t^{\infty}(v - t)p_1(v)dv}, \ and \ \mathcal{RV}(A, m) = m(t).$$

*Moreover, $r_{\mathcal{R}}(A)$ is monotonically increasing w.r.t. $\rho$, $c$, and $\alpha$, while $\mathcal{RV}(A, m)$ is monotonically increasing w.r.t. $\rho$ and monotonically decreasing w.r.t. $c$ and $\alpha$, for any merit function $m$.*

**Proof:** We discuss three metrics separately.

**Metric $\mathcal{RV}(A, m)$.**    Recall that $A$ is a threshold function characterized by $t$. Thus, agents with the sum of valuation and initial ability $v + a > t$ get spots. Since $p_a$ is a point mass at 0, we know that the score of each selected agent is exactly $t$. Thus, the average revenue $\mathcal{RV}(A, m) = m(t)$.

Next, we prove the monotonicity of $\mathcal{RV}(A, m)$. Since $m$ is monotonically increasing, we only need to prove the monotonicity of $t$ with respect to $\rho, c, \alpha$. Recall that $t$ is the solution of Equation (4), which can be rewritten as

$$(1 - \alpha)F_1(\zeta) + \alpha F_1\left(\frac{\zeta}{\rho}\right) = 1 - c.$$

Let $f(\zeta) = (1 - \alpha)F_1(\zeta) + \alpha F_1\left(\frac{\zeta}{\rho}\right) + c$. Then $t$ is the solution of $f(\zeta) = 1$.

Since $F_1\left(\frac{\zeta}{\rho}\right)$ is a monotonically decreasing function of $\rho$, we know that $f(\zeta)$ is also a monotonically decreasing function of $\rho$. Thus, the solution $t$ increases with $\rho$. Since $F_a(\zeta - \rho v) \geq F_a(\zeta - v)$, $f(\zeta)$ is an increasing function of $\alpha$. Also note that $f(\zeta)$ is an increasing function of $c$. Thus, as $c$ or $\alpha$ increase, solution $t$ decreases.

Overall, we prove that $\mathcal{RV}(A, m)$ is monotonically increasing w.r.t. $\rho$ and monotonically decreasing w.r.t. $c$ and $\alpha$, for any merit function $m$.

**Metric $r_{\mathcal{R}}(A)$.** Recall that $F_\ell$ is a cumulative density function (CDF) of the sum of valuation and initial ability such that for any $\zeta \in \mathbb{R}_{\geq 0}$, $F_\ell(\zeta) = \Pr_{v \sim p_\ell, a \sim p_a}[v + a \leq \zeta]$. Thus, $F_\ell$ is the CDF of $p_\ell$ when $p_a$ is a point mass at 0. Then the linearity of expectation yields:

$$\mathbb{E}_{v_i, a_i}[|S \cap G_1|] = (1 - \alpha)n(1 - F_1(t)) \text{ and } \mathbb{E}_{v_i, a_i}[|S \cap G_2|] = \alpha n(1 - F_2(t)).$$

This translates to:

$$\mathbb{E}[\mathcal{R}_1(A)] = \frac{\mathbb{E}_{v_i, a_i}[|S \cap G_1|]}{(1 - \alpha)n} = 1 - F_1(t) \text{ and } \mathbb{E}[\mathcal{R}_2(A)] = \frac{\mathbb{E}_{v_i, a_i}[|S \cap G_2|]}{\alpha n} = 1 - F_2(t).$$

Since $p_2(v) = \frac{1}{\rho} p_1\left(\frac{v}{\rho}\right)$, we know that $F_2(t) = F_1(t/\rho)$ and $F_1(t) \leq F_2(t)$. Thus, we have $\mathbb{E}[\mathcal{R}_2(A)] \leq \mathbb{E}[\mathcal{R}_1(A)]$.

As $n \to \infty$, it suffices to prove that

$$r_{\mathcal{R}}(A) \to \min\left\{\frac{\mathbb{E}[\mathcal{R}_1(A)]}{\mathbb{E}[\mathcal{R}_2(A)]}, \frac{\mathbb{E}[\mathcal{R}_2(A)]}{\mathbb{E}[\mathcal{R}_1(A)]}\right\} = \frac{1 - F_2(t)}{1 - F_1(t)} = \frac{1 - F_1(t/\rho)}{1 - F_1(t)}.$$

We first note that $|S \cap G_\ell|$ is highly concentrated at $\mathbb{E}_{v_i, a_i}[|S \cap G_1|]$ since all agents in $G_\ell$ are i.i.d. Concretely, the following inequality holds for any $t > 0$ by the Chernoff bound:

$$\Pr\left[||S \cap G_\ell| - \mathbb{E}_{v_i, a_i}[|S \cap G_\ell|]| \geq t \cdot \mathbb{E}_{v_i, a_i}[|S \cap G_\ell|]\right] \leq 2e^{-\frac{t^2 \cdot \mathbb{E}_{v_i, a_i}[|S \cap G_\ell|]}{3}}.$$

Hence, for $t = o(\sqrt{\frac{1}{n}})$, we have $\Pr\left[||S \cap G_\ell| - \mathbb{E}_{v_i, a_i}[|S \cap G_\ell|]| \geq t \cdot \mathbb{E}_{v_i, a_i}[|S \cap G_\ell|]\right] \to 0$. This implies that as $n \to \infty$,

$$r_{\mathcal{R}}(A) = \mathbb{E}_{v_i, a_i}\left[\min\left\{\frac{\mathcal{R}_2(A)}{\mathcal{R}_1(A)}, \frac{\mathcal{R}_1(A)}{\mathcal{R}_2(A)}\right\}\right] \to \min\left\{\frac{\mathbb{E}[\mathcal{R}_1(A)]}{\mathbb{E}[\mathcal{R}_2(A)]}, \frac{\mathbb{E}[\mathcal{R}_2(A)]}{\mathbb{E}[\mathcal{R}_1(A)]}\right\},$$

which completes the proof of the formula of $r_{\mathcal{R}}(A)$.

Next, we prove the monotonicity of $r_{\mathcal{R}}(A)$ with respect to $\rho$. Recall that $t$ is monotonically increasing with $\rho$. We know that $1 - F_1(t)$ is a monotonically decreasing function of $\rho$. Since $1 - F_2(t) = \frac{1 - c - (1 - \alpha)(1 - F_1(t))}{\alpha}$, we know that $1 - F_2(t)$ is monotonically increasing with $\rho$. Thus, $r_{\mathcal{R}}(A) = \frac{1 - F_1(t/\rho)}{1 - F_1(t)}$ is an increasing function of $\rho$.

**Metric $r_{\mathcal{S}}(A)$.** By a similar argument as for $r_{\mathcal{R}}(A)$, we first have that as $n \to \infty$,

$$r_{\mathcal{S}}(A) \to \min\left\{\frac{\mathbb{E}[\mathcal{S}_1(A)]}{\mathbb{E}[\mathcal{S}_2(A)]}, \frac{\mathbb{E}[\mathcal{S}_2(A)]}{\mathbb{E}[\mathcal{S}_1(A)]}\right\}.$$

Also note that

$$\mathbb{E}[\mathcal{S}_\ell(A)] = \mathbb{E}\left[\frac{1}{|G_\ell|}\sum_{i \in G_\ell}(\mathbb{I}(i \text{ selected}) \cdot v_i - e_i)\right] = \int_t^\infty (v - t)p_\ell(v)dv.$$

Then we have

$$\mathbb{E}[\mathcal{S}_2(A)] = \frac{1}{\rho}\int_t^\infty (v - t)p_1\left(\frac{v}{\rho}\right)dv = \int_{\frac{t}{\rho}}^\infty (\rho v - t)p_1(v)dv \leq \mathbb{E}[\mathcal{S}_1(A)].$$

Combining the above all, we obtain that $r_{\mathcal{S}}(A) = \frac{\int_t^\infty (v - t)p_2(v)dv}{\int_t^\infty (v - t)p_1(v)dv}$. Since $p_2(v) = \frac{1}{\rho}p_1\left(\frac{v}{\rho}\right)$, we have

$$r_{\mathcal{S}}(A) = \frac{\int_t^\infty (v - t)p_2(v)dv}{\int_t^\infty (v - t)p_1(v)dv} = \frac{\rho \int_{t/\rho}^\infty (v - t/\rho)p_1(v)dv}{\int_t^\infty (v - t)p_1(v)dv}$$

Next, we analyze the monotonicity of $r_{\mathcal{S}}(A)$ with respect to $\rho$. Let $g(x) = \int_x^\infty (v - x)p_1(x)dx$, which is monotonically decreasing of $x$. We have $r_{\mathcal{S}}(A) = \frac{\rho \cdot g(t/\rho)}{g(t)}$. Since $t$ is monotonically increasing with $\rho$, $g(t)$ is also monotonically increasing with $\rho$. Thus, to prove that $r_{\mathcal{S}}(A)$ is monotonically increasing with $\rho$, it suffices to prove that $t/\rho$ is monotonically decreasing with $\rho$. Recall that we have shown that $F_1(t/\rho) = F_2(t)$ is monotonically decreasing with $\rho$. This implies that $t/\rho$ is indeed monotonically decreasing with $\rho$, completing the proof.

Overall, we have completed the proof of the theorem. $\qquad\square$

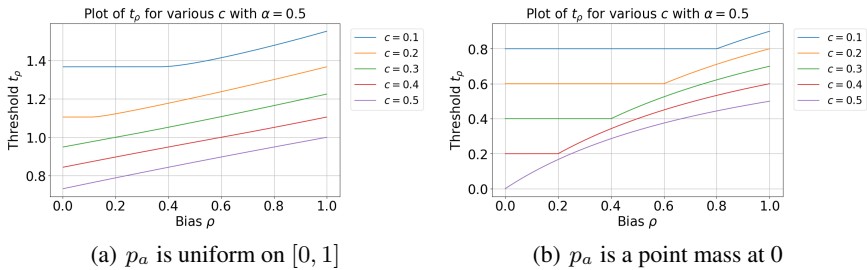

|  |  |
|---|---|
| (a) $p_a$ is uniform on $[0, 1]$ | (b) $p_a$ is a point mass at 0 |

Figure 13: Plots of $t$ versus $\rho$ for various $c$ with $\alpha = 0.5$ for the uniform distribution.

**Remark F.2** (**Monotonicity of** $r_{\mathcal{R}}(A)$ **and** $r_{\mathcal{S}}(A)$ **w.r.t.** $c, \alpha$). *By Theorem 3.2, we note that when fixing $\rho$, both $r_{\mathcal{R}}(A)$ and $r_{\mathcal{S}}(A)$ are functions of $t$. Since $t$ is monotonically decreasing with respect to $c$ and $\alpha$, we only need to investigate the monotonicity of $r_{\mathcal{R}}(A)$ and $r_{\mathcal{S}}(A)$ with respect to $t$. Proposition 4.1 demonstrates that $r_{\mathcal{R}}(A)$ and $r_{\mathcal{S}}(A)$ are monotonically decreasing with $t$, and hence, monotonically increasing with $c$ and $\alpha$. Below, we provide an example with $p_1$ to show that this monotonicity does not always hold.*

*Let $\varepsilon > 0$ be a sufficiently small number and $\rho = 0.5$. Let $p_1$ be supported on $\Omega_1 = [0, 2]$ such that $p_1(v) = 1 - \varepsilon$ for $v \in [0, 0.5] \cup [1.5, 2]$ and $p_1(v) = \varepsilon$ for $v \in (0.5, 1.5)$. Then $F_1(v) = (1 - \varepsilon)v$ for $v \in [0, 0.5]$, $0.5 - \varepsilon + \varepsilon v$ for $v \in (0.5, 1.5)$, and $(1 - \varepsilon)v + 2\varepsilon - 1$. By Theorem 3.2, we have $r_{\mathcal{R}}(A) = \frac{1 - F_1(t/\rho)}{1 - F_1(t)}$. Then in this case, $r_{\mathcal{R}}(A) = \frac{1 - (1 - \varepsilon)/2}{1 - (1 - \varepsilon)/4} \approx \frac{2}{3}$ when $t = 0.25$; while $r_{\mathcal{R}}(A) = \frac{1 - (1 - \varepsilon)/2 - 0.5\varepsilon}{1 - (1 - \varepsilon)/2} \approx 1$. Thus, $r_{\mathcal{R}}(A)$ is not monotonically decreasing with $t$. A similar computation can be done for $r_{\mathcal{S}}(A)$.*

By a similar argument as for Theorem 3.2, we provide the following formulas of metrics for general densities. The computation is straightforward and we omit here.

**Theorem F.3** (**Metrics in general**). *Let policy $A$ be defined as in Theorem 3.1, characterized by $t$ being the unique solution of Equation (2). Then for any densities $p_1$, $p_2$, and $p_a$,*

$$r_{\mathcal{R}}(A) = \min\left\{ \frac{1 - F_2(t)}{1 - F_1(t)}, \frac{1 - F_1(t)}{1 - F_2(t)} \right\},$$

$$r_{\mathcal{S}}(A) = \min\left\{ \frac{\int_t^\infty \int_{\Omega_2} \min\{v, \zeta - t\} p_2(v) p_a(\zeta - v) dv d\zeta}{\int_t^\infty \int_{\Omega_1} \min\{v, \zeta - t\} p_1(v) p_a(\zeta - v) dv d\zeta}, \frac{\int_t^\infty \int_{\Omega_1} \min\{v, \zeta - t\} p_1(v) p_a(\zeta - v) dv d\zeta}{\int_t^\infty \int_{\Omega_2} \min\{v, \zeta - t\} p_2(v) p_a(\zeta - v) dv d\zeta} \right\},$$

$$\mathcal{RV}(A) = \frac{1 - \alpha}{1 - c} \int_t^\infty \int_{\Omega_1} m(\max\{t - \zeta + v, 0\}) p_1(v) p_a(\zeta - v) dv d\zeta$$
$$+ \frac{\alpha}{1 - c} \int_t^\infty \int_{\Omega_2} m(\max\{t - \zeta + v, 0\}) p_1(v) p_a(\zeta - v) dv d\zeta.$$

In the following, we complete the analysis from Sections 3 and 4 for the case where $p_1$ and $p_2$ are uniform densities. The visualization of $t$ for them can be found in Figure 13. We first give the proof of Equation (5) for the case that $p_a$ is uniform.

**Proposition F.4** (**Complete version of Equation (5)**). *Let $\alpha, c \in (0, 1)$ and $\rho \in (0, 1]$. Let $p_1$ be uniform on $[0, 1]$, $p_2$ be uniform on $[0, \rho]$, and $p_a$ be uniform on $[0, 1]$. Let $t \in [0, 2]$ be the solution to the equation $\int_0^1 (1 - \alpha) \cdot \min\{1, (\zeta - v)_+\} dv + \int_0^\rho \frac{\alpha}{\rho} \cdot \min\{1, (\zeta - v)_+\} dv = 1 - c$. Then if $0 < c \le \frac{1 - \alpha}{2}$,*

$$t = \begin{cases} 2 - \sqrt{\dfrac{2c}{1 - \alpha}}, & \rho < 1 - \sqrt{\dfrac{2c}{1 - \alpha}}, \\ \dfrac{2 - \alpha + \alpha/\rho}{1 - \alpha + \alpha/\rho} - \dfrac{\sqrt{2c(1 - \alpha + \alpha/\rho) - \alpha(1 - \alpha)(1 - \rho)^2/\rho}}{1 - \alpha + \alpha/\rho}, & \rho \ge 1 - \sqrt{\dfrac{2c}{1 - \alpha}}, \end{cases}$$

*if $\frac{1 - \alpha}{2} < c \le \frac{1}{2}$,*

$$\begin{cases} \dfrac{2 - \alpha + \alpha/\rho}{1 - \alpha + \alpha/\rho} - \dfrac{\sqrt{2c(1 - \alpha + \alpha/\rho) - \alpha(1 - \alpha)(1 - \rho)^2/\rho}}{1 - \alpha + \alpha/\rho}, & \rho < \dfrac{2c - 1 + \alpha}{\alpha}, \\ -\dfrac{\alpha}{1 - \alpha} + \dfrac{\sqrt{\alpha^2 + (1 - \alpha)(2 + \alpha\rho - 2c)}}{1 - \alpha}, & \rho \ge \dfrac{2c - 1 + \alpha}{\alpha}, \end{cases}$$

if $\frac{1}{2} < c \leq 1$,

$$\begin{cases} -\dfrac{\alpha}{1-\alpha} + \dfrac{\sqrt{\alpha^2 + (1-\alpha)(2 + \alpha\rho - 2c)}}{1-\alpha}, & \rho < \frac{-\alpha + \sqrt{\alpha^2 + 8(1-\alpha)(1-c)}}{2(1-\alpha)}, \\[3mm] \sqrt{\dfrac{2(1-c)}{1-\alpha+\alpha/\rho}}, & \rho \geq \frac{-\alpha + \sqrt{\alpha^2 + 8(1-\alpha)(1-c)}}{2(1-\alpha)}. \end{cases}$$

**Proof:** Let $f(\zeta) = \int_0^1 (1-\alpha) \cdot \min\{1, (\zeta - v)_+\}dv + \int_0^\rho \frac{\alpha}{\rho} \cdot \min\{1, (\zeta - v)_+\}dv$. We first note that $f(\zeta)$ is a monotone increasing function with $f(0) = 0$ and $f(2) = 1$. By analyzing the value of $\min\{1, (\zeta - v)_+\}$, we also have the following equation:

$$\min\{1, (\zeta - v)_+\} = \begin{cases} 0, & 0 \leq \zeta \leq v, \\ \zeta - v, & v \leq \zeta \leq v + 1, \\ 1, & v + 1 \leq \zeta \leq 2. \end{cases}$$

Accordingly, we know that

$$f(\zeta) = \begin{cases} (1-\alpha)(\zeta - v)\,|_0^\zeta + \dfrac{\alpha}{\rho}(\zeta - v)\,|_0^\zeta, & 0 \leq \zeta \leq \rho, \\[2mm] (1-\alpha)(\zeta - v)\,|_0^\zeta + \dfrac{\alpha}{\rho}(\zeta - v)\,|_0^\rho, & \rho \leq \zeta \leq 1, \\[2mm] (1-\alpha)\left(\zeta - 1 + (\zeta - v)\,|_{\zeta-1}^\zeta\right) + \dfrac{\alpha}{\rho}\left(\zeta - 1 + (\zeta - v)\,|_{\zeta-1}^\rho\right), & 1 \leq \zeta \leq 1 + \rho, \\[2mm] (1-\alpha)\left(\zeta - 1 + (\zeta - v)\,|_{\zeta-1}^\zeta\right) + \alpha, & 1 + \rho \leq \zeta \leq 2. \end{cases}$$

Thus, $f$ is a piecewise-polynomial function of $\zeta$. Solving $f(\zeta) = 1 - c$ results in Corollary F.4. $\square$

**Proposition F.5 (Restatement of Proposition 4.1).** *Let $p_1$ be uniform on $[0,1]$, $p_2$ be uniform on $[0, \rho]$, and $p_a$ be a point mass at 0. Let $A$ be the NE policy for the two-group contest as $n \to \infty$. Then*

$$t = 1 - \frac{c}{1-\alpha} \text{ if } \rho < 1 - \frac{c}{1-\alpha} \text{ and } t = \frac{\rho(1-c)}{\rho - \alpha\rho + \alpha} \text{ if } \rho \geq 1 - \frac{c}{1-\alpha}.$$

$$r_{\mathcal{R}}(A) = 0 \text{ if } \rho < 1 - \frac{c}{1-\alpha} \text{ and } r_{\mathcal{R}}(A) = \frac{\rho - \alpha\rho + \alpha + c - 1}{\alpha - \alpha\rho + c\rho} \text{ if } \rho \geq 1 - \frac{c}{1-\alpha}.$$

$$r_{\mathcal{S}}(A) = 0 \text{ if } \rho < 1 - \frac{c}{1-\alpha} \text{ and } r_{\mathcal{S}}(A) = \frac{\rho(\rho - \alpha\rho + \alpha + c - 1)^2}{(\alpha - \alpha\rho + c\rho)^2} \text{ if } \rho \geq 1 - \frac{c}{1-\alpha}.$$

*Moreover, $\mathcal{RV}(A, m) = m(t)$ for any merit function $m(\cdot)$; $r_{\mathcal{R}}(A)$ and $r_{\mathcal{S}}(A)$ are monotonically increasing functions of parameters $\rho, c, \alpha$.*

**Proof:** Note that $\mathcal{RV}(A, m) = m(t)$ is directly from Theorem 3.2.

**Computation of $t$.** Note that $F_1(v) = v$ and $F_2(v) = \min\left\{1, \frac{v}{\rho}\right\}$. Let $g(v) = (1-\alpha)v + \alpha \min\left\{1, \frac{v}{\rho}\right\}$. We note that $g(\cdot)$ is a piece-wise linear function of $v$ with an inflection point $v = \rho$. Plugging $v = \rho$ into the equation, we obtain that $\rho = 1 - \frac{c}{1-\alpha}$ which is an inflection point of $t$. Then if solution $t > \rho$, we have that $t$ is the solution of the equation $(1-\alpha)v + \alpha = 1 - c$, implying that $t = 1 - \frac{c}{1-\alpha}$. The condition for this case is $\rho < 1 - \frac{c}{1-\alpha} = t$. Otherwise if solution $t \leq \rho$, we have that $t$ is the solution of the equation $(1-\alpha)v + \alpha\frac{v}{\rho} = 1 - c$, implying that $t = \frac{\rho(1-c)}{\rho - \alpha\rho + \alpha}$. The condition for this case is $\rho \geq 1 - \frac{c}{1-\alpha}$. This completes the proof for $t$.

**Analysis for $r_{\mathcal{R}}(A)$.** By Theorem 3.2, we have $r_{\mathcal{R}}(A) = \frac{1 - \min\left\{1, \frac{t}{\rho}\right\}}{1 - t}$. By the form of $t$, we can verify the explicit form of $r_{\mathcal{R}}(A)$.

Note that when $\rho \geq 1 - \frac{c}{1-\alpha}$, we have

$$r_{\mathcal{R}}(A) = \frac{\rho - \alpha\rho + \alpha + c - 1}{\alpha - \alpha\rho + c\rho} = 1 - \frac{(1-c)(1-\rho)}{\alpha - \alpha\rho + c\rho}.$$

Let
$$f(\rho, c, \alpha) = 1 - \frac{(1-c)(1-\rho)}{\alpha - \alpha\rho + c\rho}.$$

Define the auxiliary functions:
$$X(\rho, c, \alpha) = (1-c)(1-\rho), \quad Y(\rho, c, \alpha) = \alpha - \alpha\rho + c\rho.$$

Then when $\rho \geq 1 - \frac{c}{1-\alpha}$, we have
$$f(\rho, c, \alpha) = 1 - \frac{X}{Y}, \text{ and } X(\rho, c, \alpha), Y(\rho, c, \alpha) > 0.$$

The partial derivatives w.r.t. $\rho, c, \alpha$ are:

$$\frac{\partial f}{\partial \rho} = \frac{(1-c)Y + (c-\alpha)X}{Y^2} = \frac{c(1-c)}{Y^2} \geq 0,$$

$$\frac{\partial f}{\partial c} = \frac{(1-\rho)Y + \rho X}{Y^2} \geq 0,$$

$$\frac{\partial f}{\partial \alpha} = \frac{1-\rho}{Y^2} \geq 0.$$

Thus, $r_{\mathcal{R}}(A)$ is monotonically increasing with $\rho, c, \alpha$ when $\rho \geq 1 - \frac{c}{1-\alpha}$. Moreover, the threshold $1 - \frac{c}{1-\alpha}$ is monotonically decreasing with $c, \alpha$. Thus, we conclude that $r_{\mathcal{R}}(A)$ is a monotonically increasing function of parameters $\rho, c, \alpha$.

**Analysis for $r_{\mathcal{S}}(A)$.** By a similar argument as for $r_{\mathcal{R}}(A)$, we can obtain the formulas of $r_{\mathcal{S}}(A)$ using Theorem 3.2 and the form of $t$. Note that $r_{\mathcal{S}}(A) = \rho r_{\mathcal{R}}(A)^2$. Thus, $r_{\mathcal{S}}(A)$ is monotonically increasing with $\rho, c, \alpha$.

Overall, we have completed the proof of the proposition. $\qquad\square$

# G   Additional details to Section 4

In this section, we first illustrate details for how to estimate perceived bias from JEE Advanced 2024 (Section G.1). Then we provide a robustness analysis for key findings in Section 4 by varying $\alpha$ and $c$ (Section G.2). Next, we give an illustrative example for the practical use of our model, including how to make interpretable predictions and policy interventions (Section G.3). Finally, we provide omitted details for alternative interventions in Section 4.

### G.1   Case study — estimating perceived bias from JEE Advanced 2024

To illustrate our framework in a high-stakes meritocratic setting, we calibrate the model using data from JEE Advanced 2024, the entrance examination for admission to the Indian Institutes of Technology (IITs). The gender-disaggregated statistics were published by the Government of India's Press Information Bureau [68]:

| Group | Candidates Appeared | Qualified |
|---|---|---|
| Male | 139,180 | 40,284 |
| Female | 41,020 | 7,964 |
| Total | 180,200 | 48,248 |

**Model calibration.** We define the disadvantaged group as *female candidates* and the advantaged group as *male candidates*. From the data:

- Proportion of disadvantaged applicants:

$$\alpha = \frac{41,020}{180,200} \approx 0.228$$

- Selection rate for the entire applicant pool:

$$c = \frac{48{,}248}{180{,}200} \approx 0.268$$

- Admit rate for each group:

$$\text{Female admit rate} = \frac{7{,}964}{41{,}020} \approx 0.194, \quad \text{Male admit rate} = \frac{40{,}284}{139{,}180} \approx 0.289$$

- Observed representation ratio:

$$r_{\text{obs}} = \frac{0.194}{0.289} \approx 0.671$$

**Solving for the bias parameter** $\rho$. Using the closed-form expression for the representation ratio in the uniform-valuations model:

$$r_R(\rho, c, \alpha) = 1 - \frac{(1-c)(1-\rho)}{\alpha - \alpha\rho + c\rho},$$

we plug in $r_R = 0.671$, $c = 0.268$, and $\alpha = 0.228$ to solve for $\rho$:

$$0.671 = 1 - \frac{(1 - 0.268)(1 - \rho)}{0.228 - 0.228\rho + 0.268\rho}.$$

Simplifying both sides:

$$0.329 = \frac{0.732(1 - \rho)}{0.228 + 0.04\rho},$$
$$0.329(0.228 + 0.04\rho) = 0.732(1 - \rho).$$

Compute both sides:

$$0.0749 + 0.01316\rho = 0.732 - 0.732\rho.$$

Bring all terms to one side:

$$0.74516\rho = 0.6571 \quad \Rightarrow \quad \rho \approx \frac{0.6571}{0.74516} \approx 0.882.$$

The inferred bias parameter is:

$$\rho \approx 0.882,$$

which reflects a perceived disadvantage for female candidates: they value qualification outcomes at roughly 88.2% of their male counterparts' valuation, consistent with the observed gender gap in qualification rates. This example demonstrates how our model can be applied to quantify bias in selection systems using real-world statistics.

### G.2 Robustness analysis for findings in Section 4

In this section, we assess whether our core conclusions in Section 4 depend on the specific parameter settings. To verify robustness, we conducted additional simulations varying $\alpha$ and $c$ beyond the default values. Below we summarize our findings:

**Metric robustness across group sizes.** We varied $\alpha$ from 0.5 to 0.3 (to represent smaller disadvantaged groups) and recalculated the representation ratio $r_R(A)$ and welfare ratio $r_S(A)$ across a grid of disparity levels ($\rho$) and selection rates ($c$); see Figure 14. The overall trends remain consistent: for example, $r_R(A) \leq 0.2$ still holds for $c = 0.1$ and $\rho \leq 0.85$, confirming that strategic behavior amplifies underrepresentation in highly selective settings.

**Robustness of intervention takeaways.** We varied $c$ from 0.228 (derived from the JEE Advanced data) to 0.1 and $\alpha$ from 0.268 to 0.5 and re-evaluated intervention strategies. Figure 15 plots optimal interventions for various threshold $\tau$. The overall trends remain consistent. For instance, in Figure 15(a), when $\tau \leq 0.87$, increasing access (raising $c$) remains more cost-effective. In contrast, when $\tau > 0.87$, improving group valuation (increasing $\rho$) becomes more impactful. This confirms that the recommendation to prioritize access vs. valuation interventions depending on the disparity level remains valid across reasonable choices of $\alpha$ and $c$.

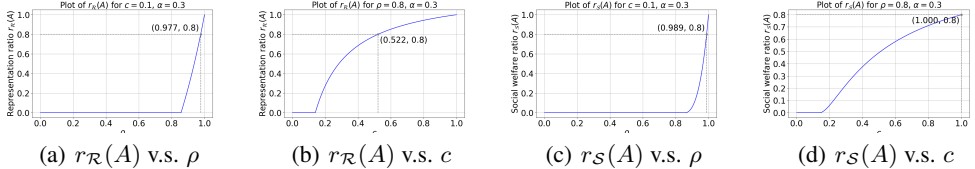

(a) $r_{\mathcal{R}}(A)$ v.s. $\rho$    (b) $r_{\mathcal{R}}(A)$ v.s. $c$    (c) $r_{\mathcal{S}}(A)$ v.s. $\rho$    (d) $r_{\mathcal{S}}(A)$ v.s. $c$

Figure 14: Plots of the representation ratio $r_{\mathcal{R}}(A)$ and the social welfare ratio $r_{\mathcal{S}}(A)$ as parameters $\rho$ and $c$ vary for Proposition 4.1, with default settings of $(\rho, c, \alpha) = (0.8, 0.1, 0.3)$. A dotted line in these plots indicates the threshold at which $r_{\mathcal{R}}(A) = 0.8$ or $r_{\mathcal{S}}(A) = 0.8$.

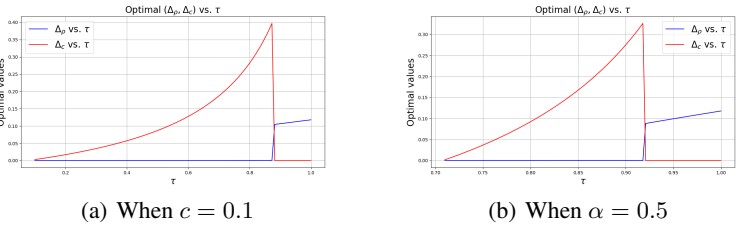

(a) When $c = 0.1$        (b) When $\alpha = 0.5$

Figure 15: Plot of optimal interventions $(\Delta_\rho, \Delta_c)$ for various $\tau$

## G.3 An illustrative example: interpreting and applying the model

We provide a concrete example to illustrate how our theoretical framework can be used to diagnose and compare policy interventions.

**Interpretable diagnostics.** Suppose a policymaker observes persistent underrepresentation of a disadvantaged group (e.g., female students) in a selective admissions process. Given data on the overall selection rate $c$, group size $\alpha$, and the observed representation ratio $r_{\mathcal{R}}(A)$, the policymaker can use our model to infer the implied *valuation gap* parameter $\rho$ (as shown in Section G.1). This parameter summarizes how much lower the disadvantaged group perceives the value of success relative to the advantaged group.

Although $\rho$ is not directly observable, its interpretation is transparent: it attributes behavioral disparities (such as lower effort investment) to structural differences in perceived incentives rather than to innate ability. In this way, the model provides a normative reading of observed disparities—as equilibrium responses to valuation asymmetries.

**Policy interventions.** Once the implied parameters are estimated, the policymaker can consider two classes of interventions:

- **Valuation-based interventions:** improving the perceived value of success (e.g., through mentorship programs or financial aid), which effectively increases $\rho$;
- **Access-based interventions:** expanding the number of available slots, thereby increasing $c$.

Our framework allows simulation of the effects of each intervention on representation, welfare, and efficiency, enabling counterfactual comparisons under a fixed behavioral model. For example, when $\rho$ is low, expanding access may yield larger gains in representation, while when $\rho$ is already high, improving valuation can be more cost-effective.

**Implementation challenges.** Estimating parameters such as $\rho$ empirically is nontrivial and remains an open direction. It would require auxiliary data sources (e.g., surveys, longitudinal effort–outcome data) or structural assumptions about the effort cost function. Nonetheless, once such estimates are available, our framework provides a transparent scaffold for interpreting behavioral disparities and evaluating the relative effectiveness of competing policy interventions.

## G.4 Details for alternative interventions

Below we provide more detailed theoretical analysis for alternative potential interventions discussed in Section 4.

**Introducing preference heterogeneity.** Recall that the institution applies group-specific merit mappings of the form: for group $G_\ell$ ($\ell = 1, 2$) and for score $s$, $m_\ell(s) = x_\ell \cdot s + y_\ell$ for group-specific parameters $x_\ell, y_\ell \geq 0$. We have the following generalized theorem of Theorem 3.1

**Theorem G.1 (Generalization of Theorem 3.1: Introducing preference heterogeneity).** *Let $\alpha, c \in (0, 1)$. For $\ell = 1, 2$, let $p_\ell$ be a density supported on a domain $\Omega_\ell \subseteq \mathbb{R}_{\geq 0}$. Let $p_a$ be a density supported on a domain $\Omega_a \subseteq \mathbb{R}_{\geq 0}$. For $\ell = 1, 2$, let $m_\ell$ be a merit function defined above. For $\ell = 1, 2$, let $F_\ell$ be a cumulative density function (CDF) of the sum of valuation and initial ability such that for any $\zeta \in \mathbb{R}_{\geq 0}$, $F_\ell(\zeta) = \Pr_{v \sim p_\ell, a \sim p_a} [x_\ell v + y_\ell + a \leq \zeta]$. Suppose $(x_1 \Omega_1 + y_1) \cup (x_2 \Omega_2 + y_2)) + \Omega_a$ is connected and densities $p_1, p_2, p_a$ are positive at any point of their own domains. Let $t$ be the unique solution to the equation*

$$(1 - \alpha)F_1(\zeta) + \alpha F_2(\zeta) = 1 - c.$$

*Then the threshold function of the NE policy defined in Eq. (3) extends to be: for $\ell = 1, 2$,*

$$s_\ell(v, a) = 0 \text{ if } v + a < \frac{t - y_\ell}{x_\ell}, \text{ and } s_\ell(v, a) = \max\left\{\frac{t - y_\ell}{x_\ell} - a, 0\right\} \text{ if } v + a \geq \frac{t - y_\ell}{x_\ell}.$$

*Moreover, the threshold $\frac{t - y_2}{x_2}$ for $G_2$ is monotonically decreasing with $x_2, y_2$.*

**Proof:** The proof for $s_\ell$ is almost identical to that of Theorem 3.1. We only need to note that for any agent $i \in G_\ell$ ($\ell = 1, 2$) with valuation-ability pair $(v_i, a_i) \in \Omega_\ell \times \Omega_a$, if its valuation $v_i \geq \frac{t - y_\ell}{x_\ell} - a_i$, then its merit must be

$$m_\ell(s_\ell(v_i, a_i)) = x_\ell \cdot \left(a_i + \max\left\{\frac{t - y_\ell}{x_\ell} - a_i, 0\right\}\right) + y_\ell \geq t,$$

which is within the top $c$-fraction and makes the agent get selected.

Regarding the monotonicity of $\frac{t - y_2}{x_2}$, note that as $x_2$ or $y_2$ increases, $F_\ell(\zeta)$ decreases. Then the solution $t$ must increase, resulting in a higher threshold $\frac{t - y_1}{x_1}$ for group $G_1$. This reduces the fraction of agents in $G_1$ to get selected, and consequently, increases the fraction of agents in $G_2$ to get selected. Then the threshold $\frac{t - y_2}{x_2}$ must decrease, which completes the proof. $\square$

Note that when $x_1 = x_2 = 1$ and $y_1 = y_2 = 0$, this theorem is exactly Theorem 3.1, and hence, is a generalization. This theorem implies that by increasing $x_2, y_2$, more agents in $G_2$ are willing to put in efforts due to lower valuation threshold $\frac{t - y_2}{x_2}$. This supports the properties discussed in Section 4.

**Setting group-specific selection rates.** Assume that the institution selects a $c$-fraction of agents from $G_1$ and $G_2$ independently. The model decomposes into two independent within-group contests, each with its own Nash equilibrium.

For the disadvantaged group $G_2$, let $F_2$ denote the CDF of its combined signal $v + a$. The equilibrium threshold $t_2$ under group-specific capacity $c$ is the unique solution to:

$$F_2(t_2) = 1 - c.$$

In contrast, under a uniform selection rate $c$ applied to the full population (original two-group contest), the common threshold $t$ solves:

$$(1 - \alpha)F_1(t) + \alpha F_2(t) = 1 - c.$$

Since $G_2$ has lower valuations by assumption, we typically have $F_2(\zeta) \geq F_1(\zeta)$ for all $\zeta$, which implies $t_2 < t$. That is, the disadvantaged group faces a lower selection bar under group-specific quotas.

As a result, agents in $G_2$ exert more effort on average under per-group capacities compared to the uniform-$c$ case. This is because effort is increasing in the probability of selection, which improves when the threshold is lowered.

