# OpenReview forum: "Strategic Costs of Perceived Bias in Fair Selection"
_NeurIPS.cc/2025/Conference — NeurIPS 2025 spotlight_

### Official Review · Reviewer_rBtT · 2025-06-18

**Clarity:** 3
**Significance:** 3
**Originality:** 3
**Rating:** 5
**Confidence:** 3

**Summary:**

This paper investigates strategic asymmetries arising from valuation differences in settings where group sizes are known and fixed, such as admissions and hiring. The authors propose a formal framework to model these strategic interactions, aiming to provide interpretable predictions and support data-grounded policy interventions. The paper presents a theoretical analysis, including definitions, theorems, and discussions of metrics.

**Questions:**

1. Could you elaborate on the specific aspects of your framework that enable "interpretable predictions" and "data-grounded policy interventions"? Perhaps provide a brief, concrete example of how a policy maker could use your framework's outputs.
2. Could you briefly summarise in the main text (e.g., in the introduction or related work section) the precise novel contribution of your work especially in contrast to the most closely related prior research?

**Ethical Concerns:**

["NO or VERY MINOR ethics concerns only"]

**Final Justification:**

I am confident with my positive score and appreciate the detailed rebuttal from the reviewers. I am confident that with the additional details from the rebuttal, the paper will be improved and should make a valuable contribution.

**Limitations:**

Yes

**Quality:**

3

**Strengths And Weaknesses:**

### Strengths
* S1 – The paper addresses a highly relevant and important problem concerning strategic behavior in critical application areas like admissions and hiring. The proposition that valuation differences lead to strategic asymmetries in fixed-group settings is compelling and aligns with real-world observations. I believe this is a very important and underresearched area of algorithmic fairness literature, focus on formal neutrality and static metrics leave much underexplored in fair selection. By focusing on valuation gaps, the work complements the dominant literature on classifier bias and post-hoc fairness metrics.

* S2 – Clear Presentation of Theoretical Framework: While the mathematical depth is significant, the paper is generally well-written, well-cited, and the notation is clear. The definitions and theorems are set out lucidly, making the overall structure and conceptual flow of the theoretical framework understandable, even without deep expertise in game theory. The metrics introduced in Section 4 are clearly defined and appear appropriate for the problem context, drawing on established concepts in the relevant literature.

### Weaknesses
* W1 – While the authors claim their work is the first to study strategic asymmetries arising from valuation differences in settings with known and fixed group sizes (a compelling claim if true), the main body of the paper does not sufficiently articulate how its specific contributions differentiate it from existing literature. Although Appendix A provides detailed related work, a concise explanation of the paper's novelty should be integrated into the main text to immediately clarify its unique contribution for the reader. Despite reviewing supplementary material it still isn't perfectly clear to me what exactly is unique about your method.

* W2 – The paper states that its "framework enables interpretable predictions and supports data-grounded policy interventions." However, it does not explicitly elaborate on how this framework achieves these goals. While concrete guidance on parameter estimation (e.g. recovering ρ from survey or wage data) is deferred to future work, more detailed discussion or concrete examples illustrating the path from theoretical insights to practical policy implications would strengthen this claim.

* W3 - Real-world fit is uncertain. The theory assumes very large applicant pools and that every person always chooses the effort level that maximises their payoff. In practice, many selection rounds involve only a few dozen or a few hundred candidates, and people often make decisions based on limited information, habits, or simple rules of thumb. Because of this, the neat patterns the model predicts might look quite different when applied to smaller, messier real-world settings.

---

> ### Author Rebuttal · Authors · 2025-07-30
>
> We thank you for your thoughtful, detailed, and insightful feedback. In response, we added an illustrative example to elaborate on the applicability of our framework and addressed your specific questions and a comment (W3).
>
> > Could you elaborate on the specific aspects of your framework that enable "interpretable predictions" and "data-grounded policy interventions"? Perhaps provide a brief, concrete example of how a policy maker could use your framework's outputs.
>
> Thank you for this insightful question. We agree that the connection between our theoretical framework and its practical applications warrants clearer articulation. Below, we offer a concrete example to illustrate how our model supports interpretable predictions and can inform data-grounded interventions—while also noting what is required to operationalize it in practice.
>
> **Interpretable predictions:** Suppose a policymaker observes persistent underrepresentation of a disadvantaged group (e.g., female students) in a selective admissions process. Given data on the overall selection rate ($c$), group size ($\alpha$), and the observed representation ratio $r_{\mathcal{R}}(A)$, the policymaker can use our model (e.g., Lines 365–372) to infer an implied valuation gap parameter ($\rho$). This parameter summarizes how much lower the disadvantaged group perceives the value of success relative to the advantaged group.
>
> Although $\rho$ is not directly observable, its interpretation is transparent: it explains behavioral disparities (e.g., lower effort investment) as arising from structural differences in perceived incentives. This enables a normative interpretation of observed disparities—not simply as gaps in ability, but as strategic responses to valuation asymmetries.
>
> **Policy interventions:** With this diagnosis, the policymaker can consider and compare two classes of interventions:
>
> - **Valuation-based interventions**: improving perceived value of success (e.g., through financial aid, mentorship), effectively increasing $\rho$.
> - **Access-based interventions**: expanding the number of available slots, thereby increasing $c$.
>
> Our framework (cf. Figure 3 and Section 8) allows one to simulate the effects of each intervention on representation and welfare, and to compare their relative cost-effectiveness. For example, when $\rho$ is low, expanding access may be more impactful; when $\rho$ is already high, improving valuation may yield better gains. This enables counterfactual comparisons under a fixed behavioral model.
>
> We acknowledge that empirically estimating parameters—especially $\rho$—is challenging and remains an open direction. It would require auxiliary data sources (e.g., surveys, longitudinal effort-outcome observations) or structural assumptions on effort cost. Nonetheless, our framework offers a transparent scaffold: once estimates are in place, it enables interpretable diagnostics and structured policy evaluation.
>
> We will include this example and discussion in Appendix G of the final version.
>
> > Could you briefly summarise in the main text (e.g., in the introduction or related work section) the precise novel contribution of your work, especially in contrast to the most closely related prior research?
>
> Thank you for the suggestion. We agree that a concise summary of our novel contributions relative to prior work should appear in the main text and will incorporate this in Section 1.
>
> **Our key contribution** is to show how group-level disparities can arise endogenously from **valuation asymmetries**, even when ability is identical, information is unbiased, and selection criteria are uniform. This provides a new structural explanation for representation gaps in competitive settings with fixed group sizes.
>
> **Compared to prior two-group contest models**:
> - [1] studies a two-player contest with heterogeneous valuations; we generalize this to many agents and multiple selection spots.
> - [24,27] analyze two-group contests but assume symmetric valuation distributions; we are the first to study equilibrium behavior under **asymmetric valuation distributions** across groups in such settings.
>
> **Compared to prior work on statistical discrimination**:
> - Classical models [4,46] generate disparities via **biased beliefs** or **informational asymmetries** from the institution side, even when true abilities are equal.
> - In contrast, our model assumes **perfect information** and **equal evaluation** for the institution, but introduces valuation asymmetries—different returns to the same success—that alone generate group disparities in effort and selection.
>
> We will add this comparison to the introduction to clarify the novelty and position our work relative to the most closely related literature.
>
>
> > W3 - Real-world fit is uncertain. The theory assumes very large applicant pools and that every person always chooses the effort level that maximises their payoff. In practice, many selection rounds involve only a few dozen or a few hundred candidates, and people often make decisions based on limited information, habits, or simple rules of thumb. Because of this, the neat patterns the model predicts might look quite different when applied to smaller, messier real-world settings.
>
>
> Thank you for raising this concern about the realism of the infinite-agent assumption. These assumptions are common in theoretical work for tractability, but as you rightly point out, they may not fully capture the messiness of real-world decision-making. However, in addition to using the large-$n$ limit to characterize Nash equilibrium policies, our results show that the infinite-population approximation is **quantitatively accurate** even at moderate population sizes.
>
> As shown in Section 3 (Lines 229–236) and Appendix C.2, the value of the finite-$n$ Nash equilibrium policy is provably $O(\sqrt{\log n / n})$-close to that of the infinite-population policy. Furthermore, Figure 1 demonstrates that for population sizes as small as $n = 600$, the observed equilibrium policies closely track the infinite limit across all selection rates $c$. This supports the use of the infinite-$n$ model as a practical and robust approximation for many real-world settings involving hundreds of candidates.
>
> While real-world decision-making can of course involve noise, habits, or bounded rationality, our model aims to capture the **incentive-aligned baseline** under strategic behavior. We believe this remains a highly informative reference point, even when actual behavior deviates from full rationality.
>
> We will clarify this justification in the final version.
>
> Once again, we thank you for your detailed feedback and for helping us improve the quality and impact of our work.

---

> > ### Comment · Reviewer_rBtT · 2025-08-02
> >
> > Thank you for your detailed responses. Your discussion is helpful and I believe inclusion of these points will help clarify this aspects of the paper.

---

### Official Review · Reviewer_uD1e · 2025-06-21

**Clarity:** 4
**Significance:** 3
**Originality:** 3
**Rating:** 5
**Confidence:** 2

**Summary:**

This paper studies a contest with two demographic groups. The groups differ only in their perception of how valuable winning the contest is. This is interpreted as the disadvantaged group having a lower valuation of the win, and how that impacts different performance criteria, including the representation ratio, the welfare ratio, and the institution’s revenue.

**Questions:**

1. How is the model or its results reflecting on the underlying “structural” and societal issues?

2. The paper considers the representation ratio to be the min of the representation ratios of the groups. Isn’t it always the case that the disadvantaged group ends up being underrepresented in the outcome? And if not, why not?

3. It seems that the model can be used to answer an additional question: if the institution sets per-group capacities (i.e., a group specific selection rate c), then how would the effort the disadvantaged group exert compare to the one when they are under a uniform c rate? (even keeping the valuations asymmetric). Is this correct? It would be interesting to know more about this aspect as well.

4. Section 8, line 299, mentions some default values when some parameters are not being varied. It is assumed then that the corresponding results sensitive to these choices? For instance, is the takeaway at the end of the section (“This suggests that expanding access is more impactful under high disparity, while improving group valuation is better when gaps are narrower.” ) independent of the \alpha and c selected based on the JEE Advanced data?

5. Typos: line 127, a missing comma in the definition of the representation ratio. Line 247: “is to that” --> “is that”

**Ethical Concerns:**

["NO or VERY MINOR ethics concerns only"]

**Final Justification:**

The rebuttal has been useful in providing clarifications to my questions, especially the additional discussions and results provided around group-specific selection rates and sensitivity analysis for the experiments. I believe this is a technically solid paper with interesting insights, and accordingly maintain my score at Accept.

**Limitations:**

Yes

**Quality:**

3

**Strengths And Weaknesses:**

**Strengths**:
- The paper provides a new perspective on the issue of disparities in competitive selection processes (like school admissions), by considering asymmetric valuations. It is also different from the literature on strategic classification because of competition between agents, and therefore complements that line of work well, too.
- The paper is quite well-written. The intuitive walkthrough of the main theorem is clear and informative. The sensitivity analysis corollaries and experiments are well-thought.
- Several of the provided insights from the theoretical analysis have the potential to be informative for questions of contest/institution design and interventions to encourage underrepresented minorities to consider exerting effort for competitive opportunities.

**Weaknesses**:
- Some of the statements in the introduction and their alignment with the results are unclear. Specifically, the paper does open by noting that structural barriers are a key driver of disparity in representation rates during competitive selection processes, including “unequal access to resources that enhance merit”. The closest element of the model to these, to me, is p_a, the distribution of initial ability. However, this is taken to be group-independent throughout the paper. Beyond this, it is not clear if there are elements in the model that are capturing the “structural” rather than “behavioral” aspects. If no such "structural" elements exist in the model, then that makes it unclear why “This paper reconciles structural and behavioral explanations”, as stated on top of page 2.

- Some additional requests for clarification and questions are added below.

---

> ### Author Rebuttal · Authors · 2025-07-30
>
> We thank you for your detailed and encouraging review. We are especially grateful for your recognition of our framework’s real-world applicability. Below we address your specific questions.
>
> >How is the model or its results reflecting on the underlying “structural” and societal issues?
>
> Thank you for this thoughtful question. We appreciate the opportunity to clarify how our model captures the distinction between structural and behavioral explanations.
>
> In our framework, structural disparities are represented by group-dependent valuation distributions—that is, differences between $p_1(v)$ and $p_2(v)$ that reflect external and systemic inequalities. These may stem from unequal access to opportunity, differences in marginal returns, labor market discrimination, or broader societal narratives about value.
>
> By contrast, behavioral disparities emerge endogenously in our model: given these structural valuation gaps, agents rationally adjust their effort levels. Even when ability distributions $p_a$ are identical across groups, the perceived return to effort is lower for disadvantaged groups, leading to systematically lower effort—and thus to disparities in selection rates.
>
> In this way, our model illustrates a concrete feedback from structural conditions to behavioral outcomes. This is what we mean by reconciling the two perspectives: our analysis shows how externally imposed valuation asymmetries can, even under unbiased and group-blind selection, result in behavioral underrepresentation.
>
> We will clarify this interpretation in Section 1 of the final version.
>
>
> > The paper considers the representation ratio to be the min of the representation ratios of the groups. Isn’t it always the case that the disadvantaged group ends up being underrepresented in the outcome? And if not, why not?
>
>
> Thank you for the question. You are correct that in the infinite population limit—our main analytical regime—the disadvantaged group is always underrepresented due to lower valuations, and the representation ratio reflects this.
>
> In the finite-$n$ case, however, randomness can occasionally lead to equal or even higher representation for the disadvantaged group, though this becomes vanishingly rare as $n \to \infty$.
>
> While the qualitative direction of underrepresentation may not be surprising, our contribution lies in quantifying how strategic behavior amplifies these disparities under varying selection pressures and valuation gaps. This allows us to make precise, interpretable predictions and compare interventions within a unified framework.
>
> We will clarify this in the final version.
>
>
> > It seems that the model can be used to answer an additional question: if the institution sets per-group capacities (i.e., a group-specific selection rate c), then how would the effort the disadvantaged group exert compare to the one when they are under a uniform c rate? (even keeping the valuations asymmetric). Is this correct? It would be interesting to know more about this aspect as well.
>
> Thank you for the insightful question. Yes, your interpretation is correct: if the institution sets **group-specific selection rates**—i.e., separate capacity constraints for each group—then the model decomposes into two **independent within-group contests**, each with its own Nash equilibrium.
>
> For the disadvantaged group $G_2$, let $F_2$ denote the CDF of its combined signal $v+a$. The equilibrium threshold $t_2$ under group-specific capacity $c$ is the unique solution to:
> $$
> F_2(t_2) = 1 - c.
> $$
>
> In contrast, under a **uniform** selection rate $c$ applied to the full population, the common threshold $t$ solves:
> $$
> (1-\alpha) F_1(t) + \alpha F_2(t) = 1 - c.
> $$
>
> Since $G_2$ has lower valuations by assumption, we typically have $F_2(\zeta) \geq F_1(\zeta)$ for all $\zeta$, which implies $t_2 < t$. That is, the disadvantaged group faces a **lower selection bar** under group-specific quotas.
>
> As a result, agents in $G_2$ exert **more effort** on average under per-group capacities compared to the uniform-$c$ case. This is because effort is increasing in the probability of selection, which improves when the threshold is lowered.
>
> We agree that this highlights an interesting direction for intervention, and we will include a remark about this in the final version.
>
>
> > Section 8, line 299, mentions some default values when some parameters are not being varied. It is assumed then that the corresponding results sensitive to these choices? For instance, is the takeaway at the end of the section (“This suggests that expanding access is more impactful under high disparity, while improving group valuation is better when gaps are narrower.” ) independent of the \alpha and c selected based on the JEE Advanced data?
>
> Thank you for pointing this out. We agree that it is important to assess whether our core conclusions depend on the specific parameter settings (e.g., $\alpha$, $c$) derived from the JEE Advanced data.
>
> To verify robustness, we conducted additional simulations varying $\alpha$ and $c$ beyond the default values. Below we summarize our findings:
>
> - **Metric robustness across group sizes:** We varied $\alpha$ from 0.5 to 0.3 (to represent smaller disadvantaged groups) and recalculated the representation ratio $r_{\mathcal{R}}(A)$ and welfare ratio $r_{\mathcal{S}}(A)$ across a grid of disparity levels ($\rho$) and selection rates ($c$). The overall trends remain consistent: for example, $r_{\mathcal{R}}(A)\leq 0.2$ still holds for $c=0.1$ and $\rho \leq 0.85$, confirming that strategic behavior amplifies underrepresentation in highly selective settings (cf. Lines 314–317).
> - **Robustness of intervention takeaways:** We varied $c$ from 0.228 to 0.1 and $\alpha$ from 0.268 to 0.5 and re-evaluated intervention strategies. The takeaway remains robust:
>   - When $\tau \leq 0.87$, increasing access (raising $c$) remains more cost-effective.
>   - When $\tau > 0.87$, improving group valuation (increasing $\rho$) becomes more impactful.
>   This confirms that the recommendation to prioritize access vs. valuation interventions depending on the disparity level remains valid across reasonable choices of $\alpha$ and $c$.
>
> We will include these robustness checks and associated plots in Appendix G of the final version.
>
> We thank you again for your thoughtful engagement, which has helped us clarify and strengthen the contributions of our work.

---

> > ### Comment · Reviewer_uD1e · 2025-08-02
> >
> > I appreciate the authors responses and clarifications, especially the additional discussions and results provided around group-specific selection rates and sensitivity analysis for the experiments.

---

### Official Review · Reviewer_H86j · 2025-07-01

**Clarity:** 3
**Significance:** 3
**Originality:** 3
**Rating:** 5
**Confidence:** 4

**Summary:**

To examine how selection rates can diverge across groups even under group-blind selection criteria, the authors model a contest setting in which agents rationally choose effort based on their expected benefit from selection. In this setup, selection depends on a combination of ability and effort, but only effort is chosen strategically. When the expected benefit of selection is lower for one group—modeled as a compressed distribution of valuation (captured by the parameter rho)—members of that group exert less effort on average, resulting in lower selection rates. The authors characterize equilibrium behavior in this setting and analyze how group disparities in selection, welfare, and average “quality” of selected candidates depend on key parameters: the degree of valuation compression (rho), the selectivity of the institution (cc), and the share of disadvantaged candidates in the population (alpha).

**Questions:**

1.	Clarify the meaning of the feedback loop- the abstract and introduction refer to a behavioral “feedback loop” as a core motivation, but the model appears static: group-based disparities arise from fixed, exogenous differences in valuations. Could you clarify whether the feedback loop is meant as part of the formal model or just the broader motivation? If the latter, it would help to revise the language to avoid suggesting a dynamic or endogenous mechanism. extensions (e.g., beliefs updating over time or feedback via algorithmic guidance tools).
2.	position the paper relative to the statistical discrimination literature- the model closely resembles ideas from the economics literature on statistical discrimination. These works emphasize feedback between group behavior and institutional inference. Your model differs in that the institution is passive and group-blind, but that distinction is not clearly drawn. I recommend engaging with this literature more explicitly to clarify what is novel about your setup and what simplifying assumptions are being made.
3.	Better motivate the setup and assumptions- for example, the assumption that one group derives less value from selection (modeled via a compressed valuation distribution) is plausible in some domains, but less so in others. I would encourage you to (a) clarify when this assumption holds, (b) discuss when it might be reversed, and (c) consider how the model might be adapted to allow for heterogeneity in valuations or outside options.
4.	Ideally, you would be able to say something about how the analysis would change if you had a strategic firm, even if you do not formally solve the model.

**Ethical Concerns:**

["NO or VERY MINOR ethics concerns only"]

**Final Justification:**

I appreciate the author comments and continue to believe this is a strong paper that should be accepted for publication.

**Limitations:**

yes

**Quality:**

3

**Strengths And Weaknesses:**

Strengths-

-	The paper introduces an elegant and intuitive framework to capture how group-based differences in the perceived value of success (landing a job, receiving a loan, etc.) can affect effort and, in turn, selection outcomes, even when the selection rule is group-blind. While this mechanism will feel familiar to economists as a classic asymmetric-contest setup, it offers a valuable perspective for a CS/ML audience, where fairness analyses often focus narrowly on constraints and selection rules. The model highlights that disparities can arise endogenously when valuations differ across groups.
-	The Nash equilibrium solution, where effort is exerted only by individuals whose valuation plus ability exceed a common threshold, is analytically clean and well suited to quantifying disparities. The resulting inequalities in selection rates and payoffs are both interpretable and empirically relevant.
-	The authors clearly explain the role of key parameters in the model (such as valuation bias rho, selection rate c, and group proportion alpha) and how changes in these parameters affect the fairness metrics.
-	The paper is well-motivated by real-world phenomena, such as wage gaps and unequal returns to opportunity, that suggest group-based differences in the perceived value of “winning” are empirically plausible and normatively important.
-	Overall, the paper brings a less-studied dimension of fairness into focus: how group disparities in incentives or perceived benefit can drive unequal outcomes even in the absence of explicit discrimination or biased selection.

Weaknesses

-	To some extent, the paper’s strength is also its weakness: while the analysis is elegant and clean, it may be too stylized to capture important real-world dynamics. In particular, it abstracts away from richer feedback mechanisms and strategic institutional behavior that are well-studied in the economics literature on statistical discrimination.
-	For example, the paper repeatedly invokes the idea of a “feedback loop,” but it is unclear what that loop entails. The model is static, and group disparities arise entirely from exogenous differences in valuations. That is, differences in selection rates do not feed back into future valuations, institutional behavior, or perceptions, making the use of “feedback” more rhetorical than actually part of the model.
-	Surprisingly, the paper does not engage with the economics literature on statistical discrimination, despite addressing similar questions. While the authors may not come from an economics background, their model aligns more naturally with this tradition than with many of the social science works cited. Foundational work such as Phelps’s “The Statistical Theory of Racism and Sexism” (1972), Arrow’s “The Theory of Discrimination” (1973), Coate and Loury’s “Will Affirmative-Action Policies Eliminate Negative Stereotypes?” (1993), and Craig and Fryer’s “Complementary Bias: A Model of Two-Sided Statistical Discrimination” (2013) examine settings where disparities emerge endogenously from a feedback loop between agent behavior and institutional inference. These models typically feature rational expectations equilibria in which institutions interpret noisy signals differently across groups, reinforcing group-level behavioral patterns.
-	More broadly, engaging with this literature would not only help position the contribution more clearly but also point to natural extensions. For instance, the current model assumes that institutional payoff is a direct function of observable scores. However, in many settings, especially in hiring or lending, payoff depends on unobserved or latent characteristics like ability (or creditworthiness), and effort merely increases the signal. Modeling the institution as one that tries to infer ability from scores (and does so differently for each group) would allow the paper to explore how strategic inference might perpetuate or mitigate disparities. It would also bring the institutional actor more clearly into focus, rather than treating it as a passive selector.
-	Another potential limitation is the assumption that the disadvantaged group has lower valuations. While this is plausible in some setting, for example, where minorities being treated poorly at work (created a lower value to the job, assuming job climate is part of the evaluation), there are others (like lending) where the opposite may be true: marginalized groups might derive higher value from access to opportunity (loans or jobs), due to lack of outside options or greater marginal returns. It would be helpful for the authors to clarify when this compressed valuation assumption is realistic and when it might not be.
-	Relatedly, the model assumes that differences in valuation are problematic, but it is worth considering when they may reflect legitimate preference heterogeneity. For example, if one group places more value on work-life balance and therefore exerts less effort for high-paying jobs, should we interpret the resulting selection rate disparities as unfair? The paper might benefit from more discussion of when group differences in effort due to valuation gaps are normatively troubling.
-	The assumption of a threshold strategy—where agents either exert just enough effort to be selected or none at all—also limits applicability. In many real-world contests (like job applications, school admissions), individuals may over-prepare or exert continuous effort for marginal improvements, especially when downstream rewards like salary or financial aid are continuous rather than binary. This opens the door to modeling richer selection environments, such as multi-stage contests or settings with heterogeneous reward structures.
-	Overall, while these issues do not undermine the core contribution, the paper would benefit from a more nuanced discussion of its simplifying assumptions, clearer positioning relative to existing models of discrimination and effort, and acknowledgment of avenues for future work.

---

> ### Author Rebuttal · Authors · 2025-07-30
>
> Thank you for your thoughtful and constructive feedback. We are grateful for your recognition of the paper’s modeling contributions, and we appreciate your detailed suggestions for further improvement, particularly regarding assumptions and connections to prior work. Below we address your questions:
>
> > 1. Clarify the meaning of the feedback loop? ...
>
> Thank you for raising this point. To clarify: the “feedback loop” is not part of the formal model, but rather serves as a motivating narrative for why disparities in perceived value can matter. The model itself is static—group-based disparities arise from fixed, exogenous differences in perceived post-selection value, with no endogenous belief updating or institutional dynamics.
>
> What we intended to highlight was the following phenomenon: when one group expects lower returns (e.g., due to underrepresentation or systemic inequities), its members rationally exert less effort, which then leads to continued underrepresentation—even under group-blind selection.
>
> We agree that the phrase “feedback loop” may suggest dynamic or time-evolving mechanisms that our model does not capture; we will remove it from the abstract and introduction and explicitly state that the model is static.
>
> >2. position the paper relative to the statistical discrimination literature ...
>
> Thank you for the references on statistical discrimination. Below we give a brief comparison. We will add a dedicated subsection in the final version to clarify connections and distinctions.
>
> - [Phelps 1972] models a profit-maximizing employer who faces noisy signals of productivity and rationally uses group-level statistics, leading to persistent wage gaps even with equal underlying abilities.
> - [Arrow 1973] emphasizes how the cost of individualized assessment incentivizes reliance on priors, which can become self-fulfilling and reinforce structural inequality.
> - [Coate and Loury 1993] show that pessimistic beliefs about a group’s productivity can result in tougher standards, reduced investment incentives, and discriminatory equilibria.
> - [Craig and Fryer 2013] extend this to two-sided settings where firms and workers both act on noisy beliefs, reinforcing low-investment, low-opportunity equilibria.
>
> A key distinction, as we understand it, is that classical models of statistical discrimination typically generate disparities through **imperfect** and **group-dependent** beliefs about identical underlying abilities. In contrast, our framework allows **perfect, unbiased** information at the institutional level and identical selection criteria for all candidates. We focus instead on **valuation asymmetries**—that is, differences in the *perceived benefit* of success across groups—and show that these differences alone can lead to disparities in effort and representation, even under meritocratic selection.
>
> To our knowledge, this offers a complementary behavioral mechanism to those studied in the statistical discrimination literature—one that centers the agent's internal valuation rather than institutional inference.
>
> > 3. Better motivate the setup and assumptions ... encourage you to a) clarify when this assumption holds, b) discuss when it might be reversed, and c) consider how the model might be adapted to allow for heterogeneity in valuations or outside options.
>
> Thank you for the suggestions. To clarify our setting: the core model assumes no outside option, group-blind selection based on a common score, and a binary reward for selection. While the model can be generalized to include outside options or heterogeneous selection (see more details below), even in this core form, valuation asymmetry can arise; not from individual preferences, but from group-dependent beliefs about the value derived post-selection. These beliefs may be shaped by structural inequality, lived experience, or algorithmic signals.
>
> We now address each part of your comment:
>
> **a) When lower valuations are plausible:** Our model applies to settings in which disadvantaged groups perceive lower returns to selection due to structural or informational asymmetries. As described in the introduction (lines 39–54), wage gaps persist across gender and race even after controlling for qualifications, and algorithmic systems often return lower-valued recommendations for marginalized groups. Such disparities shape expectations about the benefits of success and can depress effort before selection. These are the environments our model aims to capture.
>
> **b) When the asymmetry might reverse:** Indeed, in some settings (particularly in access-to-opportunity domains such as credit, housing, or education) disadvantaged groups may face higher marginal benefits from selection due to fewer outside options or steeper returns. Our model accommodates such cases by reversing which group has the compressed valuation distribution. However, our motivating focus and analysis remains on the scenario of lower perceived returns among disadvantaged groups.
>
> **c) Preference heterogeneity and outside options:** While our core model assumes homogeneous evaluations and no outside option, the framework can be extended to derive insights about this setting that institutions can strategically reduce the disparities caused by heterogeneity in valuations. For example, consider the setting where institutions may apply group-specific merit mappings of the form: for group $G_\ell$ ($\ell = 1,2$) and for score $s$,
> $$m_\ell(s) = x_\ell \cdot s + y_\ell$$
> for group-specific parameters $x_\ell, y_\ell \geq 0$. One can interpret $x_\ell$ as a *scaling factor* or "handicap", and $y_\ell$ as an *offset* or "headstart" (see Appendix A and [75]).
>
> - **Differences to homogeneous case:** For the homogeneous case, we show that 1) both representation ratio $r_{\mathrm{R}}(A)$ ($A$ is the NE policy defined in Theorem 3.1) and the social welfare ratio $r_{\mathcal{S}}(A)$ are monotonically increasing functions of bias parameter $\rho$, selectivity $c$ and disadvantaged fraction $\alpha$; 2) in highly selective environments (e.g., $c=0.1$), a slight bias ($\rho\leq 0.85$) leads to high disparities ($r_{\mathrm{R}}(A)\leq 0.2$ and $r_{\mathrm{S}}(A)\leq 0.2$); 3) Increasing $\rho$ and $c$ are two intervention ways to reduce disparities.
> By the above extension, the findings for the homogeneous case changes to be: 1) $r_{\mathrm{R}}(A)$ and $r_{\mathcal{S}}(A)$ are not only increasing functions of $\rho,c,\alpha$, but also increasing functions of $x_2,y_2$ and decreasing functions of $x_1,y_1$; 2) By selecting merit parameters with $x_2 > x_1$ and $y_2 > y_1$, disparities can remain low (e.g., $r_{\mathrm{R}}(A)\geq 0.8$ and $r_{\mathrm{S}}(A)\geq 0.8$) even in highly selective environments; 3) Besides increasing $\rho$ and $c$, increasing the value of $x_2,y_2$ is the third intervention way to reduce disparities.
>
> - The intuition behind these changes is that agents from $G_2$ are easier to select with the same score $s$ compared to the homogeneous case due to boosted merits, increasing their investments and resulting in higher selection from $G_2$. This intuition can also be verified mathematically since our techniques extend naturally to this setting and yield a NE characterization similar to Theorem 3.1:
>
> **Theorem.**  Let $\alpha$ be the fraction of $G_2$ in the population and $c$ is the selectivity. For $\ell = 1,2$, let $p_\ell$ be a valuation density of $G_\ell$ and $m_\ell$ be a merit function defined above. Then there exists a unique $t$ such that the group-specific effort policies are as follows: for an agent from group $G_\ell$ and any valuation $v$,
> if $v < \frac{t - y_\ell}{x_\ell}$, then $s_\ell(v) = 0$; otherwise, $s_\ell(v) = \max \left( \frac{t - y_\ell}{x_\ell}, 0\right)$.
>
> Moreover, the threshold $\frac{t - y_2}{x_2}$ for valuation $v$ is monotonically decreasing with $x_2,y_2$.
> This theorem implies that by increasing $x_2,y_2$, more agents in $G_2$ are willing to put in efforts due to a lower valuation threshold $\frac{t - y_2}{x_2}$, consistent with the above intuition.
>
> Additionally, our framework can incorporate **outside options**.  Let each agent in group $G_\ell$ receive a reservation payoff $\lambda_\ell \ge 0$ if she is *not* selected.  Because this payoff is earned only when losing, a higher $\lambda_\ell$ lowers the marginal benefit of exerting effort, working in the *opposite* direction of the merit parameters $x_\ell$ and $y_\ell$. Hence, increasing $\lambda_1$ (the outside option for the advantaged group) depresses their effort incentives and can be used as an additional intervention for narrowing representation and welfare gaps.
>
> We will clarify points (a) and (b) in the main body and add the insights along with accompanying results mentioned for \(c) in the appendix.
>
> > 4. ... strategic firm, even if you do not formally solve the model.
>
> Thank you for the suggestion. While our core model assumes a passive institution applying a fixed, group-blind selection rule, the extended framework mentioned above (in Question 3c) naturally accommodates *strategic* settings—where the institution selects group-dependent mappings of the form $m_\ell(s) = x_\ell \cdot s + y_\ell$ to influence contest outcomes.
>
> In this view,  $x_\ell$ and $y_\ell$ serve as *strategic choices* that determine how agent scores are mapped to merits across groups. Given our closed-form characterization of the NE in this setting, one can formulate optimization problems, e.g.,
>
> $\max_{x_1, x_2, y_1, y_2} \mathcal{S}(A(x_1, x_2, y_1, y_2))$ subject to $r_{\mathcal{R}}(A(x_1, x_2, y_1, y_2)) \geq \tau$,
>
> where $\mathcal{S}(A)$ denotes the social welfare of the selected set under NE policy $A$, $r_{\mathcal{R}}(A)$ is the representation ratio, and $\tau \in [0,1]$ is a fairness threshold.
>
> While this lies beyond the scope of our current analysis, our results make such design problems tractable, and we will note this as an avenue for future work.
>
> Thanks again for your detailed feedback.

---

> > ### Comment · Reviewer_H86j · 2025-08-08
> >
> > I appreciate these clarifications and think that adding the discussed extensions would benefit the paper.

---

### Decision · Program_Chairs · 2025-09-17

**Decision:**

Accept (spotlight)

**Comment:**

The paper studies game theoretic dynamics in contests where agents have valuation asymmetries (even when they have identical abilities). They show how group-level disparities arise from just this factor, even with unbiased information and uniform selection criteria over all groups, where they consider Nash equilibria as their solution concept. They make the key assumption that group sizes are fixed and known, which is the case in some selection contests in the real world (e.g., admissions and hiring).

Strengths:
- The paper studies a well-motivated and interesting topic. Its model is clean and succinct, and the Nash equilibrium solution is analytically clean and fits the line of inquiry.
- The authors present their work well and provide high-level intuition of key results.

Weaknesses:
- Some of the reviewers brought up good points about insufficiently deep literature review (rBtT), ambiguous wording of claims (H86j, uD1e), and confusing terminology (all); the authors promised to address these in the camera-ready version.
- H86j ad rBtT aptly note that the stylized model is also probably not sufficient to capture real-world dynamics in the setting they consider. The authors rebut this by pointing out that they ran simulations to buttress their theoretical results.